# A unified diagrammatic approach to topological fixed point models

**Andreas Bauer[1⋆], Jens Eisert[1] and Carolin Wille[2]**

**1** Freie Universität Berlin, Arnimallee 14, 14195 Berlin, Germany
**2** University of Cologne, Zülpicher Straße 77, 50937 Cologne, Germany

⋆ andibauer@zedat.fu-berlin.de

## Abstract

We introduce a systematic mathematical language for describing fixed point models and apply it to the study to topological phases of matter. The framework is reminiscent of state-sum models and lattice topological quantum field theories, but is formalised and unified in terms of tensor networks. In contrast to existing tensor network ansatzes for the study of ground states of topologically ordered phases, the tensor networks in our formalism represent discrete path integrals in Euclidean space-time. This language is more directly related to the Hamiltonian defining the model than other approaches, via a Trotterization of the respective imaginary time evolution. We introduce our formalism by simple examples, and demonstrate its full power by expressing known families of models in 2+1 dimensions in their most general form, namely string-net models and Kitaev quantum doubles based on weak Hopf algebras. To elucidate the versatility of our formalism, we also show how fermionic phases of matter can be described and provide a framework for topological fixed point models in 3+1 dimensions.

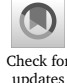

# 1 Introduction

The *phase* of a physical model is central to understanding its qualitative properties. Studying quantum phases of matter has been a major task in physics ever since quantum many-body theory was first formulated. This work addresses the question of classifying phases, that is, predicting which phases exist and providing models for each phase. Intuitively speaking, a phase of matter is an equivalence class of models under local restructuring of the degrees of freedom, as well as continuous changes which do not change the thermodynamic-limit behaviour of the model. Over the last more than a quarter of a century, condensed matter physicists have discovered a wealth of new exotic phases of matter, some of them reflecting collective states of interacting quantum systems that share little resemblance with the solids, liquids and gases of our commonplace experience. The study of quantum phases of matter has gained considerable momentum since the discovery of phases other than symmetry-breaking phases, known as *topologically ordered phases* [1].

The equivalence of two models under local restructuring of the degrees of freedom implies that any quantity we can measure, any defect we can embed, any coupling to another model we can write down, etc., corresponds to a quantity, defect, coupling, etc. for the other model. As a consequence, models in the same phase share many important properties, such as, e.g., their potential usability for the storage and processing of quantum information, their simu-latability by a classical computer, or their behaviour concerning thermalisation. That is to say,

the quest for a solid understanding of phases of matter draws also inspiration and motivation from practical and technological considerations.

The aim of this work is to classify phases of matter directly on a microscopic level using so-called *fixed-point models*, i.e., models that are exactly solvable due to having a zero correlation length. We hence do not aim at proving a classification directly from a non-fixed-point definition of phases, but make use of two clear assumptions: First, we assume that every phase possesses a fixed-point model, and second, we assume that the fixed-point model can be *topologically extended*, which means it can be defined on arbitrary manifolds and is invariant under a notion of homeomorphism. We formalise this using the language of *tensor-network path integrals*, which are tensor networks living in Euclidean spacetime. The single defining feature of tensor-network path integral fixed-point models will be the fulfilment of tensor-network equations representing a combinatorial analogue of homeomorphisms of spacetime manifolds. The focus of this work is the systematic construction and comparative study of tensor-network path integral fixed-point ansatzes, referred to as *liquids*. Different *topological liquids* provide ansatzes for fixed-point models in different dimensions, as well as for different microscopic degrees of freedom and different geometries of the tensor-network path integral in spacetime. *Liquid models* can not only describe bulk phases, but also phases of boundaries, domain walls, anyon worldlines, and arbitrary other defects.

Let us contrast and compare our formalism with existing approaches to the classification of phases of matter. In particular in the more mathematically inclined community one approach is to study abstract data not describing the model directly but only its defects, corresponding to what is known as a (non-fully extended axiomatic) *topological quantum field theory (TQFT)* [2]. Most famously, this includes modular tensor categories (MTCs) describing the anyon data for topological phases in $2+1$ dimensions [3], but also cobordism data for invertible phases [4]. Those approaches have been specifically successful in targeting also chiral phases. However, they have the problem that it is hard to know whether given defect data is realised by any microscopic model, and if so, whether it uniquely specifies the microscopic phase. In contrast, our work directly addresses explicit microscopic realisations of phases while also providing a high level algebraic classification. Another well established approach is to study microscopic *commuting-projector fixed-point models* such as the Levin-Wen string-net models [5]. While the importance of providing explicit fixed-point Hamiltonians is undisputed, the latter suffer from the short coming that the choice and justification of several properties of those Hamiltonian (as e.g. rotation or reflection symmetries of the input data) models is a bit ad-hoc and does not follow entirely from fundamental assumptions as is shown by the fact that for any construction there is a whole series of subsequent work generalising the latter step by step. In contrast, in our formalism, the models only have one single simple defining property, namely topological invariance, and with a bit of geometric intuition it is straight forward to see what the most general form of a construction is. A third approach is given by *state-sum constructions*, or equivalently *lattice TQFT* [6]. There one starts with some algebraic structure (such as a fusion category for the Turaev-Viro-Barrett-Westbury construction [7,8]), and then constructs a discrete partition function with weights depending on local variables on a triangulation. This clearly has the most similarity with our approach, but there are also several differences. First, we use the language of tensor networks to formalise state-sums, which equips us with a rigorous and clear diagrammatic notation to express different state-sum ansatzes. Second, we turn the line of reasoning around: Instead of starting with an algebraic/categorical structure, we start by directly writing down the equations corresponding to topological invariance, and only compare the result with existing algebraic structures to get inspiration for possible solutions to the equations. Third, we systematically look at the construction of new fixed-point ansatzes (liquids) which are not directly based on triangulations. A last approach is the study of so-called *MPO-injective PEPS* [9]. Even though our approach is based on tensor networks

as well, the kind of tensor network is quite different from our approach: Whereas *projective entangled-pair states (PEPS)* parametrise a (ground) state of a quantum system by a tensor network in physical space with open *physical indices* and *virtual indices* contracted between the tensors, our tensor-network path integrals in Euclidean spacetime describe a model by its imaginary-time evolution and have open indices only at a *spacial boundary*.

There are two main goals and achievements of our manuscript. The first is to establish the general formalism, and the second is to provide concrete results obtained within the formalism. Let us start with the first area, which is the main focus of this paper, and gives it partly the character of a review, even though for that, the presentation differs quite significantly from other literature. One major point is that our approach does not require any sophisticated maths, all we need is finite arrays of real or complex numbers, Einstein summations and Kronecker products. Therefore, we hope that this text is understandable for a very broad physics and mathematics audience. Moreover, we would like to stress that the logical structure of the formalism is extremely simple, and the only properties defining our fixed-point ansatz is topological invariance, along with a *Hermiticity condition* for standard quantum mechanical models, as well as a combinatorial *spin-statistics relation* [10] for models with fermionic degrees of freedom. Therefore, our line of reasoning makes it easy to understand and generalise existing fixed-point constructions as far as possible. We also demonstrate how to construct new fixed-point ansatzes with a little bit of creativity and geometric intuition. A central technical tool in our formalism are so-called *liquid mappings* which formalise diagrammatic relations between different fixed-point ansatzes and allow us to formalise a notion of equivalence. Different, but equivalent fixed-point ansatzes capture the same phases, but the models representing a phase might be simpler or practically more useful in one ansatz compared to another. However, it is also possible to construct new fixed-point ansatzes which potentially capture more general phases, as we argue in follow-up work [11, 12]. This might give rise to a unified microscopic description of phases with gapped boundary and phases without. We also make it easy to switch between different types of matter of physical description, such as classical/quantum, with/without symmetries, fermionic, etc., by simply interpreting the tensor-network diagrams using a different *tensor type* [13].

Some more concrete achievements are the following. We propose a very simple fixed-point path integral on cellulations corresponding to the Kitaev quantum double for weak Hopf algebras, which has not been known previously. In Refs. [14, 15] a generalisation of quantum double commuting-projector models to (weak) Hopf algebras was formulated, and their commuting-projector properties were derived. However, a very direct explanation of why weak Hopf algebras are used in the first place, and which extra properties we should demand, is still lacking. We provide such an explanation, showing that a combinatorial topological invariance is the only property needed to write down all the axioms of those models correctly. Our approach yields a direct geometric/topological interpretation for the axioms of weak Hopf algebras and makes it very clear which additional axioms we need to impose, such as both the algebra and co-algebra being special symmetric Frobenius algebras. Complicated Hopf algebra calculations using Sweedler's notation become much more human-readable using tensor-network notation, and can furthermore easily be verified using the geometric/topological interpretation we provide. Finally, the equivalence of string-net and quantum double models boils down to simple geometric constructions mapping cellulations to triangulations and vice versa. On a different note, it was also realised in Ref. [16] that in a way classifying topological phases on a certain level of abstraction is as simple as to write down *Pachner-move invariant simplex tensors*. However, there were two things lacking so far to make this point of view precise, namely the role of the branching structure as well as additional weights such as the ones coming from the quantum dimensions in the Turaev-Viro-Barrett-Westbury state-sum. We give a precise argument of why a branching-structure dependence is necessary to get all non-

trivial fixed-point models, and an exact explanation of where potential weights come from can be found in follow-up work [11]. We develop a technique to incorporate changes of the branching structure into the general topological invariance, without getting a huge number of complicated tensor-network equations. To this end, we extend the liquid from triangulations to cellulations involving certain non-simplex cells, and introduce branching-structure invariance via small auxiliary moves involving those non-simplex cells. We demonstrate this technique in $1+1$, $2+1$, and $3+1$ dimensions. As a last point, we propose a new fixed-point ansatz in $3+1$ dimensions. It corresponds to an algebraic structure consisting of a (special symmetric Frobenius) algebra and a fusion category, such that the algebra and category are related in a way similar to how the algebra and co-algebra are related in a bi-algebra.

An outline of this work is as follows. While this paper clearly focuses on fixed-point models, Section 2 describes the relation between our fixed-point path integrals to realistic models coming from condensed-matter physics. To this end, we propose several new definitions and conjectures on phases in tensor-network path integrals and their relation to Hamiltonian models via Trotterization. Section 3 then introduces the first examples of liquids as fixed-point ansatzes, omitting all but the necessary technical details. Sections 4, 6, and 5 then step by step add technical details and new concepts while still remaining within the simple case of $1+1$ dimensions. Section 8 applies the obtained concepts to develop two fixed-point ansatzes in $2+1$ dimensions closely related to the string-net and quantum double models. Section 9 demonstrates how fermions are incorporated into the setting, by both using fermionic tensors and introducing a combinatorial spin structure. Finally, in Section 10, we write down two new fixed-point ansatzes in $3+1$ dimensions.

## 2 Phases in tensor-network path integrals

### 2.1 Physical systems as tensor networks

At the heart of our approach are tensor networks. A *tensor* (in the conventional sense) is simply a multi-dimensional array $A_{i,j,l,\dots}$. Each of the *indices* $i,j,l,\dots$ can take a finite number of values, e.g., $i \in \{0,\dots,n_i-1\}$, where $n_i$ is called the *bond dimension* of the index. The values can be real or complex numbers. The following two operations on tensors are everything one needs to know to understand the tensor-network aspect of this work. On the one hand, the *tensor product* is the entry-wise product of two arrays, yielding an array with indices from both arrays, acting as

$$(A \otimes B)_{i,j,\dots,k,l,\dots} = A_{i,j,\dots} \cdot B_{k,l,\dots} . \tag{1}$$

On the other hand, the *contraction* is the Einstein summation over two indices with the same bond dimension, yielding a tensor with those two indices removed, acting as

$$A_{i,j,l,\dots} \quad \mapsto \quad ([A]_{i,l})_{j,\dots} = \sum_x A_{x,j,x,\dots} . \tag{2}$$

Following the familiar Penrose notation, a tensor can be graphically represented by a box with one line sticking out for each index, e.g.,

$$
\begin{array}{c}
j \\
| \\
k - \boxed{A} - i \\
\dots
\end{array}
. \tag{3}
$$

A computation using tensor products and contractions can be represented by a network-like diagram. For every (copy of a) tensor, we draw (a copy of) the corresponding shape at an

arbitrary location (possibly rotated or reflected). For every contraction between two indices, we connect the corresponding lines. E.g., the computation

$$\sum_{x,y,z,w} A_{x,y,i,z} A_{j,y,x,w} B_{w,z,k} \tag{4}$$

could be represented by

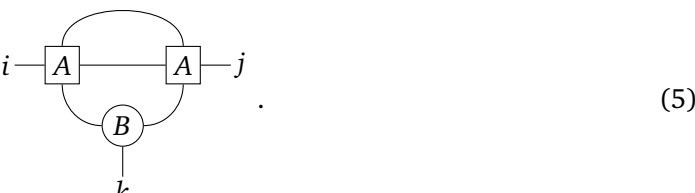

$$\tag{5}$$

In general, we might also use shapes without labels for tensors. For a more detailed introduction into tensor networks and the corresponding notation, see, e.g., Section 2.1 in Ref. [17]. We will refer to a purely combinatorial structure of a tensor-network diagram as a *network*. The lines corresponding to the contractions will also be called *bonds*, and the uncontracted indices are called *open indices*. Sometimes it will be instructive to distinguish between a tensor itself (consisting of the actual data, i.e., the explicit array), a *tensor variable* occurring in a tensor-network diagram without any data (such as $A$ and $B$ for the diagram above), and a *tensor copy* occurring in a diagram (such as either of the two copies of $A$ in the diagram above, or the single copy of $B$). In cases where there is unlikely going to be confusion, we will often refer to all three things as tensor. Similarly, we can distinguish between different bond dimensions (the actual number), and *bond dimension variables* which will sometimes be indicated by different line styles for the indices in networks. Indices of tensor variables associated to the with the same bond dimension variable must have the same bond dimension (but not necessarily vice versa), and can therefore be contracted with each other. We will often sloppily refer to the different bond dimension variables simply as different bond dimensions and believe this will rather avoid than create confusion.

It is important to stress that we do *not* use tensor networks in the usual way they are used to describe many-body systems, namely for representing a (ground) state of a local Hamiltonian. For a quantum many-body model in $d$ spacial dimensions, the latter are $d$-dimensional tensor networks in physical space, known as *PEPS* (or *MPS* for $d = 1$) [18]. The degrees of freedom correspond to open (uncontracted) *physical indices*, whereas *virtual indices* are contracted neighbouring tensors. In contrast, the tensor networks we use are *tensor-network path integrals*. Those are tensor networks which do not have a distinction between virtual and physical indices, and open indices only occur at places where we choose to cut the network at a one dimension lower *open boundary*. A diagram of such a tensor network could look like

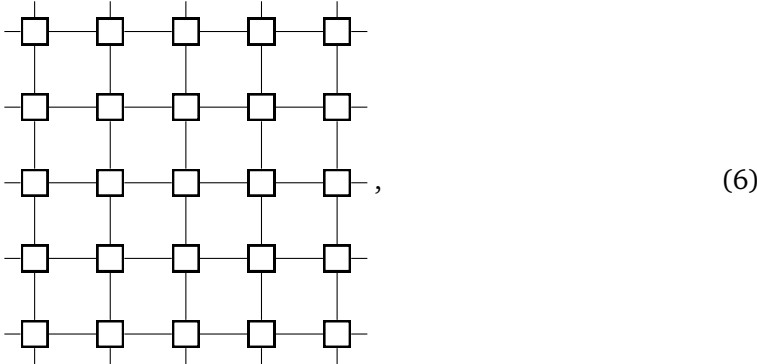

$$\tag{6}$$

with open indices only at the 1-dimensional open boundary but no open indices inside the 2-dimensional bulk. For the tensor-network path integrals we have in mind, all the tensors are

the same, i.e., the model is translation invariant and given by a single tensor, or at least by a finite system-size independent number of tensors. This allows us to speak of a thermodynamic limit and of phases of matter.

Tensor-network path integrals provide a unified way to formalise basically any physical model with a notion of space and locality. For describing the real-time dynamics of a many-body quantum model, those tensor networks are simply geometrically local quantum circuits living in spacetime, which are a discrete analogue of the continuous real-time evolution under a Hamiltonian. Also the classical statistical evolution of a local Markovian process can be formalised by such a spacetime tensor network. It is easy to see that also thermal classical statistical thermal systems such as the classical Ising model are described by tensor-network path integrals living in space only, without any need for approximation. With approximation, this holds also for thermal quantum states. Moreover, ground state properties of a quantum model are captured by a tensor-network path integral approximating the imaginary-time evolution living in Euclidean spacetime. Also very different types of models like fermionic or single-particle models can be captured by tensor-network path integrals, if we do not use arrays of complex numbers for the individual tensors, but consider more general *tensor types* [13].

What does it mean for a model to be represented by a tensor-network path integral? Ultimately, the predictions we obtain from a model are the results of local measurements. For a quantum model, the results are the statistics of the measurement outcomes, which are probability distributions and thus themselves tensors. Also the measurements themselves are specified by tensors, in the quantum case the *positive operator valued measures* (POVM) corresponding to the measured observables. The outcome tensor can be obtained from the POVM tensor and the tensor-network path integral in a simple way: We insert the measurement tensors into the tensor-network path integral and evaluate it. E.g., a POVM acting on one degree of freedom is given by a vector of density matrices, hence a 3-index tensor. If we want to obtain the joint probability distribution over outcomes of a 2-point measurement of the thermal state of a local spin Hamiltonian in 1 spatial dimension, we have to evaluate a tensor-network like the following,

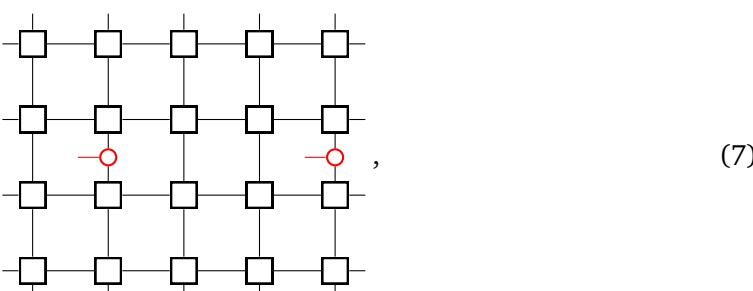

$$\tag{7}$$

with periodic boundary conditions. The two red tensors are the POVMs, and their classical indices stay open. The resulting tensor describes the probabilities of pairs of measurement outcomes at both places. Every row of tensors corresponds to the imaginary time evolution with a fixed inverse temperature $\beta$, so the described thermal state has inverse temperature $4\beta$. If we scale the network in both the horizontal spatial direction and the vertical time direction, we will obtain ground-state measurement outcomes in the thermodynamic limit.

The tensor-network path integrals which can exhibit topological phases are those describing (classical or quantum) thermal models, but most importantly those representing ground state properties via the imaginary-time evolution.

## 2.2 Trotterization

As mentioned in the previous section, tensor-network path integrals can approximate the real- or imaginary time evolution under a local quantum many-body Hamiltonian. This can be

done via *Trotterization*, which is a standard technique in the field of numerical tensor-network methods using PEPS or MPS. Here we do not use Trotterization to calculate the time evolution applied to a state, but to obtain a tensor-network path integral for the time evolution itself. In this section, we will concentrate on the simple case of a 1-dimensional quantum spin chain with a translation-invariant nearest-neighbour Hamiltonian

$$H = \sum_i h_{i,i+1}, \tag{8}$$

where $h_{i,i+1}$ is a Hamiltonian acting on two degrees of freedom, applied the spins $i$ and $i+1$. Global properties such as e.g. the boundary conditions are not relevant for the following considerations, which are applied to some small patch in the bulk of the translation-invariant model. The generalisation of the presented methods to higher dimensions or other local geometries of Hamiltonian terms is straight-forward.

We can divide the Hamiltonian terms into the ones acting on even-odd site pairs and those acting on odd-even site pairs as

$$H = \sum_i h_{2i,2i+1} + \sum_i h_{2i+1,2i+2} = H_1 + H_2. \tag{9}$$

All terms within $H_1$ act non-trivially only on non-overlapping sets of spins and therefore commute. The same holds for $H_2$. $H_1$ and $H_2$ however do not commute, and therefore

$$e^{it(H_1+H_2)} \neq e^{itH_1} e^{itH_2}. \tag{10}$$

We can still use the following *Suzuki-Trotter expansion*

$$e^{it(H_1+H_2+\dots)} = \lim_{n\to\infty} \left( e^{i\frac{t}{n}H_1} e^{i\frac{t}{n}H_2} \dots \right)^n \tag{11}$$

to obtain

$$e^{itH} = \lim_{n\to\infty} \left( \prod_i e^{i\frac{t}{n}h_{2i,2i+1}} \prod_i e^{i\frac{t}{n}h_{2i+1,2i+2}} \dots \right)^n. \tag{12}$$

Consider the expression on the right-hand side for a fixed $n$. It is a product of local operators acting on a chain of degrees of freedom. $e^{ith/n}$ is a linear operator acting on two spins, so it is a tensor with 4 indices, two for both input and output of the operator,

$$e^{i\frac{t}{n}h} \quad \to \quad \text{⊗}. \tag{13}$$

With this interpretation, the product of operators becomes a tensor network. E.g., for $n = 3$, we get the network

$$\tag{14}$$

The tensor network we are looking for should have the same notion of locality structure as the continuum time evolution. That is, every tensor should correspond to a finite space-time volume $\Delta x \times \Delta t$. So it does not make sense to directly take the above tensor network, as the time interval corresponding to a tensor scales like $1/n$. In addition, this tensor network has a trivial limit for $n \to \infty$

$$\text{⊗} = e^{i\frac{t}{n}h} \quad \xrightarrow{n\to\infty} \quad \mathbb{1} = \Big| \, \Big|. \tag{15}$$

Instead, we have to pick a fixed $\Delta t$, Trotterize $e^{i\Delta tH}$ for some $n$ and divide the resulting tensor networks into spatial unit cells. Evaluating the whole tensor-network patch inside the unit cell yields the tensor $P_n$ of the tensor network we are looking for. E.g., for $n = 3$ and $\Delta x$ consisting of 4 sites, we can choose

$$
abcdef - \boxed{}_{P_3}\, {}^{v'w'x'y} _{vwxy} - a'b'c'd'e'f' = \qquad . \tag{16}
$$

For sufficiently large $n$, the square lattice tensor network

$$
\cdots \boxed{P_n}\,\boxed{P_n}\, \cdots \atop \cdots \boxed{P_n}\,\boxed{P_n}\, \cdots \tag{17}
$$

then approximates the (imaginary) time evolution of the local Hamiltonian. However, $P_n$ does not have a sensible large $n$-limit as well. If we let $n \to \infty$, the number of indices to block on the right and left, and therefore the bond dimension of those indices, grows with $n$. More precisely, the number of blocked indices grows linearly, and thus the bond dimension increases exponentially with $n$.

Luckily, experience from numerical algorithms performing time evolution with MPS/PEPS suggests that this exponentially growing bond dimension can be truncated to a much smaller bond dimension with only a very small approximation error. Specifically, in numerical algorithms like iTEBD which apply and truncate the Trotterized time evolution to an MPS, we find that the state after a *fixed* time $\Delta t$ is well approximated by an MPS with bond dimension independent of the system size (although for very large $\Delta t$ this bond dimension might be very large). Now the crucial observation is that how large the truncated bond dimension is turns out not to depend essentially on the number $n$ of Trotter steps in which we decompose $\Delta t$, even though the bond dimension would grow exponentially in $n$ without truncation. This suggests that also the MPO given by the Trotterized $\Delta t$ time evolution itself can be truncated from something growing exponentially in $n$ to something essentially independent of $n$. In particular, if we apply the time evolution to each first qu-$d$-it of a product of bell pairs, then the result is the MPO describing the time evolution itself.

At this point we would like to stress that this consideration is completely different from the question whether ground states are well approximated by MPS, which has been famously conjectured for *gapped* Hamiltonians, and proven in some formulation in $1 + 1$ dimensions. Ground states are obtained by applying the imaginary time evolution for a time $t$ and then taking the limit $t \to \infty$. In contrast, we only consider a fixed time interval $\Delta t$ which we do not ever think of scaling. For such a constant interval, we believe that the time evolution can be truncated independent from whether the Hamiltonian has a spectral gap and whether we consider the imaginary- or real time evolution.

In order to make Trotterization into a precise conjecture, we need to take the continuum limit $n \to \infty$. The tensors $P_n$ for different $n$ have different bond dimensions for the indices on the left and right, and should be considered as vectors in different vector spaces. In order to take the limit, we need to embed all of those vector spaces into one common infinite-dimensional vector space. To define convergence, this vector space needs to be equipped with

a norm. In order to make sense of tensor networks formed from those infinite-dimensional tensors, we the norm needs to be defined for tensors with multiple infinite-bond-dimension indices, and the contraction and tensor product should be both well-defined and continuous as a (bi-)linear functions. In other words, the infinite-dimensional tensors need to form a tensor type [13]. Possible norms which define a tensor type are, e.g., the entry-wise 1-norm,

$$\|T\| = \sum_{a,b,c,\dots} |T_{abc\dots}| \,, \tag{18}$$

or a norm enforcing a $\frac{1}{n}$ decay in every individual index,

$$\|T\| = \max_{a,b,c,\dots} \quad a \cdot b \cdot c \cdots |T_{abc\dots}| \,. \tag{19}$$

Valid tensors are those infinite-dimensional arrays for which the norm is finite.

In order to perform the embedding of the different $P_n$ into one shared infinite-dimensional normed vector space, we use the concept of invertible domain walls which will be introduced more thoroughly in Section 2.5. Roughly, an invertible domain wall between two tensor-network path integrals $A$ and $B$ is a way to rewrite $A$ as $B$ and vice versa via a small set of tensor-network equations. In our case, we want to rewrite the square-lattice tensor network given by $P_n$ into one given by some other $\widetilde{P}_n$. A simple type of invertible domain wall consists in inserting at every bond a resolution of the identity

$$a \overset{I^{-1}}{\underset{}{\square}} \overset{I}{\underset{}{\square}} a' = a \rule{1.2cm}{0.4pt} a' \,, \tag{20}$$

for some invertible matrix $I$. $P_n$ and $\widetilde{P}_n$ are then related by

$$a \overset{v'}{\underset{v}{\square}} a' = a \overset{}{\square} \overset{v'}{\underset{v}{\square}} a' \,. \tag{21}$$

Now, our Trotterization conjecture can be formalised as follows. There are tensors $\widetilde{P}_n$ inside some suitable normed (infinite-dimensional) shared vector space forming a tensor type, and an invertible domain wall between each $\widetilde{P}_n$ and $P_n$, such that the sequence $\widetilde{P}_n$ converges,

$$\widetilde{P}_n \to \widetilde{P} \,. \tag{22}$$

The square-lattice (infinite-bond-dimension) tensor network $\widetilde{P}$ then *exactly* represents the imaginary time evolution, in the sense that evaluating one row of it yields the exact time evolution operator. A finite bond dimension tensor network can be obtained by simply cutting $\widetilde{P}$ at a certain bond dimension, which then yields an approximation.

It is easy to see how to generalise the Trotterization procedure to other geometric setups, such as higher dimensions, higher spatial support of the Hamiltonian terms, or presence of boundaries or defects of any kind. First we divide the terms into a constant (system-size independent) number of subsets, such that the terms in one subset all commute with each other. Then we proceed using the Suzuki-Trotter expansion applied to the division into subsets, resulting in a tensor network, which we block and truncate into finite unit cells.

## 2.3 Gapped-path and local-unitary phases of matter for Hamiltonians

In this section, we quickly review conventional notions of phases in Hamiltonian systems. So-called quantum phases of matter are defined as equivalence classes of local translation-invariant gapped Hamiltonians $H \in \mathcal{H}$. By *gapped*, we mean that there is an integer $g \geq 0$

called *ground state degeneracy* and a real number $\epsilon > 0$ called the *gap*, such that for every system size $n$ (greater than some $n_0$), the $g$ lowest eigenvalues of $H$ are separated from the rest of the spectrum by at least $\epsilon$, and among each other by $\beta_n$ such that $\beta_n \to 0$ for $n \to \infty$. Two gapped Hamiltonians $H_1, H_2 \in \mathcal{H}$ are considered equivalent if there is a continuous path of gapped Hamiltonians connecting $H_1$ and $H_2$ [1],

$$
\begin{aligned}
\widetilde{H} : [0,1] &\to \mathcal{H} \,, \\
\widetilde{H}(0) = H_1, \quad &\widetilde{H}(1) = H_2 \,.
\end{aligned}
\tag{23}
$$

Recall that $\mathcal{H}$ contains only gapped Hamiltonians, so all $\widetilde{H}(s)$ for $s \in [0,1]$ must be gapped, otherwise one speaks of a "gap closing" inducing a "phase transition". If we aim at comparing two Hamiltonians with different local Hilbert spaces, we can arbitrarily embed both into a shared local Hilbert space and use the same definition.

The importance of models with a spectral gap comes from the fact that they are 'generic' in the following sense. It is believed that for any direction of perturbation, there is a non-zero perturbation strength such that the perturbed model remains gapped for all perturbations of smaller strength. This has been proven so far only for perturbations around fixed-point models of topological order [19]. Consequently, the set of gapped models is an open subset of the space of all models, and in any few-parameter family of models (i.e., in any phase diagram), there are either no gapped models at all or they have a non-zero volume.

For generic local many-body Hamiltonians it is very hard to tell whether they possess a spectral gap, and the general question of the existence of a gap of a certain size has even shown to be algorithmically undecidable for certain families of Hamiltonians [20]. However, there is a very simple family of Hamiltonians for which it is very simple to verify the spectral gap, namely *commuting-projector Hamiltonians*, where the Hamiltonian is $-1$ times the a sum of geometrically local projectors which mutually commute. Such Hamiltonians can be solved exactly analytically, have zero *correlation length*, and are therefore often referred to as *fixed-point models*. Many topological phases (presumably all with gapped boundary) possess representatives which are fixed-point models.

The gapped-path definition is rather unconstructive which makes it very hard to classify phases of matter. To naively show that two Hamiltonians are in a different phase, we would have to look at all different continuous paths connecting them inside the infinite-dimensional space of local Hamiltonians, and assert whether an algorithmically undecidable property holds for the models on each path. There is an alternative definition which is much more constructive: Two Hamiltonians are in the same phase if their ground states are related via a finite-depth local (generalised) unitary circuit [21] (note that this definition is usually used to define phases of states on their own, however, we need to be careful to define what a "state" even means in the context of a thermodynamic limit). This definition replaces the continuous path over tensors by a single set of tensors (forming the circuit). Furthermore, it makes the physical meaning of phases very clear: Two states in the same phase are the same up to locally restructuring their degrees of freedom.

It is important to point out that the two presented definitions are *not* equivalent. In one direction, it is easy to see that a local unitary circuit gives rise to a continuous gapped path by conjugating the Hamiltonian with $e^{itV}$ from $t = 0$ to $t = 1$ where $e^{iV} = U$ for each layer $U$ of the local unitary circuit. However, conversely, a local unitary circuit cannot change the *correlation length* of a model, which is possible via a gapped path. If we want to make this possible in the local-unitary definition as well, we have to allow for some approximation error in the local unitary equivalence, which has to decrease with the depth of the circuit or the size of the support of the individual local unitaries. Alternatively, we can directly use a "fuzzy circuit", corresponding to the time evolution under a local time-dependent Hamiltonian [22]. In this case, the equivalence has been shown using the notion of quasi-adiabatic evolution [23].

Even though the gapped-path definition is more general than the exact local-unitary definition, the latter seems to be working if we restrict ourselves to fixed-point models such as commuting-projector models. There, two models in the same gapped-path phase are usually observed to be equivalent via a local unitary circuit, and giving the latter is the most common way of proving phase equivalence of models.

## 2.4 Spectral gap and phases in tensor-network path integrals

In this and the next section, we will find natural definitions of phases of matter in terms of tensor-network path integrals, by transferring the definitions from the previous section to this setting. Before we start with that, let us first argue why tensor-network path integrals are a particularly natural representation of quantum systems for the study of quantum phases of matter. The latter describe the ground state properties of Hamiltonians. Such ground states can be obtained directly from the Hamiltonian by applying the imaginary time evolution to some initial state vector $|x\rangle$ as

$$\lim_{\beta \to \infty} e^{-\beta H} |x\rangle. \tag{24}$$

However, for any finite system size, the lowest eigenvalue will not generically have a $g$-fold degeneracy, but the "ground states" will have slightly different energies. So, at a particular system size, $e^{-\beta H}$ will not converge to a ground state projector with $g$-dimensional support, but to the projector on the lowest eigenstate only. We see that in order to talk about ground states, we do not only have to scale the imaginary time $\beta$, but also simultaneously the system size $n$. This simultaneous scaling of both imaginary time and space is elegantly featured by the imaginary-time evolution tensor-network path integral.

The gapped-path definition of phases carries over to the tensor-network path integrals as follows. A gapped Hamiltonian $H$ yields a gapped imaginary-time evolution operator $e^{-\beta H}$, just that now the gap separates the *largest*-magnitude values from the rest of the spectrum. In $n$ spacetime dimensions, consider an $n-1$-dimensional constant-time layer of a tensor-network path integral, such as

$$\cdots \!-\!\square\!-\!\square\!-\!\square\!-\!\square\!-\! \cdots \tag{25}$$

for a square lattice tensor network in $1+1$ dimensions. Suppose the tensor-network path integral comes from Trotterizing an imaginary-time evolution. Then this layer, as an operator from the bottom indices to the top indices, with $N$ tensors and with periodic boundary conditions connecting left to right, approximates $e^{-\beta H}$ where $\beta$ is the chosen discretization step in time direction. Tensor networks of this form are known as *projected entangled pair operators (PEPO)*, or *matrix product operators (MPO)* [18] in $n-1=1$ dimension, and are often called *transfer operator*.

From the tensor-network path integral point of view, this notion of gap seems a bit unnatural as it specifically involves the transfer operator in imaginary time direction. The path integral lives in Euclidean spacetime, so space and imaginary time should be treated on equal footing. Instead of demanding a spectral gap of the imaginary-time transfer operator, it is natural to demand a spectral gap for the tensor network along any $n$-dimensional curve in spacetime, interpreted as an operator from one to the other side. Unfortunately, this has the technical problem that the indices (and thus the Hilbert spaces) on the two sides of an arbitrarily curved transfer operator can be different, and hence we cannot talk about eigenvalues. In the following, we give a definition which we believe is the natural tensor-network path integral analogue to the spectral gap of Hamiltonians. It is based on the observation that for a gapped Hamiltonian for intrinsic robust topological order on a sphere, the normalised opera-

tor $e^{-\beta H}$ gets exponentially close to becoming a rank-1 operator in the 'width' $\beta$ of the transfer operator.

For a more formal definition in $n$ spacetime dimensions, we consider an *annulus network A*, i.e., a patch of a translation-invariant tensor network whose topology is $S_{n-1} \times [0, 1]$. We then look at its behaviour when its *width $d_A$*, given by the minimum number of bonds it takes to get from some open index at the inside- to some open index at the outside boundary, is increased. Let $I_A$ and $E_A$ denote the *inside boundary* and *outside boundary* of an annulus network $A$ and let $X[A]$ denote a tensor-network path integral $X$ evaluated on $A$. We can interpret $X[A]$ as a linear operator from the open indices on the inside boundary to those of the outside boundary.

Equipped with this terminology, we say that a tensor-network path integral $X$ has a *robust gap* with *correlation length* bounded by some $\xi > 0$, if the following holds.

- For any interior boundary $I$, there exists a pre-factor $C_I$ and two tensors $\langle V_I |$ and $|W_I \rangle$ whose open indices match those of $I$.

- For every annulus network $A$, and every tensor $\langle T |$ with indices matching those of $E_A$, we have

$$
\begin{aligned}
\big\| \langle T | X[A] &- \langle T | X[A] | W_{I_A} \rangle \langle V_{I_A} | \big\| \\
&< \| \langle T | X[A] \| C_I e^{-\frac{d_A}{\xi}} .
\end{aligned}
\tag{26}
$$

Note that the choice of norm in Eq. (26) does not matter as the corresponding vector space has a finite dimension only depending on $I$, so we can make up for possible changes of the norm by adapting $C_I$. Also note that the tensor $\langle V_I |$ in the definition is unique, whereas the choice of $|W_I \rangle$ is essentially arbitrary. Moreover, *the* correlation length of $X$ is the smallest $\xi$ for which $X$ has a robust gap. For tensor-network path integrals coming from a Trotterized Hamiltonian, a robust gap of the path integral implies a gap of the Hamiltonian. We neither know a proof nor counter-examples for whether the opposite implication is true.

As we announced, the definition says that $X[A]$ is approximately rank-1. However, we have formulated it in such a way that for fermionic tensors or tensors with symmetries, it suffices if $X[A]$ is rank-1 within the fermion-parity-even, or symmetric subspace, since $T$ is itself a symmetric/fermionic tensor. So the same definition can be used for symmetry-breaking phases, or fermionic phases such as the Kitaev chain. If $X[A]$ is exactly a rank-1 operator (possibly in the symmetric/parity-even subspace), then it is easy to see that we can find $W_{I_A}$ and $V_{I_A}$ such that the left hand side is 0. We will define *fixed-point models* accordingly, as models for which the correlation length $\xi$ is zero, that is, both sides of Eq. (26) are exactly zero for all $d_A$ larger than some constant. Being a fixed-point model is a very restrictive condition, and the tensors of such models are often found to satisfy exact algebraic relations, which enable us to analytically calculate some of their properties.

Equipped with the correct notion of a gap, the definition of phases on the level of tensor-network path integrals is completely analogue to the Hamiltonian case: Two tensor-network path integrals $X_0$ and $X_1$ with a robust gap are *in the same phase* if there is a continuous family of tensors $X(s)$, $s \in [0, 1]$, $X(0) = X_0$, $X(1) = X_1$, such that

- $X(s)$ has a robust gap for all $s$, and

- there is a choice of $s \mapsto \langle V_I | (s)$ which is continuous.

Note the second condition is necessary, e.g., to detect first order phase transitions, by which we mean transitions between two symmetry-broken sectors via a non-trivial symmetry-breaking phase.

## 2.5 Invertible domain walls and exact phases

In Section 2.3 we have seen that local unitary circuits provide a natural alternative definition of phases if we restrict ourselves to fixed-point models. In this section, we will give an analogue of the local unitary definition for tensor-network path integrals. We will do so by defining a notion of an *invertible domain wall* which generalises local unitary circuits, and we will call equivalence classes under such invertible domain wall *exact phases*.

To start with, let us consider a particularly simple case of a local unitary circuit between two ground states of Hamiltonians, namely an on-site unitary. If we conjugate Hamiltonian terms by a unitary operator, the tensors in the (Trotterized) imaginary-time evolution get conjugated in the same way. So, if we apply an on-site unitary $U^{\otimes N}$ to a $1+1$-dimensional model as in Section 2.1, the tensor $P$ in Eq. (17) gets conjugated by $U$,

$$\widetilde{P} = P \qquad (27)$$

The unitarity of $U$ can be denoted in network notation by

$$U^\dagger U = \qquad (28)$$

Imagine starting from the conjugated square lattice tensor network. Now, replace every occurrence of the conjugated tensor $\widetilde{P}$ by its definition in terms of Eq. (27). This will create a pair of $U$ and $U^\dagger$, that is, an occurrence of the left hand side of Eq. (28), at every bond of the original network. We can remove those pairs by replacing each occurrence with the right hand side of Eq. (28), yielding the non-conjugated network

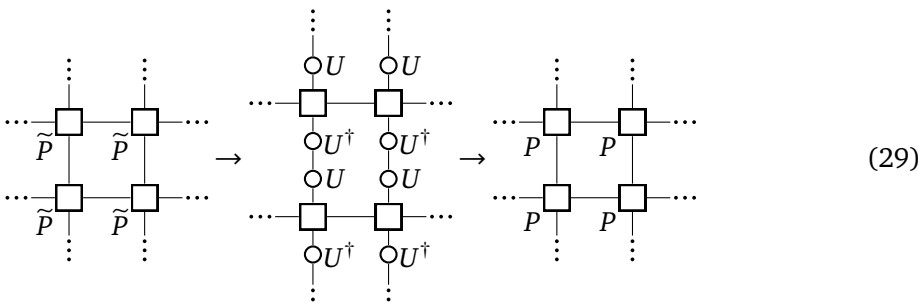

$$(29)$$

in Eq. (17). To summarise, we have defined two tensor-network equations in Eq. (27) and Eq. (28). Plugging in those equations locally, we can rewrite the tensor-network path integral $P$ as $\widetilde{P}$. It is easy to see that this implies that $P$ and $\widetilde{P}$ are really equivalent for practical purpose in the sense that for every measurement we take in $P$, there is an according measurement in $\widetilde{P}$ which yields the same result: We just need to also conjugate the measurement tensor (POVM) by $U$ as well. This simple example motivates the following general definition.

An *invertible domain wall* between two tensor-network path integrals $A$ and $B$ is defined by the following.

- There is a set of *domain wall tensors*.

- The tensors of $A$ and $B$ together with the domain wall tensors satisfy a set of tensor-network equations.

- Using the tensor-network equations we can transform the networks of $A$ into the networks of $B$ and vice versa.

If such an invertible domain wall exists, then *A* and *B* are said to be in the same *exact phase*.

We can also define an invertible domain wall if *U* is not an on-site operator, but is only a product of operators acting on constant-size non-overlapping patches. Furthermore, it suffices if *U* is an isometry rather than a unitary, such that it can also change the local Hilbert space dimension. Last, we can conjugate by more than one layer of unitaries. So for any two Hamiltonians related by a finite-depth generalised local unitary circuit, the corresponding tensor networks are related by an invertible domain wall. But invertible domain walls also go beyond conjugation by unitaries. Consider the following example for a different invertible domain wall acting on square lattices. First, we split each tensor into a network consisting of 4 tensors,

$$\underset{\widetilde{P}}{\square} \;=\; \diamondsuit\;. \tag{30}$$

The dimension of the bonds between the new tensors can be different, which we reflect by using a different line style. At every bond of the original network, we will get two of the new 3-index tensors. We replace those two by two other tensors, connected by a bond perpendicular to the old bond,

$$\gtrdot\!-\!\lessdot \;=\; \underset{\otimes}{\overset{\otimes}{\bowtie}}\;. \tag{31}$$

Now, at every vertex of the square lattice there are 4 tensors on the adjacent edges. We can block those into a single tensor again according to

$$\diamondsuit \;=\; \underset{P}{\square}\;. \tag{32}$$

Applying this invertible domain wall, we obtain a network whose tensors are now located at the positions where the vertices of the old square lattice have been previously,

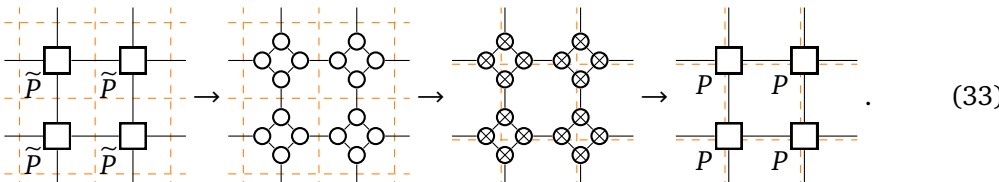

There is a second interpretation of invertible domain walls, other than transforming the tensor-network path integrals into each other by locally inserting tensor-network equations, which also explains the name. If we perform the transformation only in a part of the spacetime, we will obtain regions of *A* and regions of *B*, separated by some remaining tensors along a $d-1$-dimensional submanifold. This submanifold constitutes a domain wall between *A* and *B*. Unlike a domain wall in a physical condensed-matter model, the $d-1$-dimensional submanifold does not have to be constant in time direction, but can be arbitrarily curved in spacetime. By performing the transformations from *A* to *B* or vice versa next to the domain wall, we can arbitrarily move the domain wall around, making it a *topological* domain wall. By performing the transformation from *A* to *B* in the middle of a *A* region, we generate a small bubble of *B* within *A*. Also, we can connect two *A* regions separated by a thin bridge of *B* region. Those moves and their analogues in higher dimensions are what makes the domain wall *invertible*.

To illustrate the domain-wall picture, a particularly suitable example is given by a *simple MPO* separating two square-lattice tensor networks *A* and *B* (though the examples already

given would work as well). This invertible domain wall consists of two additional tensors,

$$
\tag{34}
$$

which together with $A$ and $B$ fulfil the equations

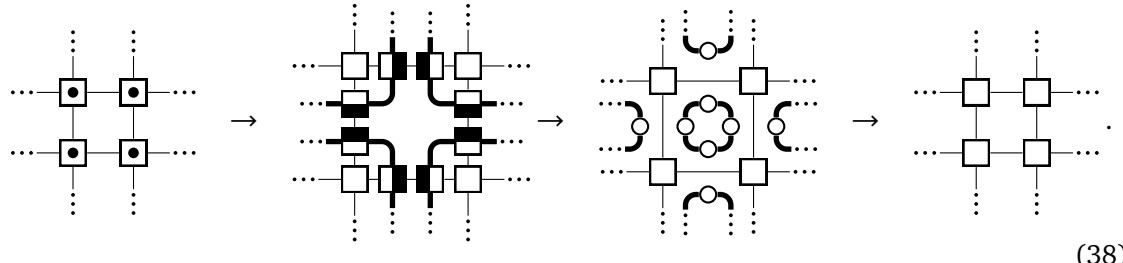

$$
\tag{35}
$$

$$
\tag{36}
$$

and

$$
\tag{37}
$$

The empty right-hand side of the last equation denotes the scalar 1.

As for the examples given before, we can again apply those equations to transform one tensor-network path integral into the other,

$$
\tag{38}
$$

We first apply Eq. (35) to every tensor of the $B$-network, then Eq. (36) at every bond of the original $B$-network, and then Eq. (37) at every plaquette of the original $B$-network, yielding the $A$-network.

On the other hand, if we apply the equations only on one side, we obtain a domain wall between the two path integrals,

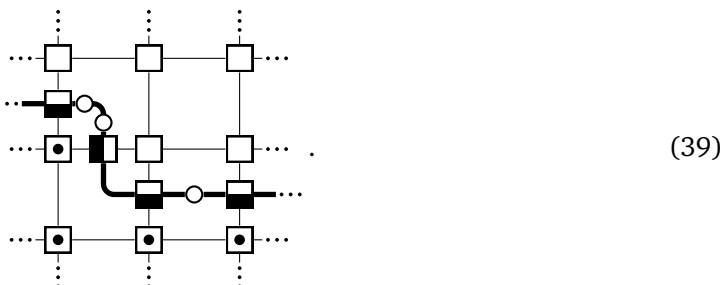

$$
\tag{39}
$$

As we see the domain wall does not have to be a straight line, but can be arbitrarily curved. For our concrete example, there are additional internal bonds of the domain wall (thick lines) which can be straight or bent. Straight bonds include one normalisation matrix, and bonds bending around a $B$-tensor include twice the normalisation matrix. The moves in Eq. (35),

Eq. (36), and Eq. (37) can be used to move the location of the domain wall. E.g., in our example, the invertible domain wall can be 'pulled through' an *A*-tensor yielding a *B*-tensor,

$$\tag{40}$$

where we used first Eq. (37), then twice Eq. (36), and then Eq. (35). Other moves that make the domain wall invertible, are for example given by creating or erasing small loops via Eq. (35), and fusing two domain wall lines via Eq. (36).

As for the Hamiltonian definitions, the gapped-path and invertible-domain-wall definitions for phases in tensor-network path integrals are not equivalent. Again, invertible domain walls of tensor-network path integrals cannot change the correlation length in contrast to gapped paths. So, two models in the same gapped-path phase do not have to be related by an invertible domain wall, and in fact generically they are not.

However, invertible domain walls might very well be more general than local unitary circuits, and therefore, in contrast to the Hamiltonian definitions, not even the converse might be true. Namely, it is not a priori clear whether an invertible domain wall between two path integrals *A* and *B* can be turned into a continuous gapped path. In some cases we can find a model for an invertible domain wall between *A* and *A* of the same form, and a continuous family of domain walls connecting it to a desired invertible domain wall between *A* and *B*. E.g., consider domain walls relating *A* and *B* by conjugation with an on-site unitary *U*. Taking $\mathbb{1}$ for *U* defines an invertible domain wall between *A* and *A*, and for every *U* we can find a Hermitian *H* such that $U = e^{iH}$. $\hat{U}(s) = e^{isH}$ then defines a continuous family of invertible domain walls interpolating between $\hat{U}(0) = \mathbb{1}$ and $\hat{U}(1) = U$. On the other hand, let us restrict to real tensors which is what we need to model spin systems with a time-reversal symmetry, and consider an invertible domain wall consisting of on-site orthogonal maps *O*. If *O* is has determinant $-1$, i.e., it involves a reflection, we cannot interpolate between $\mathbb{1}$ and *O* by a continuous path as before. So in this case, it is unclear whether we can interpolate between the original model and the *O*-conjugated model with a gapped path, without breaking the time-reversal symmetry.

In practice, invertible domain walls are an accepted way for defining phases for fixed-point models, even though their relation to gapped paths has not been rigorously settled. E.g., the phase equivalence of certain Levin-Wen models via the Morita equivalence of the underlying fusion categories is an example of an invertible domain wall.

## 2.6 Fine-graining/renormalization of tensor-network path integrals

The main body of this work is about the classification of phases via fixed-point models. A priori, it is unclear whether every phase possesses such a fixed-point model. A vague argument of why this is the case is provided by the idea of *renormalization group (RG) flow*. As this term as such is rather uninstructive and used for too many distinctly different concepts, we will use to the simpler name *fine-graining mapping*. Given a tensor-network path integral *T*, we can construct a new path integral $T_\lambda$ by taking blocks whose linear size $\lambda$ is called the *fine-graining scale*, and grouping them together into a single tensor each. E.g., for $\lambda = 3$ and a square-lattice

tensor network in $1 + 1$ dimensions, we define a new tensor by

$$
\begin{array}{c}
b_1 b_2 b_3 \\
c_1 c_2 c_3 \;\text{—}\!\bullet\!\text{—}\; a_1 a_2 a_3 \\
T_3 \\
d_1 d_2 d_3
\end{array}
:=
\begin{array}{c}
b_1 \quad b_2 \quad b_3 \\
c_3 \text{—}\square\text{—}\square\text{—}\square\text{—} a_3 \\
c_2 \text{—}\square\text{—}\square\text{—}\square\text{—} a_2 \\
c_1 \text{—}\square\text{—}\square\text{—}\square\text{—} a_1 \\
d_1 \quad d_2 \quad d_3
\end{array}
, \tag{41}
$$

forming a new tensor-network path integral,

$$
\tag{42}
$$

This comes at the expense of the bond dimension of the new tensor network increasing exponentially in $\lambda$.

Two points in the new tensor-network path integral with a combinatorial distance $d$ have a distance $\lambda d$ in the original network. Thus, if the correlation length of the old tensor-network path integral was $\xi$, then the new correlation length is $\xi/\lambda$. The idea is that by choosing larger and larger $\lambda$, we will eventually arrive at a fixed-point model with $\xi = 0$. Often, renormalization is thought of as an iterative procedure where in each step we block by a factor of 2 (or another small number), corresponding to only considering the subsequence $T_{2^x}$ instead of all $T_\lambda$, which explains the name "fixed-point model" as a fixed point of this iteration.

There are, however, two major difficulties with this idea. First, to arrive at a fixed-point model, we would need the sequence $T_\lambda$ to converge. As such, this does not make any sense, as the different $T_\lambda$ have different bond dimensions corresponding to different vector spaces. To make the definition work, we use the same formulation as for the Trotterization limit in Section 2.2. That is, we choose a norm for the space of infinite-bond-dimension tensors which is compatible with tensor product and contraction, such that the normalizable tensors form a tensor type. Then, for every $\lambda$ we choose an invertible domain wall between each $T_\lambda$ and some $\widetilde{T}_\lambda$ which all live in the same infinite-bond-dimension normed space of tensors. A simple example for this would be applying on-site invertible matrix $S_\lambda$ to all horizontal and vertical bonds of the network,

$$
\begin{array}{c}
b \\
c \text{ - }\boxed{\bullet}\text{ - } a \\
\widetilde{T}_\lambda \\
d
\end{array}
=
\begin{array}{c}
c \\
S_b \\
b \text{ - }S_b^{-1}\!\!\;\bullet\;\boxed{\bullet}\;\bullet\!\!\;S_\lambda\text{ - } a \\
T_\lambda \\
S_b^{-1} \\
d
\end{array}
, \tag{43}
$$

but in reality slightly more complex invertible domain walls seem to be necessary (on-site transformations yield a representation of the renormalization as a *tree tensor networks*, whereas nearest-neighbour disentanglers as in *MERA* correspond to a more complex invertible domain wall). Now, the question is whether there exists such a sequence of invertible domain walls such the $\widetilde{T}_\lambda$ converges,

$$
\lim_{\lambda \to \infty} \widetilde{T}_\lambda = \widetilde{T} . \tag{44}
$$

Note that even though the sequence is defined in an infinite-dimensional normed vector space of tensors, it is possible that the limit $\widetilde{T}$ has only a finite number of non-zero entries, such that the resulting fixed-point model is again finite-dimensional. All in all, however, it is an open question if and under what circumstances we invertible domain walls, such that $\widetilde{T}_\lambda$ converges.

The second problem with fine-graining is that it is not necessarily true that the blocked model $T_\lambda$ is in the same (gapped-path) phase as the original model $T$. There are indeed examples where fine-graining yields a different phase, namely fixed-point models consisting of topological defect networks. The simplest example for this is the classical 2-dimensional anti-ferromagnetic Ising model at zero temperature, given by a square-lattice tensor network of bond dimension 2,

$$
\begin{matrix} & b \\ & | \\ c - \boxed{\phantom{x}} - a \\ & |\, T \\ & d \end{matrix}
= \begin{cases} 1 & \text{if} \quad a = b \neq c = d \\ 0 & \text{otherwise} \end{cases} .
\tag{45}
$$

After fine-graining with $\lambda = 2$, we get a bond dimension 4 tensor, which can be mapped back to bond dimension 2 via an on-site unitary. The result is the tensor describing the *ferromagnetic* Ising model at zero temperature,

$$
\begin{matrix} & b \\ & | \\ c - \boxed{\bullet} - a \\ & |\, T'_2 \\ & d \end{matrix}
= \begin{cases} 1 & \text{if} \quad a = b = c = d \\ 0 & \text{otherwise} \end{cases} .
\tag{46}
$$

The anti-ferromagnetic and ferromagnetic phases are different, and the two tensor-network path integrals cannot be transformed into each other by a gapped path or an invertible domain wall. Thus, the sequence $T'_\lambda$ cannot converge to a fixed-point model. Instead, $T'_\lambda$ alternates between a ferromagnetic phase for odd $\lambda$ and an anti-ferromagnetic phase for even $\lambda$. Note that this phenomenon does not occur if we explicitly allow changes of the unit cell, or equivalently, explicit *breaking of translational symmetry* in the definition of a phase. That is, models $X(s)$ of an interpolating continuous gapped path or the invertible domain walls can have a larger unit cell than the original models, or the tensors of $X(s)$ or a domain wall are allowed to be alternate with a periodicity larger than 1. Then the anti-ferromagnetic and ferromagnetic models are in the same phase, and their fixed-points are connected by an invertible domain wall. The breaking of the translational invariance is usually considered part of the definition of a phase in the context of topological order. In $3+1$ dimensions, *fracton phases* provide more interesting examples of networks of topological defects [24]. For those models, $T'_\lambda$ can consist of a tensor product of many copies of the same model, whose number scales with $\lambda$, which is known as *bifurcation* [25].

## 2.7 Topological extendibility and liquid models

Many gapped phases of matter obey a very fundamental property that their path integral can be defined on arbitrary topological manifolds and is invariant under arbitrary homeomorphisms. We say "many" since this is not precisely true for all phases, as we will discuss in more detail later. The topological invariance of the path integral implies different well-known appearances of topology in the study of such phases, such as a ground-state degeneracy that depends on the topology of the physical space, and their low-energy description in terms of *topological quantum field theory*. Accordingly, such phases of matter are often referred to as *topological phases*. Topological invariance will be the one and only property we use to construct and study fixed-point models in this work.

The fundamental property of topological invariance can be formalised using the language of tensor-network path integrals. First of all we note that tensor-network path integrals coming from condensed-matter models or similar are usually defined on a regular, translation-invariant lattice only. Being able to define the model on spacetime manifolds of arbitrary topology means that we need to be able to extend its definition from tensor networks living on square lattices to tensor networks living on irregular lattices, such as arbitrary triangulations. E.g., the square-lattice tensor network in Eq. (6) could be extended by allowing vertices of the background lattice where 3 or 5 instead of 3 plaquettes meet,

$$\rightarrow \qquad . \tag{47}$$

Or, we could still demand 4 faces meeting at a vertex, but in addition to 4-gon faces also allow triangle faces and 5-gon faces, represented by two further tensors,

$$\rightarrow \qquad . \tag{48}$$

Even if the original model had a robust gap, it cannot be guaranteed that also the extended model has a robust gap. This is because the extended model has more possible network annuli, i.e., networks of topology $S_{n-1} \times [0, 1]$, as now also irregular network annuli are allowed in the definition of a robust gap. We say that a tensor-network path integral with robust gap on regular, translation-invariant lattices is *topologically extendible* if there exists an extension to irregular lattices which also has a robust gap.

As the chosen name suggests topologically extendible tensor-network path integrals indeed obey topological invariance, which follows automatically from the robust gap. To see this in $n$ spacetime dimensions, consider two different networks representing an $n$-ball which look within a combinatorial-distance-$d$ from the boundary. In other words, we look at two different networks filling the interior of the same annulus network of width $d$. Since the annulus operator from the inside to the outside has a rank-1 apart from errors exponentially small in $d$, the evaluations of the two $n$-balls are (approximately) equal as well, up to a prefactor. E.g., for $n = 1 + 1$, we have schematically,

$$\left\| \quad - \alpha \quad \right\| \tag{49}$$

$$< C_\circ e^{-\frac{d}{\xi}} .$$

The annulus of width $d$ is in blue, and the two different interiors in red. $C_\circ$ is a pre-factor only depending on the boundary of the red fillings. $\alpha$ is a scalar pre-factor, which is necessary in the general case, but can be normalized to 1 for many phases (namely for phases without a chiral anomaly). Thus, the topological extendibility implies that we can arbitrarily change the irregular network in the middle of some patch of tensor-network path integral, which is the lattice analogue of homeomorphism invariance.

Now consider a fixed-point model, where the annulus operator is exactly a rank-1 operator, at least for sufficiently large $d$. If such a model can be topologically extended (in a way that

the extended model is still fixed-point), we get exact equations between tensor networks, such as

$$\tag{50}$$

corresponding to a re-cellulation

$$\tag{51}$$

We have found that the topologically extended fixed-point model is given by a set of tensors subject to a set of tensor-network equations. The tensor-network equations allow to arbitrarily deform the networks as long as we preserve the topology, i.e., they are a discrete analogue of homeomorphisms in the continuum. The approach in this work is to come from the other side: We start with a set of tensor variables and tensor-network equations which are a continuum version of topological manifolds, and use it as an ansatz for fixed-point path integrals of topological order. Focus of this work will be the construction of different such fixed-point ansatzes and the study of their equivalences on a purely combinatorial level.

To be able to talk on such a combinatorial level, we will introduce a small set of new vocabulary. We will refer to the combinatorial/diagrammatic structure of a finite set of tensor-network equations for a finite set of tensor variables as a *liquid*. The tensor-network diagrams are called *networks*, the diagram pairs of the tensor-network equations *moves*. The actual choice of tensors fulfilling the equation is referred to as *model* of the liquid. We will formalize different fixed-point ansatzes as different liquids, and their models are fixed-point models for topological phases. In order to prove the equality of different liquids on a diagrammatic level develop a tool called a (weakly invertible) *liquid mapping*, which for which we also find many other important applications in the study of fixed-point models.

All in all, liquid models provide a classification of phases of matter (in terms of tensor-network path integrals) under three assumptions, which are very plausible but hard to prove: First, the assumption that the gapped tensor-network path integral can be topologically extended, which is the core property of topological order. Second, the assumption that the phase possesses a fixed-point model which is justified by the concept of renormalization. Third, the assumption that invertible domain walls are equivalent to gapped paths for fixed-point models. What we obtain is an identification of phases with the solutions to a finite set of tensor-network equations for a finite set of tensor variables (the liquid model), modulo another set of variables and equations (the invertible domain wall). So in other words, we identified phases with instances of some (tensor-network-) linear algebraic structure, and in fact the latter are often similar to well-known algebraic structures such as weak Hopf algebras or fusion categories. Note that we do not classify the instances of algebraic structures themselves in the sense of efficiently enumerating them, and this is also not done by any existing classification apart from very restricted settings or in 1+1 dimensions. E.g., such a classification for weak Hopf algebras is far out of reach, and will most likely remain so for the forseeable future. Note that this is not a fundamental problem since it is only the simplest (low-bond-dimension) models which are physically relevant, and those can eventually be worked out. In this work, we will not be concerned with the construction of such new models, but we will be concerned with questions one level higher, namely the derivation and equivalence of different fixed-point ansatzes.

The tools and the perspective developed in this work gives us a straight-forward way to derive new fixed-point ansatzes, and understand and generalise existing ansatzes using only

one single property, namely topological invariance. In fact, it is possible to rigorously argue for some fixed-point ansatzes that they are universal in the sense that they can emulate all other fixed-point ansatzes. In Ref. [11], we pursue this line of thought and find new fixed-point ansatzes that are very different from any previously known ansatz, and have to potential to capture phases without gappable boundaries. Here, we will focus on applying our formalism to obtain ansatzes which are more similar to existing models, and deepen the understanding of the latter as well as generalising them.

Let us close this section by expanding on a comment from the beginning, namely that it is *not* true that any gapped (fixed-point) tensor-network path integral can be topologically extended. Tensor-network path integrals with a robust gap are analogous to gapped Hamiltonians with a further condition which referred to as *topological quantum order* [19]. However, in this sense the term does not correspond to topological invariance of the path integral, and so is a misnomer from this point of view. Surely, the name 'topological' may be justified from different perspectives, for example from the fact that phases are path-connected regions in the topological space of models, or from the use of vector bundles describing the band structure of non-interacting free-fermion Hamiltonians in momentum space.

All known examples of fixed-point models which do not have topological extendibility still have a very close relation to models that do. Namely, they are equivalent to a regular translation-invariant grid of topological defects. A topological defect refers to a submanifold (of some fixed dimension) where a tensor-network path integral is altered, subject to moves (tensor-network equations) which allow for arbitrary topology-preserving deformation of the embedded submanifold. An example is the classical zero-temperature Ising model partition function in 2 dimensions, and the topological defect consisting in spin flips along a 1-dimensional submanifold. Placing this defect along *every* horizontal and vertical line of a square lattice yields the anti-ferromagnetic Ising model which we looked at in Section 2.6. This tensor-network path integral has a robust gap if we do impose the spin-flip symmetry, but the rigidly embedded defects hinder it from being topologically extended. Another example is given by a toric code in $2 + 1$ dimension with the $1 + 1$-dimensional topological duality defect placed on every plane of a cubic lattice parallel to the $x$ and $t$ axes. The presence of such defects has been called *weak symmetry breaking* and present in the abelian topological phase of the honeycomb model [3]. Again, the presence of those rigidly placed defects hinders the model to be topologically extendable.

All examples of the above type have the property that they become topologically extendable when we enlarge the unit cell or equivalently fine-grain at a particular scale. As changing the unit cell is often considered valid and part of the definition of a phase, those examples might appear boring to many physicists. However, there are examples of models in higher dimensions which do not become topologically extendible no matter what size of unit cell we chose, namely so-called *fracton phases* which appear in $3 + 1$ or higher dimensions. In Ref. [24], many of those fracton models are shown to be equivalent to grids of topological defects (called *defect networks* there). In accordance with our claims it has been conjectured that such defect networks represent all gapped ("topological") phases. Note that our tensor-network path integral picture of grids of defects is a bit more general than the Hamiltonian picture since we also allow defects which are not parallel to the time direction.

# 3 Toy examples for liquids

In this section, we introduce many of the relevant concepts at the hand of three simple toy fixed-point ansatzes, i.e., three different liquids where we keep the amount of technical details at a minimum. We also discuss the relation of the liquid models to tensor-network path

integrals linked to physics, and to algebraic structures linked to mathematics.

## 3.1 Toy liquid in $1+1$ dimensions

As a first example we consider phases $1+1$ dimensions, that is, in one spatial dimension, which can be topologically extended. Despite the fact that non-trivial intrinsic topological order in the conventional sense only exists in $2+1$ dimensions, this example is well suited to illustrate important concepts on a diagrammatic level. Moreover, we would like to mention that, if we impose (on-site) symmetries on the models, we can obtain models for non-trivial *symmetry-breaking phases* as well as *symmetry protected topological (SPT) phases*. For pedagogical reasons, we neglect the following two important technical details in this section. In Section 4 we will see that we need to distinguish the indices of a tensor for a better representation of topological space-time, and in Section 6 we will see that we need to add an orientation and Hermiticity move in order to give the models a standard quantum mechanical interpretation.

### 3.1.1 Square lattice model and extended model

A tensor-network path integral in $1+1$ dimensions coming from a condensed-matter model, such as the Trotterization of a quantum spin chain Hamiltonian, looks like

$$\tag{52}$$

This is a (tensor) network formed by copies of a single 4-index tensor variable, with a single bond dimension variable.

We now assume that the model is a fixed-point model and that it can be extended to arbitrary triangulations of 2-manifolds. The easiest way to do this is to associate one copy of a 3-index tensor to every triangle of the triangulations of 2-manifolds, and contract indices between tensors at adjacent triangles,

$$\tag{53}$$

The 3-index tensor variable alone is an example of a *liquid*, which we will call the *triangle liquid*. So far, the notion of liquid is rather empty, but it will become non-trivial when we later add *moves*. The actual tensor determining the value of the 3-index variable is a *model* of the liquid. The square-lattice tensor-network path integral is also a model of a liquid, determined by a 4-index tensor variable, but one to which we will not add any moves. The extended model is related to the square-lattice model as follows. We can divide each square into two triangles, and use as square tensor the tensor obtained by contracting two triangle tensors according to

$$\tag{54}$$

The pure combinatorics of an equation like is a first example of a *liquid mapping* from the square-lattice liquid to the triangle liquid. This means that, conversely, it will map models of the triangle liquid to models of the square-lattice liquid. Liquid mappings are a central technical tool of this work. They will be used to formalize various operations in a unified way, just to name a few,

- the equivalence between ad hoc topological liquids and their more sophisticated simplified forms, e.g., in Section 4.4,

- the equivalence between different liquids describing the same type of topological order, e.g., seen in Section 8.3,

- the relation between topological liquids and known algebraic structures, as described, e.g., in Section 3.1.2,

- topological deformations, such as reshaping a boundary into a bulk as in Section 3.3.3, or compactifications or suspensions like the 2D embedding mapping in Section 8.2,

- the relation of liquids to commuting-projector Hamiltonians, as seen, e.g., in Section 3.1.3.

The key property of the extended model is its topological invariance, which we will formulate in terms of *moves*, i.e., tensor-network equations for the tensor variables of the liquid. Those moves need to form a combinatorial analogue of continuum homeomorphisms in terms of triangulations. This is given by the so-called *Pachner moves* [26] which act as

$$
\tag{55}
$$

It is known that any two (combinatorial) triangulations of the same manifold are related by Pachner moves. Conversely, it is easy to see that Pachner moves preserve the topology of a triangulation.

Now, tensor-network equations can be obtained by associating tensors to the patches on the left- and the right hand side of the Pachner moves. Topological invariance of the model means that the evaluations of the two corresponding tensor networks are equal,

$$
\tag{56}
$$

The pure combinatorics of such diagrams will be referred to as *moves*, which are part of the liquid. A model of a liquid must fulfil all the equations given by the moves.

### 3.1.2 Relation to algebraic structures

In this section, we relate the above moves to algebraic structures. An *algebra* is a linear map $\cdot : V \otimes V \to V$, where $V$ is a vector space. A finite-dimensional algebra is represented by its structure coefficients, which form a 3-index tensor,

$$
\tag{57}
$$

Here, we think of $\cdot$ as a linear map from the top two indices to the bottom index. So, an algebra is nothing but a model of the liquid depicted above. An algebra is *associative*, if

$$(a \cdot b) \cdot c = a \cdot (b \cdot c) . \tag{58}$$

This can be formulated as an equation between two tensor networks, namely

$$\tag{59}$$

So, associativity defines a move, and associative algebras are models of the liquid defined by that move. There are many other examples of algebraic structures which are models of liquids, such as Frobenius algebras, unital algebras, commutative algebras, Hopf algebras, representations of algebras, etc. Every model of the triangle liquid can be turned into an algebra by a liquid mapping from the algebra liquid to the topological liquid

$$\tag{60}$$

This is true because if we substitute the mapping into the associativity axiom of the algebra liquid, we obtain the 2-2 Pachner move of the triangle liquid,

$$\tag{61}$$

Thus, every model of the triangle liquid yields an algebra which is automatically associative. In general, a liquid model from a source to a target liquid associates a target network to every source tensor variable such that the *mapped moves* of the source liquid are moves of the target liquid (or can at least be derived from the latter as we will explain later).

We often observe that topological liquids have liquid mappings from well-known algebraic structures. However, often there is no inverse liquid mapping from the topological liquid to the algebraic liquid. This means that models of topological liquids define some algebraic structure, but the algebraic structure misses some additional axioms which are needed for a topological fixed point model. This is mostly due to the fact that the networks in the moves of algebraic structures always allow for a global "flow of time", as from the top to bottom in the algebra diagrams above. Liquids describing topological fixed-point models however do not have this flow of time since their networks represent Euclidean spacetime.

### 3.1.3 Commuting-projector Hamiltonians and stacking

We have already introduced the notion of a liquid mapping to formalise the relation between the square-lattice model and the triangle-liquid model, and between the algebra-liquid and the triangle liquid. Here, we will give two more examples for operations which can be neatly formalised by a liquid mapping, namely the construction of a commuting-projector Hamiltonian, and the operation of embedding two non-interacting copies of a model into the same space-time. Both will lead to generalisations of the notion of liquid mapping we introduced so far.

If we apply the Trotterization procedure in Section 2.1 to any Hamiltonian, we never directly obtain a fixed-point model. This is because the first excited state always has a finite energy, corresponding to a finite "correlation length in time direction". However, as mentioned in Section 2.3, we do not need Trotterization if we have a commuting-projector model.

In this case, we can directly take the limit $\beta \to \infty$ for the individual Hamiltonian terms, obtaining a *local ground state projector*. For a spin chain with nearest-neighbour Hamiltonian, this projector is a 4-index tensor,

$$\begin{array}{c}a \qquad b \\ \square \\ c \qquad d\end{array} \tag{62}$$

acting as an operator from the bottom to the top two indices. The imaginary time evolution is represented by a product of these operators $P$ at different places, which yields a network of the form

$$ \tag{63} $$

In order to relate this tensor-network path integral (given by a 4-index tensor) with a triangle-liquid model (given by a 3-index tensor), we have to choose an according liquid mapping which transforms the network above into the network representing a triangulation of the plane. One such mapping is, e.g., given by

$$\begin{array}{c}a \qquad b \\ \square \\ c \qquad d\end{array} := \begin{array}{c}a \qquad b \\ \circ\!-\!\circ \\ c \qquad d\end{array} . \tag{64}$$

The fact that the local ground state projector forms a commuting-projector model corresponds to two tensor-network equations. The projector property is

$$\begin{array}{c}a \qquad b \\ \square \\ \square \\ c \qquad d\end{array} = \begin{array}{c}a \qquad b \\ \square \\ c \qquad d\end{array} , \tag{65}$$

and the commutativity is

$$\begin{array}{cc}a \quad b \quad e \quad a \\ e \, \square \qquad \square \, b \\ \square \, f \quad = \quad c \, \square \\ c \quad d \qquad d \quad f\end{array} . \tag{66}$$

So commuting-projector models themselves are models of the above *commuting-projector liquid* with two moves. If we plug the mapping Eq. (64) into the move Eq. (65), we get

$$\begin{array}{c}a \quad b \\ \circ\!-\!\circ \\ | \quad | \\ \circ\!-\!\circ \\ c \quad d\end{array} = \begin{array}{c}a \quad b \\ \circ\!-\!\circ \\ c \quad d\end{array} . \tag{67}$$

This is not a move of the triangle liquid. However, it is equivalent to a sequence of moves,

$$\begin{array}{c}a \quad b \quad a \\ \circ\!-\!\circ \quad \circ \\ | \quad | \quad = \quad \triangleright\!\circ\!-\!\circ \\ \circ\!-\!\circ \qquad b \\ c \quad d \quad c \quad d\end{array} = \begin{array}{c}a \quad b \\ \circ\!-\!\circ \\ c \quad d\end{array} . \tag{68}$$

In the first step we applied the 2-2 Pachner move, and in the second step the 1-3 Pachner move. We will refer to moves equivalent to sequences of moves of a liquid as *derived moves*, and the corresponding sequences as *derivations*. We will generalise our notion of a liquid mapping by allowing the mapped moves to be derived moves of the target liquid.

As another example of a liquid mapping, consider "stacking two copies" of a model, i.e., embedding both models in the same space(-time) in a non-interacting way. If a model has topological invariance, then so will the stacked model. Stacking is nothing but a mapping from a liquid (here the $1+1$ dimensions topological liquid) to itself

$$
\underset{cc'}{\overset{aa' \quad bb'}{\bigcirc}} := \underset{c}{\overset{a \quad b}{\bigcirc}} \underset{c'}{\overset{a' \quad b'}{\bigcirc}} . \tag{69}
$$

This provides a slight generalisation of the concept of a liquid mapping introduced so far, as every open index on the left hand side corresponds to two open indices on the right hand side. In the notation, this was indicated by using as labels on the left the concatenation of the corresponding labels on the right. If we apply this mapping to any network, we will obtain two copies of that network. E.g., the mapped 2-2 Pachner move gets mapped to a move relating two disconnected networks consisting of two tensors each on each if its sides. Obviously, this mapped move can be derived by applying the 2-2 Pachner move separately to each copy.

### 3.1.4 Models

As we have seen in Section 3.1.2, models of the triangle liquid correspond to associative algebras, for which a few additional axioms hold[1]. Such algebras fall into discrete families, up to basis changes. One family of algebras which also yield topological models is given by the algebra of complex functions over an $x$-element set under point-wise multiplication, for arbitrary $x$. This corresponds to the choice

$$
\underset{c}{\overset{a \quad b}{\bigcirc}} = \underset{c}{\overset{a \quad b}{\bullet}} = \begin{cases} 1 & \text{if } a = b = c \\ 0 & \text{otherwise} \end{cases}, \tag{70}
$$

for $0 \le a, b, c < x$, which will also be referred to as the *delta tensor*, and denoted by a small dot. Delta tensors can be defined for an arbitrary number of indices, with entry 1 if all the index values are equal and 0 otherwise.

If we evaluate such a model on a network representing a triangulation of a sphere, we get the number $x$. So we see that every family yields a different topological invariant and thus all families correspond to different phases. If we equip the tensor with the regular representation of $\mathbb{Z}_x$ acting on every index independently (or any other permutation representation of a group without non-trivial closed subsets) those models are fixed-point models for *symmetry-breaking phases*. E.g., for $x = 2$, the liquid is equivalent to the ordered-phase fixed point of the $1+1$-dimensional *Ising model*. This is a chain of qubits with a nearest-neighbour Hamiltonian

$$
H = \sum_i h_i = -\sum_i Z_i Z_{i+1} . \tag{71}
$$

This is a commuting-projector Hamiltonian given by the local ground state projector

$$
\lim_{\beta \to \infty} e^{-\beta h} \sim \frac{1}{2}(\mathbb{1} - Z_0 Z_1) . \tag{72}
$$

---

[1]We are not yet fully precise here, but in the end the models will be equivalent to something like *commutative special Frobenius algebras*.

So using the notation from Section 3.1.3, we get

$$
\begin{array}{c} a \quad \quad b \\ \square \\ c \quad \quad d \end{array} = \begin{cases} 1 & \text{if} \quad a = b = c = d \\ 0 & \text{otherwise} \end{cases} . \tag{73}
$$

Indeed, this is exactly what we get if we plug the model in Eq. (70) into the commuting-projector mapping in Eq. (73).

### 3.1.5 Phases and invertible domain walls

In this section, we will illustrate how the definition of a phase in terms of invertible domain walls from Section 2.5 applies to models of topological liquids. The algebra of functions over a 2-element set yields a model as we have seen in Section 3.1.4. Another model of the liquid is given by the $\mathbb{Z}_2$ group algebra

$$
\begin{array}{c} a \quad \quad b \\ \mathbb{Z}_2 \circ \\ c \end{array} = \begin{cases} \frac{1}{\sqrt{2}} & \text{if } a + b + c = 0 \mod 2 \\ 0 & \text{otherwise} \end{cases} . \tag{74}
$$

The two algebras are isomorphic via a basis change known as the *Hadamard transformation*

$$
a \overset{\square}{\underset{\text{H}}{\quad}} b = \frac{1}{\sqrt{2}} \begin{pmatrix} 1 & 1 \\ 1 & -1 \end{pmatrix} . \tag{75}
$$

Basis changes are a very specific example of invertible domain walls, and thus the two liquid models are in the same phase. Concretely, we have

$$
\begin{array}{c} a \quad \quad \quad b \\ \text{H} \diamond \mathbb{Z}_2 \circ \diamond \text{H} \\ \square \text{H} \\ c \end{array} = \begin{array}{c} a \quad \quad b \\ \delta \circ \\ c \end{array} . \tag{76}
$$

$H$ happens to be its own inverse

$$
a \overset{\square}{\underset{\text{H}}{\quad}} \overset{\square}{\underset{\text{H}}{\quad}} b = a \quad\text{——}\quad b . \tag{77}
$$

Now, start with a network of the $\delta$ liquid model, e.g.,

$$
\delta \diamond\text{——}\diamond \delta . \tag{78}
$$

Then, applying Eq. (76) from right to left to a network of the $\delta$ liquid model, we obtain a network like

$$
\begin{array}{c} \diamond\text{H} \quad \text{H}\diamond \\ \mathbb{Z}_2\circ\square\square\circ\mathbb{Z}_2 \\ \text{H H} \\ \diamond\text{H} \quad \text{H}\diamond \end{array} . \tag{79}
$$

Finally, applying Eq. (77) from left to right to all pairs of adjacent Hadamard tensors, we obtain

$$
\begin{array}{c} \diamond\text{H} \quad \text{H}\diamond \\ \mathbb{Z}_2\circ\text{——}\circ\mathbb{Z}_2 \\ \diamond\text{H} \quad \text{H}\diamond \end{array} . \tag{80}
$$

As we have seen, using Eqs. (76), (77), we can transform any $\delta$-tensor network into the according $\mathbb{Z}_2$-tensor network. If the network has open indices, we will end up with some residual

Hadamard transformations near this open boundary. We found that the $\mathbb{Z}_2$-model and the $\delta$-model are related by an invertible domain wall, and thus in the same phase.

As another example, consider the model

$$
\underset{x}{\overset{a \quad b}{\bigcirc}}_{c} = 2^{(-3/2)} \quad (\forall a, b, c) . \tag{81}
$$

We will show that this model is in the same phase as the trivial model where each tensor is the number 1. We notice that the tensor above is the tensor product of three times the same vector

$$
\underset{x}{\overset{a \quad b}{\bigcirc}}_{c} = \quad , \tag{82}
$$

where

$$
\bullet\!\!-a = \frac{1}{\sqrt{2}} \quad (\forall a) . \tag{83}
$$

Furthermore, this vector is normalized, i.e. (note that the empty network evaluates to the scalar 1),

$$
\bullet\!\!-\!\!\bullet = \qquad . \tag{84}
$$

The above two moves again define an invertible domain wall. We start with a tensor network of the $x$-model

$$
x \,\bigcirc\!\!-\!\!\bigcirc\, x \quad , \tag{85}
$$

and then apply Eq. (82) from left to right to every $x$ tensor,

$$
\tag{86}
$$

Finally, we apply Eq. (84) to every pair of vectors, which yields the empty network everywhere except for at a potential boundary

$$
\tag{87}
$$

So we see that this model is in the same phase as the trivial model, again via a very simple invertible domain wall.

## 3.2 Non-triangular toy liquid in $2+1$ dimensions

In the next example, we explore a simple topological fixed-point ansatz, i.e., a topological liquid, in $2+1$ dimensions. Again, a condensed-matter model yields a tensor-network path integral on a regular (e.g., cubic) lattice by Trotterization of a Hamiltonian imaginary-time evolution or by other means and again, we assume that for each phase there exists a fixed-point model which can be topologically extended to arbitrary triangulations of 3-dimensional manifolds. The most straight-forward construction analogous to the triangle liquid would be to associate one tensor to every simplex and to formulate the topological invariance analogous to the $1+1$-dimensional case as invariance under 3-dimensional Pachner moves. In this section we will instead sketch a less standard formulation of topological invariance that is based on a different way to combinatorially discretise a manifold and is referred to as *face-edge liquid*. Here, we again focus on the concept and omit technical detail, while in Section 8.2 all technical details are added and we discover that this liquid can be identified with quantum double models. The equivalence between the standard simplex based liquid and the non-standard liquid presented here is then shown in Section 8.3.

### 3.2.1 The face-edge liquid

For the *face-edge liquid* we allow arbitrary cellulations of a 3-manifold, but we demand that every face is either a triangle or a 2-gon and that every edge is 3-valent or 2-valent (i.e., it is adjacent to three or two faces). A moment of thought reveals that every cell complex can be brought into this form, e.g., a 4-gon can be split into two triangles with a 2-valent edge in between as

$$\qquad \rightarrow \qquad . \tag{88}$$

Dually, a 4-valent edge can be split into two 3-valent edges with a 2-gon face in between,

$$\qquad \rightarrow \qquad . \tag{89}$$

There is one 3-index tensor variable associated to every triangle and different 3-index tensor variable to every 3-valent edge. This means that a model of this liquid is given by *two* potentially different 3-index tensors. At every pair of adjacent triangle and 3-valent edge, there is a bond between the two corresponding tensor indices. Since the face and edge tensors are different, we use two different shapes to represent them,

$$\qquad , \qquad . \tag{90}$$

The 2-gons and 2-valent edges are not explicitly represented by tensors. Instead, the two edges adjacent to a 2-gon, and likewise the two faces adjacent to a 2-valent edge are directly connected by a bond. E.g., two 3-valent edges separated by a 2-gon are represented as

$$\qquad . \tag{91}$$

As in $1+1$ dimensions, two combinatorial triangulations correspond to the same manifold exactly if they are related by 3-dimensional Pachner moves. For the particular combinatorial network structure chosen here there is an equivalent set of moves, which can be divided into 3 groups. First, there are moves involving only triangles separated by 2-valent edges, which equal the 2-dimensional Pachner moves for the face tensors only, namely

$$\qquad = \qquad \tag{92}$$

and the same for the 1-3 Pachner move. Then, moves involving only 3-valent edges separated by 2-gon faces. In terms of cell complexes, those moves are Poincaré dual to the moves above. In network notation they look the same apart from that we have to use filled circles instead of empty circles. Finally, the most important move involves both face and edge tensors. It merges two triangles with two shared 3-valent edges into a single triangle with one adjacent 3-valent edge, i.e.,

$$\qquad = \qquad . \tag{93}$$

### 3.2.2 Bi-algebras

As in the $1 + 1$-dimensional case, the present liquid has a great similarity to a well known algebraic structure. To see this, we first note that there is an obvious liquid mapping from the triangle liquid in $1 + 1$-dimensions to the present liquid, in which the triangle (as part of the 2-dimensional cell complex) is mapped to the triangle (as part of the 3-dimensional cell complex). Dually to that, there is a liquid mapping in which the triangle is mapped to the 3-valent edge. Thus, by the means of these two mappings every model of the present liquid gives rise to two associative algebras.

Two associative algebras (more precisely, an algebra and a *co-algebra*) are called a *bi-algebra*, if they fulfil certain additional axioms. The main axiom (which states that the co-algebra is an algebra homomorphism) is precisely the move in Eq. (93). Thus, the present liquid is basically the bi-algebra liquid, together with a few additional moves which make it "more topological". This observation could be formalized as a liquid mapping from bi-algebras to the present liquid.

### 3.2.3 Models

Specifying the tensor type to array tensors, we look for models of the liquid, i.e., solutions to the move equations. The similarity to bi-algebras greatly helps assessing the situation: First, we know that bi-algebras fall into a discrete set of families related by basis changes, and so do the models of the present liquid. Second, there are many known examples for bi-algebras, many of which also yield models of the present liquid. Thus, in practice, we can look at the simplest examples of bi-algebras, see whether they can be turned into models of the present liquid, and check whether some of the models are in the same phase. Moreover, we will see that it is "rather unusual" for different models to be in the same phase, and that one can usually show their distinctness by evaluating closed networks.

As a particular example we recall that every group defines a bi-algebra, which can be turned into a model of the present liquid. Those models are equivalent to the *Kitaev quantum double models* [27], which are models for *intrinsic topological order* in $2 + 1$ dimensions. E.g., if we pick the group $\mathbb{Z}_2$, the index configurations are the group elements $\{0, 1\}$, and the edge and face tensors are

$$
a \underset{c}{\overset{b}{\bigcirc}} = \begin{cases} 1 & \text{if } a + b + c = 0 \mod 2 \\ 0 & \text{otherwise} \end{cases} ,
$$

$$
a \underset{c}{\overset{b}{\bullet}} = \begin{cases} 1 & \text{if } a = b = c \\ 0 & \text{otherwise} \end{cases} .
$$

(94)

It is easy to see that each tensor satisfies the 2-2 and 1-3 Pachner move, and both tensors together fulfil the move in Eq. (93). Actually, for the sake of simplicity, we are ignoring a global factor of $1/2$ missing in the 1-3 Pachner move for face tensors. This will be fixed in Section 8.2.

This model corresponds to a commuting-projector Hamiltonian model known as the *toric code* [27] defined for qubits on the edges of a square lattice and Hamiltonian given by

$$
H = \sum_i A_i + \sum_j B_j .
$$

(95)

Here, $i$ runs over all plaquettes of the lattice and each $A_i$ is defined as (suppressing the site index)

$$
A = -Z_0 Z_1 Z_2 Z_3 ,
$$

(96)

where $0, 1, 2, 3$ label the 4 edges adjacent to the corresponding plaquette. Dually, $j$ runs over all vertices and

$$B = -X_0 X_1 X_2 X_3 \,, \tag{97}$$

where $0, 1, 2, 3$ label the 4 edges adjacent to the corresponding vertex. The commuting projectors themselves can be written as 8-index tensors

$$P_A = \frac{1}{2}(\mathbb{1} - Z_0 Z_1 Z_2 Z_3) \quad \rightarrow \quad \boxed{P_A} \,,$$

$$P_B = \frac{1}{2}(\mathbb{1} - X_0 X_1 X_2 X_3) \quad \rightarrow \quad \boxed{P_B} \,. \tag{98}$$

They are commuting because adjacent plaquettes and vertices always share two adjacent edges, and $Z_0 Z_1$ commutes with $X_0 X_1$. A tensor network representing the imaginary time evolution of the model is given by stacking layers of those commuting projectors.

To compare our topological model with the given commuting-projector model, we need a liquid mapping from the commuting-projector liquid to the topological liquid. A little bit of geometric imagination shows that the replacement

$$\boxed{P_A} := \quad , $$

$$\boxed{P_B} := \quad \tag{99}$$

will turn a stack of commuting projectors into a cellulation of the same space-time. Indeed, we find that via this mapping, the topological model Eq. (94) is mapped to the commuting-projector model Eq. (98).

## 3.3 Toy liquid with boundary in $1+1$ dimensions

### 3.3.1 Regular-lattice and extended model

As a third example, let us look at models in $1+1$ dimensions with physical boundary. From condensed-matter physics, we get a tensor-network path integrals

$$\tag{100}$$

on a regular square lattice with boundary. Formally, this is a liquid model with two tensor variables represented by two different shapes. Further, the bond dimension between the boundary

tensors is allowed to be different from the one of the bulk tensors, so the liquid has two different *bond dimension variables* as well, which are represented by a thicker line style for the boundary bonds.

We then assume that we have a fixed-point model which can be extended to arbitrary triangulations of arbitrary manifolds with boundary,

$$\tag{101}$$

The mapping from the square-lattice liquid to the topological liquid is obvious.

In the bulk, topological invariance is still ensured by Pachner moves, which only involve the triangle tensor, making the triangle liquid from Section 3.1 a sub-liquid of the current liquid. However, for topological invariance at the boundary, we need add the following additional move,

$$\tag{102}$$

The geometric interpretation of this move is to attach/remove a triangle to/from the boundary, which allows us to arbitrarily deform the boundary without changing the topology.

### 3.3.2 Representations and models

As the liquids above, our boundary liquid is again very similar to a very well-known algebraic structure, namely *representations*. A *representation* of an algebra $A$ is a linear map

$$R : V \otimes A \rightarrow V , \tag{103}$$

satisfying

$$R(R(x,a),b) = R(x,a \cdot b) . \tag{104}$$

This equation can written in tensor-network notation, and looks exactly like Eq. (102).

In order to find models of the present liquid, we start with models of the $1+1$-dimensional liquid in Section 3.1, and extend them by a choice of boundary tensor. Let's start with the model given in Eq. (70) related to the algebra of functions over some finite set $B$. For every $x \in B$, there is the corresponding *irreducible representation*, which defines a choice of boundary tensor,

$$\underset{\square}{\overset{a}{\vdash}} = \begin{cases} 1 & \text{if} \quad a = x \\ 0 & \text{otherwise} \end{cases} . \tag{105}$$

The boundary indices without labels are trivial, that is, they have bond dimension 1. For a 2-element set $B$, this corresponds to a non-symmetric boundary condition of the Ising model, where any spin near the boundary is fixed to the value $x$.

### 3.3.3 Bulk-to-boundary mapping

A 2-manifold with boundary might also be interpreted as a manifold with one puncture for every boundary circle. Imagine filling each such a puncture with a disk. On the combinatorial level, we can do this by adding one additional vertex corresponding to the centre of the disk, and one additional triangle for every boundary edge, spanned by this boundary edge and

the central vertex. Consider the boundary-less network for the filled, and the network-with-boundary for the non-filled triangulation. They can be mapped onto each other by reinterpreting the triangle tensors for the additional triangles as the boundary edge tensor for the boundary edges. Such a reinterpretation can be formalised as the liquid mapping

$$\hspace{1cm} (106)$$

If we apply this mapping to the move Eq. (102) of the boundary liquid, it turns directly into the 2-2 move in Eq. (56) of the bulk liquid. This example shows two new features of liquid mappings. 1) For a mapping from a liquid with multiple tensor variables, we have to give one network for each tensor variable. 2) If the liquid has different bond dimension variables, then each bond dimension variable of the source liquid is mapped to a bond dimension variable of the target liquid. In the present example, both the bulk and boundary bond dimension are mapped to the bulk bond dimension,

$$\hspace{1cm} (107)$$

In general, each bond dimension variable of the source liquid can be associated with a collection of bond dimension variable of the target liquid which can contain the same bond dimension variable multiple times, as we have seen in Eq. (69).

## 4 Topology and non-commutativity

In this section we will revisit the example of topological order in $1+1$ dimensions from Section 3.1 and discuss an important issue that we have not addressed so far. If we want the liquid to represent topological manifolds, we need to add more structure to the network. In particular we will motivate that it is necessary to distinguish the different indices of a tensor variable and show how this can be implemented concretely. The additional structure makes the liquid more complicated than the liquid from the previous example. To handle this complexity in the most efficient way, we seek a way to simplify the liquid without losing its ability to describe topological phases. In doing so, we invoke the concept of liquid mappings and introduce the notion of equivalent tensor liquids. We present what we believe to be the simplest representative of a topological liquid in $1+1$ dimensions and classify its phases.

### 4.1 Distinguishing indices

In Section 3.1, we have represented the triangulation of a manifold by a network which is really just a trivalent graph, with one node at every triangle,

$$\hspace{1cm} (108)$$

We would like the combinatorics of networks and moves to reflect continuum manifolds and homeomorphisms in a faithful way. However, the network combinatorics introduced so far does not uniquely encode the full combinatorial information of the triangulation. Imagine rebuilding the triangulation from the network's graph by replacing each vertex with a triangle and gluing the triangles associated to connected vertices along common edges. We encounter two problems: 1) The combinatorial structure of the graph does not distinguish between the three adjacent bonds, so we cannot tell which edges of the triangles we have to glue together. 2) Two edges can be glued in two opposite ways. E.g., consider the following graph that corresponds to two triangles with all edges glued together pairwise,

$$\hspace{1cm} (109)$$

This graph does not determine the topology of the resulting manifold. If we glue one of the three edge pairs, we obtain a 4-gon. Depending on how we glue the remaining edges of the 4-gon, we can obtain a sphere, a real projective plane, a torus, or a Klein bottle.

The second problem can be solved by giving each edge an orientation and demanding that those orientations match when we glue two edges of two triangles. The first problem is solved by realising that any manifold can be triangulated using only triangles with non-cyclic edge orientations [2]. This is also known as a triangulation with a *branching structure*. For a fixed triangle, the non-cyclic edge orientations induce an ordering of its vertices,

$$\tag{110}$$

This allows us to distinguish the three edges and refer to them by their source and target vertex. In our network notation, we allow rotating/reflecting the shapes of individual tensor copies, which makes it impossible to distinguish the three indices if the shape is a small circle. The shape for the tensor variable representing a branching-structure triangle should have less symmetry, which we implement by next-to-shape markers,

$$\tag{111}$$

The clockwise or counter-clockwise flags allow to uniquely identify the 3 indices of the tensor with the edges 01, 02 or 12 of the branching structure triangle, as indicated by the red labels. Note that here and in the subsequent, such red labels are not part of the formal graphical notation, but serve as an aid to identify the network notation with its geometric interpretation in terms of cell complexes. Networks using the new shape representing a branching-structure triangle uniquely specify the triangulation and thus the topology. E.g., the network

$$\tag{112}$$

represents a sphere unambiguously.

In our new (and final) notion of networks, the indices of a tensor are always distinct. However, we can still interpret the old notion, where (some of) the indices have not been distinguished. Indistinguishability of indices means that we are allowed to permute them, which is nothing but a move. Now, whenever we choose a shape for a tensor which has rotation/reflection symmetries due to which we cannot distinguish some of the indices, we implicitly assume that all the corresponding index permutation moves (or better, a set of moves generating all the permutation moves) are part of the liquid. Explicitly, we will denote such permutation moves using cycle notation, e.g.,

$$\overset{a}{\underset{c}{\searrow}}\overset{b}{\swarrow} \overset{\text{sym}}{=} (ab)\,, \qquad \overset{a}{\underset{c}{\searrow}}\overset{b}{\swarrow} \overset{\text{sym}}{=} (bc)\,. \tag{113}$$

If we instead use a shape without any symmetries, such as the one in Eq. (111), the index permutations can be denoted as ordinary moves

$$\tag{114}$$

---

[2]This can be seen by refining a non-oriented triangulation via a construction known as *barycentric subdivision,* which can be equipped with a canonical non-cyclic edge orientation.

If we interpret those moves in terms of triangulations, they correspond to cutting out a triangle and gluing it in a different way. Such an operation generally changes the topology of the triangulation. So the liquid we introduced in Section 3.1 has moves which are sufficient to have topological invariance, but also additional moves which go beyond topological invariance. We therefore expect that models of this liquid are too restricted and do not contain the most general fixed point models for topological order.

## 4.2 Non-simplified liquid

The branching structure/flags also need to be incorporated into the moves of the liquid. There are different ways a branching structure can be added to the Pachner moves. For the 2-2 Pachner moves (keeping in mind moves are not actually different if they are just rotated/reflected or we exchanged the left and right side), we count 3 different versions. One of them is

$$\tag{115}$$

Another one can obtained by, e.g., inverting the orientation of the $2-3$ edge. Note that if we glue the two patches above at their boundary, we obtain the surface of a branching-structure tetrahedron. In general, every Pachner move corresponds to a decomposition of that tetrahedron into two parts, and the 3 versions of the 2-2 Pachner move correspond to the 3 different decompositions of the tetrahedron into two faces on each side.

In the new network notation, the move becomes

$$\tag{116}$$

The red labels identify the tensors in the network with the triangles in the geometric interpretation. E.g., 023 refers to the triangle in Eq. (115) whose 0-vertex is the vertex 0, 1-vertex is 2, and 2-vertex is 3. Note again that the red labels are only hints for the reader and not part of the actual notation. Also, the open index labels were chosen in accordance with the names of the corresponding edges.

Analogously, there are now 4 different versions of the 1-3 Pachner move, corresponding to the 4 decompositions of the branching structure tetrahedron into two patches with 1 and 3 triangles each. One of them is

$$\tag{117}$$

As some sort of convention, we might also want to introduce the following *triangle cancellation move*

$$\tag{118}$$

implying that a non-cyclic 2-gon can be shrunk to a single edge, which is represented by a free bond in network notation as

$$\tag{119}$$

If we glue one edge of the left hand side of Eq. (118) to one boundary edge of any patch of a triangulation (including itself), this can be undone with Pachner moves. So the Pachner moves imply that the corresponding tensor is a projector, and contracting any index of any other tensor of the model with this projector yields the same tensor again. However, they do clearly not imply the triangle-cancellation move, and formally, the liquids with and without that move are inequivalent (in a sense that we will make precise soon).

However, when considering ordinary models of the liquid (with real or complex tensors), Eq. (119) can be viewed as a convention that does not hurt to impose. The projector in Eq. (119) has a $n$-dimensional support, and there exists an isometry which identifies this $n$-dimensional support with an $n$-dimensional vector space. Applying this isometry to every index of every tensor yields a model which is equivalent, as the tensors are invariant under applying the corresponding projector. In doing so, the tensor corresponding to the projector itself becomes the identity matrix.

In total, we end up with a liquid with 8 moves that we refer to as the "non-simplified liquid". As the moves correspond to equations between tensor networks that we need to solve in order to find models, it is important that the moves of a liquid are as simple as possible. In the following, we will find a "simplified liquid" which is equivalent to the non-simplified liquid, but has less and simpler moves.

## 4.3 Simplified liquid

The simplified liquid has one additional tensor variable whose geometric interpretation is a 2-gon cell with cyclic edge orientations,

$$
\begin{array}{c}
\text{1} \\ \text{0}
\end{array} \quad \rightarrow \quad 01 \longrightarrow\!\!\bigcirc\!\!\longrightarrow 10 \; . \tag{120}
$$

The new tensor will be denoted by a circle as well, however, it can be distinguished from the triangle tensor due to the different number of indices. Of course, a 2-gon cannot be embedded non-degenerately into Euclidean space without bending its edges. But this is no cause of a problem as we are talking about combinatorial/topological cell complexes and not geometric ones. The 2-gon is rotation symmetric which corresponds to a move

$$
a \longrightarrow\!\!\bigcirc\!\!\longrightarrow b \stackrel{\text{sym}}{=} (ab) \tag{121}
$$

justifying the choice of shape. This move can be derived from the moves below, however.

The moves of the simplified liquid only contain one single Pachner move, namely the 2-2 Pachner move in Eq. (116). All other 2-2 Pachner moves can be derived via additional moves of the simplified liquid related to symmetries of the triangle. In contrast to the liquid in Section 3.1, rotating or reflecting the triangle would change the branching structure. However, the changes of the edge orientations can be undone by gluing the cyclic 2-gon to the involved edges. This yields, e.g., the *(12)-triangle symmetry move*

$$
\begin{array}{c}
\text{1} \\ \text{0} \quad \quad \text{2}
\end{array} \quad \longleftrightarrow \quad
\begin{array}{c}
\text{1} \\ \text{0} \quad \quad \text{2}
\end{array} \; , \tag{122}
$$

where the nomenclature refers to effectively interchanging the role of the vertices 1 and 2. In network notation, this is

$$
\begin{array}{c} 01 \\ \bigcirc\!\!\longrightarrow\!\!\bigcirc\!\!\longrightarrow 21 \\ 012 \quad 12 \\ 02 \end{array}
\;=\;
\begin{array}{c} 01 \\ \bigcirc\!\!\longrightarrow 21 \\ 021 \\ 02 \end{array} \; . \tag{123}
$$

In order to generate the full symmetry group $S_3$ of the triangle, one only needs one further move, the *(01) triangle symmetry move*

$$\tag{124}$$

Again, in network notation, this amounts to

$$\tag{125}$$

The 1-3 Pachner moves can be derived from the 2-2 Pachner moves via the triangle cancellation move in Eq. (119), which is also part of the simplified liquid. Analogously, there is the *2-gon cancellation move*

$$\tag{126}$$

In network notation, we have

$$a \multimap\!\!\!\bigcirc\!\!\!\multimap b = a \!-\!\!\!- b \ . \tag{127}$$

## 4.4 Equivalence of the simplified and non-simplified liquid

In this section, we will motivate why the simplified and non-simplified liquids are "equivalent". For this, we should be able to rewrite networks of the simplified liquid as networks of the non-simplified liquid, and vice versa. This can be formalized by two liquid mappings $\mathcal{M}_1$ and $\mathcal{M}_2$, going from the non-simplified liquid to the simplified liquid, and back.

Note that the tensor variables of the non-simplified liquid are identified with a subset of the tensor variables of the simplified liquid. So there is a "trivial" candidate for the mapping $\mathcal{M}_1$, mapping the triangle of the non-simplified liquid to the triangle of the simplified liquid. In order to show that this defines indeed a liquid mapping, we need to show that the mapped non-simplified moves are derived from the simplified moves. As the mapping is "trivial", the mapped non-simplified moves just look like the non-simplified moves.

- One of the branching-structure 1-3 Pachner moves is derived from the 2-2 Pachner move in Eq. (116) and the triangle cancellation move in Eq. (119):

$$\tag{128}$$

- All other versions of branching structure 2-2 Pachner moves are derived from the 2-2 Pachner move in Eq. (116), together with the two triangle symmetry moves. E.g., the following 2-2 Pachner move

$$\tag{129}$$

is derived by

$$
(130)
$$

The bar over the referenced equation denotes that the move is applied from right to left.

- Similarly, all other 1-3 Pachner moves are derived from the move above in Eq. (128), together with the 2-gon cancellation move and the triangle symmetry moves.

The mapping $\mathcal{M}_2$ is only slightly more complicated.

- The triangle part of both liquids and accordingly mapped onto the itself (as part of the non-simplified liquid).

- A 2-gon cell can be triangulated using two triangles

$$
(131)
$$

Accordingly, the mapping for the 2-gon is given by

$$
a \to \bigcirc \to b := a \to \bigcirc\bigcirc \to b \ . \tag{132}
$$

Again, we have to find derivations for the mapped simplified moves from the non-simplified moves. E.g., if we plug the mapping Eq. (132) into the 2-gon cancellation move Eq. (127), we obtain

$$
a \to \bigcirc\bigcirc\to\bigcirc\bigcirc\to b = a \text{---} b \ . \tag{133}
$$

This can be derived by 1) a 2-2 Pachner move, 2) a 1-3 Pachner move, and 3) the triangle cancellation move. We will not explicitly give derivations for each mapped move here. Instead, we would like to remark that the mapped moves (except for the 2-gon and triangle cancellation moves) correspond to re-triangulations of a disk. It is known that any two triangulations of the same (piece-wise linear) manifold are related by a sequence of Pachner moves [26]. So, if we rely on this statement about the geometric interpretation, we know that derivations for all mapped moves must exist.

So, we have found two liquid mappings going from the non-simplified liquid to the simplified liquid and back. However, this alone does not really mean anything, e.g., between any two liquids there's the *trivial mapping* which maps every bond dimension variable to the empty collection of bond dimension variables, and every tensor variable to the empty network. What we additionally need is that that if we go from the non-simplified liquid to the simplified liquid and back, we end up with the same network. In other words, $\mathcal{M}_2 \circ \mathcal{M}_1$ should be the identity, and the same should hold for $\mathcal{M}_1 \circ \mathcal{M}_2$.

We find indeed that $\mathcal{M}_2 \circ \mathcal{M}_1$ is the identity on the triangle, and also $\mathcal{M}_1 \circ \mathcal{M}_2$ is the identity on the triangle. However, if we apply $\mathcal{M}_1 \circ \mathcal{M}_2$ to the cyclic 2-gon

$$a \rightarrow\!\bigcirc\!\rightarrow b \overset{\mathcal{M}_2}{:=} a \rightarrow\!\bigcirc\!\bigcirc\!\rightarrow b \overset{\mathcal{M}_1}{:=} a \rightarrow\!\bigcirc\!\bigcirc\!\rightarrow b \, , \tag{134}$$

we find that it does not map the 2-gon to itself. This is again fine, as the equation between the very left and the very right is a derived move of the simplified liquid,

$$
\begin{aligned}
a \rightarrow\!\bigcirc\!\rightarrow b &\overset{\overline{(119)}}{=} a \rightarrow\!\bigcirc\!\bigcirc\!\rightarrow\!\bigcirc\!\rightarrow b \\
&\overset{\overline{(127)}}{=} a \rightarrow\!\bigcirc\!\bigcirc\!\rightarrow\!\bigcirc\!\rightarrow b \\
&\overset{(123)^2}{=} a \rightarrow\!\bigcirc\!\bigcirc\!\rightarrow\!\bigcirc\!\rightarrow b \overset{(125)}{=} a \rightarrow\!\bigcirc\!\bigcirc\!\rightarrow b \, .
\end{aligned}
\tag{135}
$$

If we apply $\mathcal{M}_1 \circ \mathcal{M}_2$ to any model of the simplified liquid, we will get the same model again. So the models of the simplified liquid are in one-to-one correspondence with the models of the non-simplified liquid, which motivates the use of the word "equivalent". We will call two mappings such that both $\mathcal{M}_2 \circ \mathcal{M}_1$ and $\mathcal{M}_1 \circ \mathcal{M}_2$ are the identity up to moves *weak inverses* of another. Two liquids are considered *equivalent* if there are mappings between them which are weak inverses of another.

One might think that reducing the number of moves from 8 to 5 is not a significant improvement. Let us justify why it actually is. The key task is finding models for our liquid, which means solving the tensor-network equations given by the moves. As a measure of "complexity" of a liquid it thus makes sense to consider the computational cost of evaluating the two networks of each move, and in particular its scaling with the index dimension $d$. This scaling is always polynomial, but the exponents depend on the move. Very roughly, the exponent will increase proportionally to the "linear size" of a network. Thus, we have a strong preference for moves with small networks. For evaluating a 2-2 Pachner move we need of the order of $d^5$ $+$ and $\cdot$ operations. The same holds for a 1-3 Pachner move. All other moves in this section have smaller exponents and thus have a vanishing contribution to the overall complexity when scaling $d$. So from that perspective we have reduced the complexity from 7 moves to 1 move rather than from 8 to 5 moves.

## 4.5  Models

We might look for models of the liquid with complex tensors as tensor type. However, we will see in Section 6, that such models are unphysical, as they are not Hermitian. In contrast, models with real tensors as tensor type have a physical interpretation, namely as fixed-point models for topological order in spin systems protected/enriched by a *time-reversal symmetry*: For a spin system, a time-reversal symmetry is an anti-unitary which squares to the identity. We can always change the basis, such that this anti-unitary is given by complex conjugation in that basis. Then, obeying the symmetry means that all tensors of the model are only allowed to have real entries.

### 4.5.1  Matrix algebra models

The point of this section was to get rid off all index permutation symmetries. For the related algebra liquid, this corresponds to removing the commutativity axiom. Thus, also non-commutative algebras yield models of the new liquid, such as the algebra of $n \times n$ matrices

(for any $n$):

$$
\begin{array}{c}
a_0 b_0 \quad a_1 b_1 \\
\underset{a_2 b_2}{M}
\end{array}
= (n^{-1/2})
\begin{array}{c}
a_0 \qquad b_1 \\
b_0 \qquad a_1 \\
\\
b_2 a_2
\end{array}
\quad . \tag{136}
$$

However, we observe that this model is not in an interesting phase. If we consider the network representing some triangulation and use the tensors from Eq. (136), we see that it decomposes into disconnected loops around vertices and scalars $n^{-1/2}$ at every triangle. Each loop evaluates to the scalar $n$. Physically, a tensor network consisting only of scalars corresponds to a trivial model without any degrees of freedom.

Another way to motivate that this model is trivial is to see that due to the quantum mechanical interpretation of the model we can generally neglect scalar pre-factors. This is because the predictions of a quantum model are tensors whose entries are probabilities of measurement outcomes, which have to sum to 1. Alternatively, we can be fine with any tensor and fix the latter constraint by hand by normalizing with a prefactor. Then, tensors which differ by a prefactor correspond to the same physical predictions. The measurement-outcome tensors can be obtained by simply contracting space-time tensor networks containing the time-evolution tensors of the model as well as state-preparation and measurement tensors [17]. Instead of neglecting prefactors after contraction, we can already do this at the level of the single tensors constituting the model. As neglecting pre-factors is compatible with Kronecker products and Einstein summations, "arrays modulo pre-factors" defines another tensor type, which we will refer to as *projective tensors*. If we interpret the model in terms of projective tensors, it is actually formally in a trivial phase.

Mathematically, the evaluation of such a model can be computed as a sum of local numbers after taking the logarithm of each scalar, which is known as a *classical invariant* of a manifold. Simple combinatorics shows that the evaluation is given by $n^\chi$, where $\chi$ is a classical invariant known as the *Euler characteristic* of the manifold.

### 4.5.2 Quaternion models

Another model is given by the quaternion algebra, whose indices take values in the set $\{\mathbf{1}, \mathbf{i}, \mathbf{j}, \mathbf{k}\}$,

$$
\underset{c}{\overset{a \quad b}{\mathbb{H}}}
=
\begin{cases}
1/2 & \text{if} \quad b = \mathbf{1} \text{ and } a = c \\
1/2 & \text{if} \quad a = \mathbf{1} \text{ and } b = c \\
1/2 & \text{if} \quad (a, b, c) \text{ is even permutation of } (\mathbf{i}, \mathbf{j}, \mathbf{k}) \\
-1/2 & \text{if} \quad (a, b, c) \text{ is even permutation of } (\mathbf{i}, \mathbf{k}, \mathbf{j}) \\
-1/2 & \text{if} \quad c = \mathbf{1} \text{ and } a = b \\
0 & \text{otherwise.}
\end{cases}
\tag{137}
$$

If we interpret this algebra as a complex algebra, it is isomorphic to the algebra of $2 \times 2$ matrices, which would correspond to a physically trivial model again. However, as a real algebra it is distinct from any matrix algebra or $\delta$-algebra, and corresponds to a non-trivial phase. This can again be seen by evaluating the model for a closed network representing a non-orientable manifold, e.g., on the real projective plane, where we get $-2$. The fact that the model becomes trivial when we drop the reality constraints indicates that we have a model for a time-reversal *SPT phase*, i.e., a phase which becomes trivial after we allow breaking the symmetry, in contrast to a *symmetry breaking phase* or a *symmetry enriched topological (SET) phase*.

### 4.5.3 Cluster Hamiltonian

The quaternion algebra model is equivalent to a commuting-projector model known as *cluster Hamiltonian* [28, 29], which is known to represent the only non-trivial SPT phase protected by time-reversal symmetry in $1 + 1$ dimensions [30]. The Hamiltonian is given by

$$H = \sum_i -X_{i-1}Z_i X_{i+1} \,. \tag{138}$$

Its time reversal symmetry is given by the anti-unitary operator

$$T = K \bigotimes_i Z \,, \tag{139}$$

where $K$ denotes complex conjugation. After a change of basis, the time-reversal symmetry operator is given by complex conjugation in that basis alone:

$$H = \sum_i -Y_{i-1}Z_i Y_{i+1} = \sum_i (XZ)_{i-1}Z_i(XZ)_{i+1} \,, \tag{140}$$
$$T = K \,.$$

The local ground state projector acting on three neighbouring qubits is given by

$$P = (1 - XZ \otimes Z \otimes XZ)/2 \,. \tag{141}$$

In order to compare the cluster Hamiltonian with our liquid model, we actually have to break translation invariance, and block pairs of neighbouring qubits. The new local ground state projector acting on two qubit pairs is given by the product of two old ground state projectors, i.e.,

$$P_{\text{blocked}} = (1 - XZ \otimes Z \otimes XZ \otimes \mathbb{1})(1 - \mathbb{1} \otimes XZ \otimes Z \otimes XZ)/4 \tag{142}$$
$$= (1 - XZ \otimes Z \otimes XZ \otimes \mathbb{1} - \mathbb{1} \otimes XZ \otimes Z \otimes XZ - XZ \otimes X \otimes X \otimes XZ)/4 \,.$$

As in Section 3.1, this projector is interpreted as a 4-index tensor

$$\begin{array}{c} \text{out1} \qquad \text{out2} \\ \square \\ \text{in1} \qquad \text{in2} \end{array} \,, \tag{143}$$

which defines a model of a liquid for rhombus-like cellulations of space-time.

Again, the comparison between the topological liquid model and the commuting-projector model is done by a liquid mapping. As before, the geometric interpretation is given by refining the "rhombic cellulation" of spacetime into a triangulation

$$\tag{144}$$

The only difference is that now the triangulation has a branching structure. In network notation, we get

$$\tag{145}$$

To show that the mapping above is in fact an equality for the chosen models, we identify the basis elements of the quaternion algebra $\{\mathbf{1}, \mathbf{i}, \mathbf{j}, \mathbf{k}\}$ with the two-qubit configurations

$\{|0,0\rangle, |0,1\rangle, |1,0\rangle, |1,1\rangle\}$ and write the tensors appearing in Eq. (145) as a collection of two-qubit operators from the vector space associated to index $a$ to the vector space associated to the index $b$, indexed by the index $c$

$$
\begin{aligned}
&\overset{b}{\underset{a}{\bigcirc}}\!\!-c = (\mathbb{1}\otimes\mathbb{1}, \mathbb{1}\otimes XZ, XZ\otimes Z, XZ\otimes X)/2\,, \\[2em]
&c\!-\!\!\overset{b}{\underset{a}{\bigcirc}} = (\mathbb{1}\otimes\mathbb{1}, -Z\otimes XZ, -XZ\otimes\mathbb{1}, X\otimes XZ)/2\,.
\end{aligned}
\tag{146}
$$

Summing over the index $c$ yields the right hand side of Eq. (142), which shows that our liquid model is equivalent to the cluster Hamiltonian model under the chosen mapping.

# 5 A non-triangle liquid

In this section, we will discuss a first example for a liquid which is not directly based on triangulations. We will then show how liquid mappings can be used to establish its equivalence to the triangle liquid on a purely diagrammatic level.

## 5.1 The edge liquid

The new topological liquid in $1+1$ dimensions presented in this section, the *edge liquid*, corresponds to a different way of combinatorially representing 2-manifolds. As the name suggests, instead of representing each triangle by a tensor, we associate a copy of a 4-index tensor to every edge. More precisely, each edge and the associated tensor are decorated with little arrows, which allows us to introduce an orientation later in Section 6 by assigning a clockwise- or counter-clockwise 'helicity',

$$
\bullet\!-\!\!\nearrow\!\!-\!\bullet \quad \rightarrow \quad \boxtimes\,.
\tag{147}
$$

An edge with different helicity is represented by the same tensor, just flipped,

$$
\bullet\!-\!\!\searrow\!\!-\!\bullet \quad \rightarrow \quad \boxtimes\,.
\tag{148}
$$

Two edge tensors share a common bond if they are adjacent to a common vertex and a common triangle. This description also works for cell complexes with arbitrary $n$-gons as faces instead of just triangles, e.g.,

$$
\tag{149}
$$

There are 3 moves. The first move has a geometric representation as taking the endpoint

of one edge and moving it along another edge,

$$\leftrightarrow \qquad . \tag{150}$$

In terms of networks, we have

$$= \qquad . \tag{151}$$

The second move corresponds to removing a 'dangling' edge with a vertex that is only adjacent to this edge,

$$\leftrightarrow \qquad . \tag{152}$$

In terms of networks, this is

$$= a \text{———} b \ . \tag{153}$$

The third move is dual to the latter and removes a 'loop' edge which has the same endpoint twice,

$$\leftrightarrow \qquad . \tag{154}$$

In terms of networks,

$$= a \text{———} b \ . \tag{155}$$

## 5.2 Equivalence of the liquids

For the triangle liquid, we could make use of a theorem by Pachner to argue that it is a 'topological' liuid. For the edge liquid, such a formal argument is missing, though it seems conceivable that the presented deformations of cellulations are as powerful as Pachner moves. Instead of trying to directly proof a second Pachner theorem for the edge liquid, we take a simpler route and show its local combinatorial equivalence to the triangle liquid. As in Section 4.4, we prove this equivalence using two weakly inverse liquid mappings, from the edge to the triangle liquid and back. Starting with the former, we can map a cellulation underlying an edge-liquid network to a triangulation underlying a triangle-liquid network by replacing every edge by two triangles,

$$\rightarrow \qquad . \tag{156}$$

In other words, the triangulation is obtained by subdividing each face into triangles in a 'pizza-like' manner, with a new vertex in the middle. Formally, this liquid mapping from the edge liquid to the triangle liquid is given by

$$\tag{157}$$

Applying the mapping above to the move in Eq. 155, we obtain the equation

$$\tag{158}$$

which can be derived from the complete set of triangle-liquid moves. This is also easily seen for the other moves of the edge liquid. Thus, under the mapping defined in Eq. (157) a model of the triangle liquid is mapped to a model of the edge liquid.

Let us now give the opposite mapping from the triangle liquid to the edge liquid. A branching structure triangulation can be turned into a cellulation by replacing every triangle by two of its edges,

$$\tag{159}$$

Note that an edge of the triangulation might yield two edges of the cellulation, one coming from both adjacent triangles, enclosing a 2-gon face. It might also yield one or no edges, depending on the branching structure of the two adjacent triangles. Formally, the mapping is given by

$$\tag{160}$$

As indicated by the two-letter labels on the left-hand side, each index of the triangle liquid corresponds to two indices of the edge liquid. If we want to use this formula to get a triangle-liquid model from an edge-liquid model, we have to evaluate the right-hand side, and then block pairs of indices into single indices.

In order to establish that the two mappings above are weak inverses we in fact need to generalize our notion of a weak inverse. As in Section 4.4, applying both mappings $\mathcal{A} \to \mathcal{B} \to \mathcal{A}$ to a single tensor does not result in a network consisting of that same tensor again, but to a larger network. However, in contrast to the mappings in Section 4.4, the resulting network is not even related to the original 1-tensor network via the moves of the liquid. This is obvious from the fact that the open indices after applying both mappings are different after applying both mappings. The same is true for the other composition of the two mappings.

It is still true, however, that the two mappings applied to a network without open indices is equivalent to the original network again via moves, since this double-mapping is nothing but a refinement of the triangulation/cellulation. If there are open indices then the networks are equivalent up to moves everywhere in the interior away from the open indices. Thus, for our new generalized notion of weak inverse, it suffices if a doubly-mapped network is equivalent to the original network via moves only away from the open indices.

The two mappings being not literal inverses of another implies that applying both mappings in the reverse direction to a model of $\mathcal{A}$, we will in general not end up with the same model

again. Indeed, as one of the mappings involves a blocking of indices, the bond dimension of the resulting model will be the square of the original bond dimension. However, the two mappings being weak inverses of another implies that the model and the twice-mapped model will be related by an invertible domain wall. So even if the original and the resulting models are different, they are in the same exact phase. Thus, the phases of equivalent liquids, such as the edge liquid and the triangle liquids, are in one-to-one correspondence. They are therefore equally powerful for classifying or representing phases of matter. However, the representation of a phase as model of $\mathcal{A}$ might be more convenient than its representation as a model of $\mathcal{B}$, or vice versa.

# 6 Orientation and Hermiticity

In this section we will provide a liquid whose models have a standard quantum mechanical interpretation, by adding an orientation and a Hermiticity move. We will illustrate this at hand of the triangle liquid from Section 4, but the generalization to arbitrary topological liquids is straight-forward. The corresponding models are basically equal to 2-dimensional *lattice TQFTs* as formulated in Ref. [6].

## 6.1 Hermiticity and orientation-reversal

Objects like Hamiltonians, state vectors or time evolution operators, which occur in the usual pure-state formulation of quantum mechanics, are complex tensors. A "physical" Hamiltonian is Hermitian, which means that interchanging input and output indices of the corresponding complex tensor is equal to complex conjugation, e.g.,

$$
\begin{array}{ccc}
\overset{a \quad b}{\boxed{H}} = \overset{a' \ b'}{\boxed{H^*}} := \overset{a' \ b'}{\left(\boxed{H}\right)_K} \\
\underset{a' \ b'}{} \quad \underset{a \ b}{} \quad \underset{a \ b}{}
\end{array}
\tag{161}
$$

As complex conjugation is not part of network notation, we introduce the following extension to network notation. Every part of a network encircled by a line of the following style

$$
\wwwwww{K}\wwwwww
\tag{162}
$$

will be complex conjugated. We will sometimes omit the label $K$. Complex conjugation commutes with tensor products and contractions, which gives us diagrammatic equivalences such as

$$
b\overset{a}{\underset{c}{\square}}\overset{d}{\underset{e}{\triangleright}} = b\overset{a}{\underset{c}{\square}}\overset{d}{\underset{e}{\triangleright}} = b\overset{a}{\underset{c}{\square}}\overset{d}{\underset{e}{\triangleright}} = b\overset{a}{\underset{c}{\square}}\overset{d}{\underset{e}{\triangleright}} .
\tag{163}
$$

The Hermiticity of the Hamiltonian carries over to the tensors of the tensor network in the Trotterized imaginary time evolution, and implies that inverting the time direction is equivalent to complex conjugation. In a topological manifold there is no "time direction", but inverting any direction is still an orientation-reversing map. Thus, a "physical" model of a topological liquid in complex tensors should have the property that orientation reversal equals complex conjugation. The networks of the liquids we introduced so far represent manifolds without an orientation, so it's impossible to formulate the Hermiticity condition for their models.

On the other hand, we could say that for liquids without orientation, orientation reversal is a trivial operation. Thus, for models of such liquids the Hermiticity condition implies that they are invariant under complex conjugation alone, i.e., purely real. As we have seen Section 4, such models are indeed physical and correspond to phases with a time-reversal symmetry.

There is the possibility of real models of unoriented liquids to emulate general physical models (even ones without time-reversal symmetry) by increasing the bond dimension, called *realification* (cf. Ref. [17,31]). Realification is an operation that maps every Hermitian complex model of an oriented liquid to a real model of the corresponding unoriented liquid, such that the former can be identified with a subset of the latter. However, it is more straight-forward to add an orientation to the liquid and consider models with complex tensors.

## 6.2 Non-simplified liquid

An orientation can be added to a triangulation by specifying for each triangle whether it is oriented "clockwise", or "counter-clockwise". Clockwise and counter-clockwise triangles are represented by two different tensor variables in a network. The clockwise triangle is defined by the fact that its 01 edge (with respect to the branching structure) is oriented clockwise,

$$\tag{164}$$

The opposite is true for the counter-clockwise triangle

$$\tag{165}$$

In network notation, the two tensors are distinguishable, as we add an inward arrow marker to every index corresponding to a clockwise oriented edge. The clockwise triangle has two clockwise edges, whereas the counter-clockwise triangle only has one. As we allow reflecting the shapes of individual tensor copies in a network, it would be impossible to distinguish the two input indices of the clockwise triangle. To fix this problem, we add a little "spiral" to the circle, which defines what the counter-clockwise direction is.

In an oriented triangulation every edge is a clockwise edge of one triangle, and a counter-clockwise edge of another triangle. Thus, the networks obey the constraint that every bond is between an index with an arrow and an index without an arrow. Alternatively, the diagrams can be interpreted as instances of a slightly refined graphical calculus, where indices are divided into *output* and *input* indices, and bonds must always connect one input and one output index. The refined graphical calculus can be fulfilled by more general data structures, namely tensor types where each basis has a *dual*. For all the tensor types in this work (i.e., arrays and fermionic tensors) the dual will be trivial. Thus, we will not explicitly distinguish input and output indices.

Each Pachner move exists with two different orientations as well. So naively we would end up with a liquid with 14 Pachner moves plus the triangle cancellation move (which is reflection symmetric), which we will call the "non-simplified liquid". Alternatively, we could take the simplified unoriented liquid with two copies of every tensor variable and every move (unless they are reflection symmetric). However, there is a simpler equivalent liquid, as the next section shows.

## 6.3 Simplified liquid

The simplified liquid contains the clockwise triangle in Eq. (164) as a tensor variable, but not the counter-clockwise triangle. The latter can be constructed from the former by gluing with a cyclic 2-gon, as shown in Eq. (175). For the cyclic 2-gon, we take both the clockwise and counter-clockwise version,

$$\tag{166}$$

The moves only contain a single 2-2 Pachner move, namely the one consisting of only clockwise triangles

$$\tag{167}$$

In the oriented case, the triangle only has a $\mathbb{Z}_3$ rotation symmetry, generated by the *(120) triangle symmetry move*

$$\tag{168}$$

In network notation we have

$$\tag{169}$$

Furthermore, there are two cancellation moves. The triangle cancellation move depicted in Eq. (118) has one clockwise and one counter-clockwise triangle. The latter is not part of our tensors, so the *oriented triangle cancellation move* has the cyclic 2-gon instead a free bond on the other side:

$$\tag{170}$$

In network notation, we find

$$\tag{171}$$

Second, the *oriented 2-gon cancellation move* is

$$a \rightarrow\!\!\bigcirc\!\!\leftarrow \bigcirc\!\!- b = a - b \ . \tag{172}$$

The clockwise 2-gon is rotation symmetric, so we would expect the following symmetry move

$$a \rightarrow\!\!\bigcirc\!\!\leftarrow b \stackrel{\text{sym}}{=} (ab) \,, \tag{173}$$

which is also implied by the choice of shape. Indeed, this move is directly derived from the oriented triangle cancellation move in Eq. (171). The analogous symmetry move for the counter-clockwise 2-gon

$$a -\!\!\bigcirc\!\!- b \stackrel{\text{sym}}{=} (ab) \tag{174}$$

can be derived from the oriented 2-gon cancellation move. The proof that the non-simplified and simplified liquids are equivalent is analogous to the non-oriented case.

## 6.4 Hermiticity move

As we mentioned above, physical models should obey the Hermiticity condition that orientation reversal equals complex conjugation. Using the extended network notation introduced in Eq. (162), this condition can be written down as a move as well. This move equates the complex conjugated clockwise triangle and the counter-clockwise triangle. The latter can be constructed from the former by gluing a cyclic 2-gon according to

$$\tag{175}$$

One would expect that we also need to add the analogous move which relates the clockwise 2-gon and the counter-clockwise 2-gon via complex conjugation. However, this move can be derived from the moves defined so far. For the sake of demonstrating how to operate with networks containing (complex conjugation) mappings, we will explicitly give the derivation

$$\tag{176}$$

## 6.5 Models

As in the unoriented case, the oriented liquid is equal to associative algebras with some extra axioms, and thus, its models can be classified. Complex models of the oriented Hermitian liquid are not actually more general than real models of the unoriented liquid: By a change of basis, each complex model can be brought into a form where it is purely real. Contrary, there are even less models, in the sense that models which are in different phases as real models can be in the same phase as complex models.

An example for this is the model coming from the quaternion algebra. As a complex model, it is equal to the model coming from the $2 \times 2$ matrix algebra, after the following basis change:

$$G := 2^{-1/2}(\mathbb{1}, iX, iZ, iY) , \tag{177}$$

where $X$, $Z$ and $Y$ are the corresponding Pauli matrices, and the four entries correspond to $\mathbb{1}$, $\mathbf{i}$, $\mathbf{j}$, and $\mathbf{k}$. If we choose an ordering of the four entries of $2 \times 2$ matrices, we can write $G$ properly as a $4 \times 4$ unitary matrix.

# 7 Beyond-topological moves

Topological fixed-point models might be further restricted by imposing invariance under moves beyond topological invariance. In this section, we discuss one simple example for this, namely the restriction to so-called *invertible* phases. A model or phase is said to be invertible if "stacking two orientation-reversed copies yields a trivial phase". Thus, there must be an invertible domain wall between this double-layered tensor network and the trivial model. One way of ensuring this is to demand invariance not only under homeomorphisms, but also under the following *surgery operations*:

- A 0-surgery (or, equivalently, a backwards 3-surgery) consists in removing a 2-sphere

$$\leftrightarrow \qquad . \tag{178}$$

- A 1-surgery (or backwards 2-surgery) consists in cutting out an annulus and pasting two disks

$$\leftrightarrow \qquad . \tag{179}$$

Two manifolds are related by surgery operations iff they are cobordant, i.e., their disjoint union can be identified with the boundary of a manifold of one dimension higher. Two layers of 2-manifold can be removed with the invertible domain wall using surgery operations as

$$\rightarrow \qquad \rightarrow \qquad \rightarrow \qquad . \tag{180}$$

We start by applying 2-surgeries with one disk in each of the two layers, as indicated by the blue circles. This yields a "double-layer with holes". For each pair of neighbouring holes there is a non-contractible loop winding through both of them. Next, we apply a 1-surgery to the annulus-like neighbourhood of every such non-contractable loop, (whose boundaries were indicated by blue lines). This yields a collection of disconnected 2-spheres, which can be removed by 0-surgeries.

Combinatorially, surgery operations can be implemented by the following moves.

- The 0-*surgery move*

$$\leftrightarrow \qquad , \tag{181}$$

where the left hand side depicts a cellulation of a sphere by a clockwise (front) and a counter-clockwise (back) 2-gon, and the right hand side is the empty manifold. In network notation, this looks like

$$= \qquad . \tag{182}$$

- The 1-*surgery move*

$$\leftrightarrow \qquad , \tag{183}$$

where the left hand side depicts a triangulation of an annulus, and on the right we have a triangulation of two disks. In network notation, we find

$$00 \rightarrow \overset{001}{\bullet} \overset{011}{\bullet} \leftarrow 11 = 00 \rightarrow \overset{002}{\bullet} \quad \overset{113}{\bullet} \leftarrow 11 . \tag{184}$$

Note that both moves are non-topological, as at least one of their networks (in fact both) does not represent a disk. In particular, the right hand side of the 1-surgery move consists of two disconnected components. Also note that it does not matter how exactly we implement the surgery operation concretely in terms of networks. All equations between networks of the correct topology are equivalent via the topology-preserving moves. This is true for general topology-changing moves.

Models of the invertible liquid are models of the topological liquid which fulfil the additional equations Eq. (182) and Eq. (184). As such only the trivial model (every tensor being equal to the number 1) fulfils these equations. However, physically it is fine if the equations only hold up to pre-factors, such that we can ignore Eq. (182) as it is only equates scalars. In this case, also the model based on the matrix algebra in Eq. (136) is invertible, as plugging it into Eq. (184) yields

$$
\begin{array}{c} a \\ a' \end{array} \!\!\!\!\!\!\!\!\! \bigcirc\!\!\!\bigcirc \!\!\!\!\!\!\!\! \begin{array}{c} c \\ c' \end{array} = \begin{array}{c} a \\ a' \end{array} \!\!\!\!\!\!\!\! \bigcirc \quad \bigcirc \!\!\!\!\!\!\! \begin{array}{c} c \\ c' \end{array} . \tag{185}
$$

As we have seen in Section 4.5, this model is still in a trivial phase, as the defining tensor is a tensor product of three identity matrices and thus the resulting network can be reshaped into a product of independent loops. However, it can become non-trivial if we add symmetries. That is, we equip each index with a representation of a group (or some form of Hopf algebra, see Ref. [13]). Tensors with symmetries (of a fixed group) constitute a different tensor type, as symmetries are consistent with contraction and tensor product.

The tensor in Eq. (136) can be equipped with many different symmetries. In particular having a representation act twice on each of the two (row and column) index components independently leaves the tensor invariant, because each of the three identity matrices is invariant under that symmetry separately. For that exact same reason though, the model with such symmetries is still in a trivial phase. So we need a representation that does not split into a product of two representations on the two index components. One possibility for that is to take a projective representation on each of the two components, such that both projective representations together form a proper representation on the composite index. The simplest group which has a non-trivial projective representations is the Klein 4-group $\mathbb{Z}_2 \times \mathbb{Z}_2$. The projective representation $R$ is given by the Pauli matrices:

$$
\begin{aligned}
R((0,0)) &= \mathbb{1}, & R((0,1)) &= X, \\
R((1,0)) &= Z, & R((1,1)) &= iY .
\end{aligned}
\tag{186}
$$

In Section 4.5, we have seen models for SET (or SPT) phases protected by a time-reversal symmetry. The present model is an example for an SPT phase protected by an ordinary symmetry, which is also invertible. Note that projective representations are classified by the second $U(1)$-valued cohomology group of the symmetry group, and our liquid models are equivalent to the isometric MPS in Ref. [32] and to the models in terms of "dimer crystals" in Ref. [30].

# 8 Non-chiral topological order in $2+1$ dimensions

In this section, we discuss *non-chiral intrinsic* topological order for spin systems, i.e., systems without fermionic degrees of freedom. Whereas global symmetries and fermions (see Section 9) can be easily incorporated into our framework, it is an open question whether there exist models for topological liquids which represent chiral phases. For all liquids presented in this paper there are mappings from a commuting-projector liquid, and there exist no-go theorems about commuting-projector models describing chiral phases [33]. In our framework we

can circumvent those no-go theorems as there exist more general liquids which do not yield commuting-projector models (cf. Ref. [12]), however, concrete examples of models representing chiral phases remain elusive to date. On the contrary non-chiral topological order is well captured within our formalism and we illustrate this fact by providing two different, yet equivalent topological liquids that cover the most general known models of non-chiral topological order.

## 8.1 Volume liquid

In this section we describe a liquid whose models are similar to fixed point models originally introduced as a state-sum invariant by Turaev and Viro [7,8]. Later this construction has been rephrased as a Hamiltonian model for topological order by Levin and Wen [5] referred to as string-net models. The liquid we present here is a straight forward generalization of the oriented topological liquid in $1+1$ dimensions from Section 6 to $2+1$ dimensions.

### 8.1.1 The non-simplified liquid

A 3-manifold can be represented by a simplicial complex (a decomposition of the manifold into tetrahedra) with the following 3-dimensional Pachner moves. The 2-3 Pachner move replaces two tetrahedra glued together at a single face with three tetrahedra glued together such that each pair of tetrahedra shares one common face and all three tetrahedra share a common edge

$$\leftrightarrow \qquad . \tag{187}$$

The 1-4 Pachner move replaces a single tetrahedron with 4 tetrahedra, such that every pair shares a common face, every collection of three tetraheda shares a common edge and all tetrahedra share a common vertex

$$\leftrightarrow \qquad . \tag{188}$$

A triangulation is represented by a network with one 4-index tensor at every tetrahedron, and one bond between each pair of tetrahedra sharing a face. In order to obtain a liquid with models for a very general class of topological phases, we have to take care of the following details.

To properly represent 3-dimensional manifolds combinatorially, we need to distinguish the different faces of a tetrahedron. On a geometrical level this can be achieved by introducing a branching structure. That is, analogously to the 2-dimensional case, we add an orientation to all edges which is not cyclic around any triangle. The branching structure allows us to uniquely label the vertices of a triangle,

$$\rightarrow \quad T_{\mid 012} \, , \tag{189}$$

which represents a bond dimension variable $T$. This ensures that there is only one way to glue two triangles. It also allows us to uniquely label the vertices of the tetrahedron, yielding a

tensor variable with distinct indices

$$T \xrightarrow[012]{\overset{013}{\underset{123}{0123}}} T .$$

(190)

In network notation the 4 indices are distinguished by their location relative to the small black "arrow" inside the square which allows an unambiguous identification despite the fact that we are allowed to rotate and reflect the shape in the diagrams.

We want the models to have a pure-state quantum mechanical interpretation. Thus, we have to work with complex tensors, and we need to introduce an orientation. The orientation allows us to distinguish between the *counter-clockwise* tetrahedron above (whose 01 edge of the 012-triangle is oriented counter-clockwise), and the clockwise tetrahedron which is represented by the different tensor variable

$$T \xrightarrow[012]{\overset{013}{\underset{123}{0123}}} T .$$

(191)

In order to access even more general models, we choose a slightly more complicated network representation of the triangulation. For every edge encircled by tetrahedra we chose one favorite adjacent face shared by one tetrahedron-pair and insert a 2-index tensor at the corresponding bond. Those tensors correspond to three different variables (called *edge weights*), depending on whether the edge is the 01, the 02, or the 12 edge of its favourite face represented by

$$T \xrightarrow[\text{back}]{\overset{012}{\phantom{0}}} T ,$$

(192)

$$T \xrightarrow[\text{back}]{\overset{012}{\phantom{0}}} T ,$$

(193)

$$T \xrightarrow[\text{back}]{\overset{012}{\phantom{0}}} T .$$

(194)

We can imagine to "inflate" the triangle on the left-hand side of Eq. (192), Eq. (193), and Eq. (194) and into a pillow-like volume with three corners, whose boundary consists of two triangles, one in the back and one in the front. We might think of the edge weight as being contained in the volume. Here and in the following, we will mark edges at the boundary of a volume, which contain an edge weight, with a tick.

It turns out that if we would write down the liquid without edge weights, we would only get models for symmetry breaking order, and none for (actual, irreducible) topological order [3]. However, there are simpler ways to decorate the liquid, which already have non-trivial models,

---

[3]The edge weight are closely related to the quantum dimensions in the conventional fusion-category framework of non-chiral topological order. No edge weights would mean that all quantum dimensions and the total quantum dimension are 1.

yet not the most general ones. E.g., it suffices to add a 0-index tensor [4] associated to every vertex, in order to get models for all discrete (Dijkgraaf-Witten [34]) gauge theories.

After the refinements above, we do not have a single 2-3 and 1-4 Pachner move, but one move for each choice of orientations, branching structure, and possitions of the edge weights. A list of all moves can be obtained in a straight-forward fashion and here we only present one specific example of a 2-3 Pachner move with the special property that all tetrahedra are oriented counter-clockwise. This move will be relevant in the next section, where we present a simplified, yet equivalent liquid. In terms of cell complexes, it looks like

$$\text{(195)}$$

We observe that the geometric depiction does not reveal where we put the edge weight of the inner 13 edge on the right hand side. However, this information is contained in the corresponding network notation

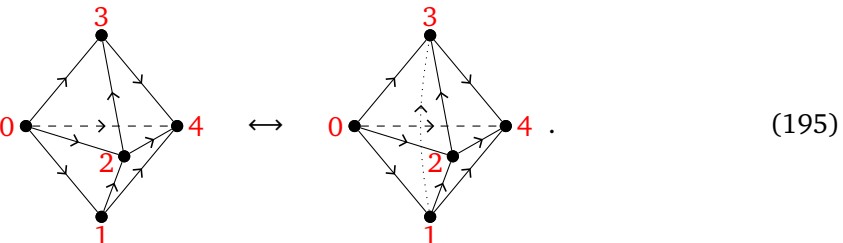

$$\text{(196)}$$

Apart from the 2-3 and 1-4 Pachner moves, we impose the following *full tetrahedron cancellation move* analogous to the *triangle cancellation move* in Eq. (119). Geometrically, it consists in taking a volume glued from a clockwise and a counter-clockwise tetrahedron at three of their faces, and shrinking it down to a single face

$$\text{(197)}$$

In network notation, this face is represented by a free bond

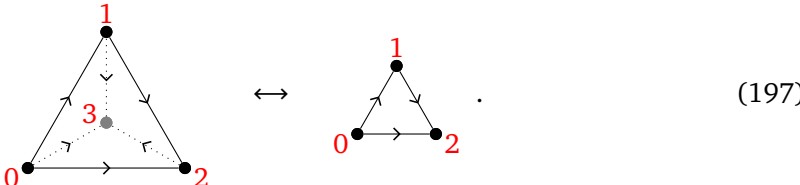

$$\text{(198)}$$

As in the $1+1$-dimensional case, the volume on the left hand side of Eq. (197) would be represented by a projector in a real/complex model, and the move corresponds to the convention of restricting everything to the support of that projector.

---

[4]The corresponding scalar would be the inverse total quantum dimension in the fusion-category formulation.

### 8.1.2 The simplified liquid

The liquid presented in the preceding section is quite complicated, as it consists of a large number of slightly different versions of Pachner moves. In the following we present an equivalent "simplified" liquid with only one single Pachner move, together with many simple additional moves. The simplified liquid has a geometric interpretation as well – networks do not correspond to triangulations, but more generally to cellulations with different faces and volumes.

The simplified liquid consists only of the counter-clockwise tetrahedron and several additional tensors which can be used to flip the edge orientations, and thus allow us to effectively reconstruct the clockwise tetrahedron from the counter-clockwise one (cf. Eq. (217)). The main new ingredient of the simplified liquid is to allow for 2-gon faces. Thus, first of all, we introduce an additional bond dimension variable $D$, corresponding to a 2-gon with cyclic edge orientations

$$0 \overset{\curvearrowright}{\underset{\curvearrowleft}{\bullet\quad\bullet}} 1 \quad \rightarrow \quad D\Big|_{01} \,. \tag{199}$$

The 2-gon has a rotation symmetry, so there are 2 different ways to identify two glued 2-gons. In order to make the gluing unambiguous, we determine one "favourite edge", marked by the small half circle, such that those favourite edges have to coincide when gluing.

The new tensors used to flip edge orientations are called *flip hats*. They correspond to 3-cells whose boundary consists of two triangles and one 2-gon and which appear in four different variants depending on orientation and the choice of the favorite edge. I.e., there is

- the *clockwise* 01 *flip hat*

$$ \rightarrow \quad T \xrightarrow{\phantom{xx}102\phantom{xx}} \overset{012}{\underset{01}{\square}} \overset{012}{\phantom{xx}} T \,, \tag{200}$$

- the *counter-clockwise* 01 *flip hat*

$$ \rightarrow \quad T \xrightarrow{\phantom{xx}102\phantom{xx}} \overset{012}{\underset{01}{\blacksquare}} \overset{012}{\phantom{xx}} T \,, \tag{201}$$

- the *clockwise* 12 *flip hat*

$$ \rightarrow \quad T \xrightarrow{\phantom{xx}021\phantom{xx}} \overset{012}{\underset{12}{\square}} \overset{012}{\phantom{xx}} T \,, \tag{202}$$

- and the *counter-clockwise* 12 *flip hat*

$$ \rightarrow \quad T \xrightarrow{\phantom{xx}021\phantom{xx}} \overset{012}{\underset{12}{\blacksquare}} \overset{012}{\phantom{xx}} T \,. \tag{203}$$

In addition, there is the 2-*gon flip* which interchanges favourite edges

$$0 \overset{\curvearrowright}{\underset{\curvearrowleft}{\bullet\quad\bullet}} 1 \quad \rightarrow \quad D \xrightarrow{\phantom{x}10\phantom{x}} \overset{01}{\Diamond} \xrightarrow{\phantom{x}01\phantom{x}} D \,. \tag{204}$$

The boundary of this volume consists of two 2-gons. The favourite edge of the 2-gon on the front is the 01 edge, whereas for the 2-gon on the back it is the 10 edge.

At last, we need to introduce the edge weights for the simplified liquid. Of the three edge weights from the previous section, it suffices to take the 01 edge weight in Eq. (192), since the other edge weights can be constructed using the tensors above. We additionally introduce the 2-*gon edge weight*

$$
0 \bullet \!\!\!\!\!\overset{\mathscr{G}}{\longleftrightarrow}\!\!\!\!\! \bullet 1 \quad \rightarrow \quad D \xrightarrow[\text{back}]{01} \xrightarrow{} \text{front} D \,,
\tag{205}
$$

which is a volume like the 2-gon flip, but the favourite edge of both the back and front 2-gon is the 10 edge. According to the name, one of its edges (the 10 edge) carries an edge weight and is therefore marked by a tick. We show in Eq. (215) that in fact all edge weights can be constructed from the 2-gon edge weight only, such that the 01 edge weight is merely an auxiliary tensor.

The moves of the simplified liquid contain only one single Pachner move, which we choose to be the one with only counter-clockwise tetrahedra in Eq. (196). Instead of the other Pachner moves, there are a number of simpler moves involving the additional tensor variables, from which the former can be derived. In the following, we give a selection of those moves in terms of cell complexes as well as in network notation. The remaining moves can be found in Appendix C. For the cell complexes we can only easily draw the 1-skeletons which do not in general unambiguously determine the cellulation. The network notation on the other hand is clear and completely unambiguous, but does not make the geometric interpretation apparent.

The moves can be divided into three groups. First, there are moves corresponding to symmetries of the tensors from which we can derive all other versions of the 2-3 Pachner moves. E.g., there is the (01)(23) *tetrahedron symmetry move* for the corresponding permutation of the tetrahedron vertices. This permutation changes the edge orientations of the (01) and (23) edge, which is done by using two pairs of flip hats. In the corresponding re-cellulation,

$$
\quad \leftrightarrow \quad \text{same 1-skeleton} \,,
\tag{206}
$$

both sides are glued from one tetrahedron and one flip hat from each of the two pairs. On the left, the flip hats are glued to the triangles 013 and 123, whereas on the right, they are glued to the triangles 102 and 032. In network notation, this is

$$
\tag{207}
$$

Also the flip hats have a symmetry, namely a $\pi$ rotation around the axis going through the "tip" of the hat and the centre of the 2-gon. This rotation changes the favourite edge of the 2-gon which can be undone by gluing a 2-gon flip to the 2-gon, as, e.g., in the *clockwise* 01 *flip hat rotation move*

$$
\quad \leftrightarrow \quad ,
\tag{208}
$$

in network notation

$$012 - \boxed{\phantom{x}}^{102}_{10} - 102 \quad \overset{01}{\diamond} = 012 - \boxed{\phantom{x}}^{012}_{01} - 102 \ . \tag{209}$$

The second group consists of cancellation moves, which allow us to derive the 4-1 Pachner moves from the 2-3 Pachner moves. E.g., gluing two flip hats at one triangle and one 2-gon yields the same pillow-like volume as in Eq. (197), which can be shrinked to a triangle, as in the *oriented* 01 *flip hat cancellation move*

$$\qquad\qquad \longleftrightarrow \qquad\qquad . \tag{210}$$

Again, the triangle is interpreted as a free bond in network notation

$$012b - \boxed{\phantom{x}}^{012}\ {}^{102}_{10}\ \blacksquare^{012} - 012f \ = \ 012b \longrightarrow 012f \ . \tag{211}$$

Similarly, the *tetrahedron cancellation move* equates two tetrahedra glued at two triangles on the left hand side with two flip hats glued at the 2-gon on the right hand side,

$$\qquad\qquad \longleftrightarrow \qquad\qquad , \tag{212}$$

$$\begin{array}{c} {}^{132}\ \diamond\ {}^{023} \\ {}^{013} \\ {}^{032}\ {}^{0132}\ \diamond\ {}^{0123}\ {}^{123} \end{array} = \begin{array}{c} 032 - \blacksquare^{023} - 023 \\ | \\ 132 - \blacksquare_{123} - 123 \end{array} . \tag{213}$$

The third group consists of moves relating the edge weights. In our case there is only one such move, which can be viewed as the definition of the triangle weight from the 2-gon weight

$$\qquad\qquad \longleftrightarrow \qquad\qquad . \tag{214}$$

In network notation, this is

$$b \overset{012}{\longrightarrow} f \ = \ b - \blacksquare^{102} \overset{\phantom{x}}{\underset{10\quad 01}{\diamond}} \boxed{\phantom{x}}^{012} - f \ . \tag{215}$$

The complete definition contains a few more moves, for which we refer to Appendix C.

Note that there is also a similar variant of the liquid without an orientation. The tensor variables are the same, just that we do not distinguish between clockwise and counter-clockwise versions. The moves are similar, just that there are also tetrahedron symmetry moves corresponding to reflections of the tetrahedron. E.g., there is a (01) tetrahedron symmetry move with only one single flip hat on each side.

### 8.1.3 Equivalence of the simplified and non-simplified liquid

The equivalence of the simplified liquid and the non-simplified liquids is shown via mappings from one to the other and vice versa. The mappings have to be weak inverses, as introduced in Section 4.4.

We first present the mapping from the non-simplified liquid to the simplified liquid. The triangle bond dimension $T$, the counter-clockwise tetrahedron, and the 01 edge weight are shared by both liquids and are accordingly mapped onto themselves. The clockwise tetrahedron of the non-simplified liquid can be constructed from the counter-clockwise tetrahedron and two flip hats

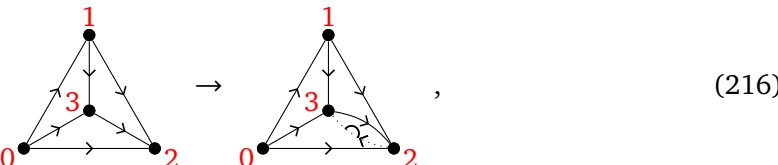

yielding the mapping

$$\text{(217)}$$

Likewise, the 02 and 12 edge weights of the non-simplified liquid can be constructed from the 01 edge weight and two flip hats. E.g., the 02 edge weight is obtained by

$$\text{(218)}$$

yielding the mapping

$$\text{(219)}$$

Let us quickly sketch how the mapped moves of the non-simplified liquid are derived by the moves of the simplified liquid. First note that using flip hat cancellation moves like Eq. (211), we can insert pairs of flip hats at triangles between tetrahedra. Then, using symmetry moves like Eq. (207) individual flip hats can be moved through tetrahedra between different faces adjacent to a fixed edge. Imagine introducing a pair of flip hats at a face adjacent to an inner edge, moving one of the flip hats once around that edge, and then removing the pair of flip hats. This flips the orientation of the inner edge. Via this and similar derivations, we can obtain all different 2-3 Pachner move from only the single move in Eq. (196). Moreover, consider the full tetrahedron cancellation move in Eq. (198) and observe that it can be derived from the tetrahedron cancellation move in Eq. (213) together with a flip hat cancellation move. With the aid of the just derived full tetrahedron cancellation move, we can bring one of the tetrahedra on the left hand side of the 2-3 Pachner move in Eq. (196) over to the right hand side, and obtain a 1-4 Pachner move. Again, we can use the tetrahedron symmetry moves and flip hat cancellation moves to derive all other versions of the 1-4 Pachner move.

Next, we consider the converse mapping from the simplified liquid to the non-simplified liquid. A 2-gon can be triangulated by a pair of triangles, and gluing two 2-gons can be replaced by gluing two triangle pairs instead

$$\text{(220)}$$

So we make the following identification between bond dimensions

$$D\big|_{01} := T\big|_{012} T\big|_{102} . \tag{221}$$

Next, we consider the mapping of the additional tensors. The clockwise 01 flip hat can be triangulated by two tetrahedra

$$ \to \qquad . \tag{222}$$

In terms of networks, we have

$$ \tag{223}$$

Every time we would glue two 2-gons of the simplified liquid, we now glue two triangle pairs instead. In doing so, the edges 02 and 12 edge in Eq. (220) (the edges 03 and 13 in Eq. (222)) become inner edges, so we have to add the corresponding edge weights. In general, we will include the edge weights of the 02 edge (which is the 03 edge in Eq. (222)) on the side with the clockwise 2-gon and the edge weight of the 12 edge on the side of the counter-clockwise 2-gon. The mapping of the counter-clockwise 01 flip hat is the similar – we just reverse the orientation and include the 13 edge weight instead of the 03 edge weight

$$ \tag{224}$$

The mapping of the 12 flip hats is defined analogously. At last, the 2-gon flip is mapped to two open bonds

$$ aa' \longrightarrow\!\!\blacklozenge\!\!\longrightarrow bb' := \qquad \tag{225}$$

and the 2-gon edge weight can be emulated by an edge weight for one of the two triangles

$$ aa' \longrightarrow\!\!\blacktriangleright\!\!\longrightarrow bb' := \qquad \tag{226}$$

All of the simplified moves correspond to re-triangulations, so they must be implied by the Pachner moves. A technical exception to this are moves involving the 2-gon flip and the cancellation moves, for which it is easy to find derivations. E.g., the 2-gon flip cancellation move in Eq. (343) of Appendix C simply becomes

$$ \tag{227}$$

Finally, we have to show that the mappings are weak inverses to each other. The 2-gon bond dimension of the simplified liquid under the double mapping yields twice the triangle

bond dimensions. Thus, we have to use the generalized notion of weak inverse introduced in Section 5.2.

Applied to the case of the twice-mapped 2-gon, we note that every 2-gon in a network is surrounded by a pair of flip hats and two such pairs of flip hats can never overlap in any network. Thus we call the network formed by the pair of flip hats *non-overlapping*. The non-overlapping network consisting of two flip hats has only triangle open indices, and is indeed equivalent to itself after mapping twice. The equivalence corresponds to a recellulation of two flip hats into four tetrahedra

$$
\leftrightarrow
\qquad . \tag{228}
$$

Thus, applying the above recellulation to all pairs of flip hats in a network is a sequence of moves relating a simplified-liquid network with the doubly-mapped simplified-liquid network. We conclude that the mappings are weak inverses of another and that the phases of the simplified and non-simplified liquid models are in one-to-one correspondence. In fact, even models themselves are in one-to-one correspondence up to a basis transformation acting on the 2-gon indices of the simplified liquid.

### 8.1.4 Hermiticity

If we want to impose Hermiticity, e.g., in order to allow for an interpretation of the liquid models in terms of a imaginary time evolution tensor network, we have to include a move that equates the clockwise tetrahedron and the complex conjugated counter-clockwise tetrahedron. The latter is not a tensor of our simplified liquid, but can be constructed via Eq. (217) as

$$
c - \boxed{\phantom{x}} - a = c - \blacksquare - a := b - \boxed{\phantom{x}} \blacktriangleright \blacksquare - a \quad . \tag{229}
$$

Also the 2-gon edge weight changes its orientation under complex conjugation

$$
b - \diamondsuit - a = b - \blacktriangleleft - a \quad . \tag{230}
$$

Note that we do not need to impose a Hermiticity move relating the flip hats and their orientation-reversed versions. This is because the flip hats always occur in pairs sharing a 2-gon, and the Hermiticity of each such pair can be derived from the moves above. Also, the Hermiticity move inverting the orientation of the triangle edge weight can be derived from the moves above.

### 8.1.5 Commuting-projector Hamiltonian

Let us briefly show how models of the present topological liquid yield commuting-projector models, formalized by a liquid mapping from the commuting-projector liquid to the topological liquid. A convenient layout for commuting-projector models are models on a regular triangular grid with one degree of freedom on each triangle. There is one Hamiltonian term on each vertex involving the six degrees of freedom at the surrounding triangles. So, the local ground

state projector is a tensor with 12 indices,

$$
\rightarrow
\tag{231}
$$

Commutativity of the projectors centered around neighbouring vertices yields three different moves, e.g.,

$$
=
\tag{232}
$$

Additionally, there is the projector move

$$
=
\tag{233}
$$

The mapping from this commuting-projector liquid to the topological liquid is as follows. A space-time given by stack of commuting-projector tensors can be transformed into a cellulation of a space-time volume by replacing each projector tensor with a "double-pyramid" cell. The latter is a volume whose boundary consists of an identical upper and lower part, both equal to the patch of six triangles above

$$
.
\tag{234}
$$

As depicted above this volume can be triangulated with six tetrahedra, all sharing the (67)-edge, yielding the liquid mapping

$$
:=
\tag{235}
$$

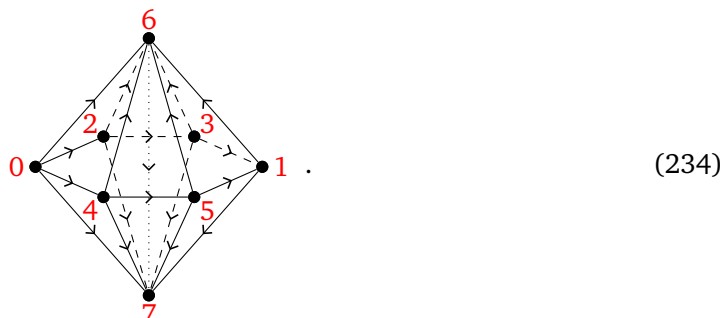

In the next section we will describe how models of the topological liquid can be considered blocked versions of Turaev-Viro state-sums. With this interpretation, Eq. (235) is nothing but a formal representation of the well-known relation between the latter state-sum, and the (suitably generalized) Levin-Wen string-net models.

### 8.1.6 Relation to the Turaev-Viro state-sum

Models of the liquid presented here are closely related to the *Turaev-Viro state-sum construction* [7, 8]. Whereas in the state-sum construction one starts with a *fusion category* and proves topological invariance from the properties of the latter, we take the opposite direction, and start from topological invariance to get to an algebraic structure similar to that of a fusion category. Bare fusion categories are *not* exactly the right structure needed for topological models and many versions of fusion categories with some additional structures exist in the literature. In Ref. [7], the input data of the state-sum construction is restricted to a specific class of examples, namely quantized enveloping algebras of $sl_2$. In Ref. [8], the state-sum construction is formulated for arbitrary *spherical* fusion categories. It is natural to assume that the model also works for multi-fusion categories with an adapted sphericality condition [35]. The string-net models in Ref. [5] are simplified further in order to be more accessible to the physics community and have additional restrictions such as a very strong notion of tetrahedral symmetry and vanishing Frobenius-Schur indicators. These restrictions render them incompatible with general twisted Dijkgraaf-Witten gauge theories [34, 36], but were partially removed for the Abelian case in Ref. [37].

In contrast, the algebraic structures we obtain in our approach are per construction the right ones to describe fixed-point models of topological phases with gappable boundary. Finding instances of our algebraic structures (i.e., models of liquids) is not fundamentally harder than finding instances of well-known algebraic structures such as fusion categories, as both are solutions to a set of polynomial equations. The only difference is, that for well-known structures there already exist a hand full of examples in the literature.

We now compare the liquids presented here to the Turaev-Viro state-sum models and show that they are equivalent up to technical details. Both constructions associate tensors to tetrahedra of a simplicial complex. The tensor of our liquid model has four indices associated to the faces of the tetrahedron, while the tensor in the Turaev-Viro construction is determined by the so-called *F-symbol* and the *quantum dimension d* of a *fusion category*. It has 10 indices, six of which are associated to the edges of the tetrahedron, and the remaining four to the faces, i.e.,

$$[F^{ab}_{cd}]^{i\alpha\beta}_{j\gamma\delta}(d_j)^{-1} \quad \rightarrow \quad \text{(diagram)} . \tag{236}$$

Just as in our liquid, if two tetrahedra are adjacent to the same face, the corresponding face indices of the tensors are contracted. However, the number $x$ of tetrahedra adjacent to a single edge can be more and less than 2, and we contract all the edge indices coming from those tetrahedra by an $x$-index delta tensor. Moreover, at each edge there is the vector $d$ containing the quantum dimensions which is connected to the corresponding delta tensor via another index. With these choices, we see that the *pentagon equation* for the $F$-symbol corresponds to invariance under the 2-3 Pachner move.

The $F$-symbol is not a tensor in the conventional sense, as one and the same face index can have different dimensions depending on the values $i, j, k$ of the indices at the surrounding edges. Those dimensions are collected into an object $N^{i,j}_k$ known as the *fusion rules*. $F$ can

be made into an ordinary tensor by fixing the dimension of the face indices to the maximal possible number in $N_k^{i,j}$, and filling up the new tensor entries with zeros. In the common examples (e.g., the toric code or the double Fibonacci model) all $N_k^{i,j}$ are either 0 or 1, so the face indices can be omitted (i.e., set to dimension 1) and $N_k^{i,j} \neq 0$ is interpreted as a constraint on the edge indices instead.

There are two ways to make the Turaev-Viro state-sum into a model of our liquid. The first is to reshape the tensor $F$ into a proper four-index tensor. To this end, we copy all edge indices using delta tensors, and block each copy with one of the adjacent face indices

$$\delta l j k \,\text{—}\, \boxed{\phantom{x}} \,\text{—}\, \beta d e f \;\;\overset{\gamma g h i}{\underset{\alpha a b c}{}}\; = \; \delta \,\cdots\; \beta \;. \tag{237}$$

Each of the new face indices is a composite of three of the old edge indices and one old face index. The dimension of this composite is fixed and given by

$$\sum_{i,j,k} N_k^{i,j} \;. \tag{238}$$

The second possibility is to interpret the $F$-symbol as a tensor of a different type, called *label-dependent tensors* [13]. The data determining such a tensor consists of a set of labels together with one array (of varying dimension) for each value of the labels. When we interpret the moves of the liquid using this tensor type, they turn into the equations of the Turaev-Viro model in their original form.

It is also possible to start from a (conventional) model of the topological liquid and arrive at a state-sum in the Turaev-Viro form in a natural way by using the fact that complex algebras (with a few special properties that we have in this case) can be block-diagonalized. For more details on this procedure we refer to Appendix D.

## 8.2 Face-edge liquid

In Section 3.2, we have encountered another way to represent 3-dimensional topological manifolds as a liquid, namely by associating tensors to faces and edges instead of volumes. In this section we look at this construction in more detail. Models of the resulting liquid are very similar to the *Kitaev quantum double model* [27] generalized to weak Hopf algebras [14, 15]. They are also similar to the *Kuperberg invariant* of 3-manifolds [38]. As in the sections above, the more general version of the liquid in Section 3.2 has edge orientations, which allow us to distinguish the indices of the face tensors. Dually, we add dual orientations to the faces, that is, a favourite adjacent volume.

### 8.2.1 Tensors and moves of the face-edge liquid

*Tensor Variables.* The tensors of the face-edge liquid are a collection of decorated face and edge tensors from which all other possible decorations can be generated. One possible choice is to use the 2-cells of the simplified $1 + 1$-dimensional liquid in Section 6 with the following orientations

- The clockwise triangle

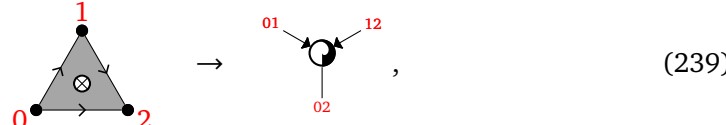

$$(239)$$

where the crossed circle in the middle of the triangle represents the dual orientation of the triangle pointing into the plane and we put an ingoing arrow to the indices corresponding to the two edges which are oriented clockwise when looking along the dual orientation.

- The clockwise and counter-clockwise cyclic 2-gon

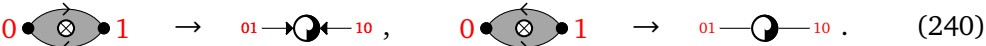

$$(240)$$

- The clockwise 3-valent edge

$$(241)$$

where the circle with the dot represents the dual orientation going out of the plane and the index corresponding to the face whose dual orientation is counter-clockwise when looking along the orientation is marked with an ingoing arrow. With this choice of ingoing/outgoing arrows, every bond in a network representing a piece of 3-manifold will be between one index with ingoing arrow, and one without.

- The 2-valent edge with clockwise and counter-clockwise dual orientations

$$(242)$$

In order to get models for a large class of topological phases (i.e., presumably all topological phase with a gappable boundary), we need to introduce one more additional structure in the network representation of cell complexes – the *corner weight*. A *corner* denotes a volume and an adjacent vertex. At every corner we find an alternating loop of edges and faces connected by bonds. We introduce a 2-index tensor called the *corner weight*

$$(243)$$

and require that at every corner a corner weight tensor is placed between exactly one edge-face pair. For example for the following corner enclosed by three faces and three edges a corner weight is located between the (13)-edge and the (123)-face

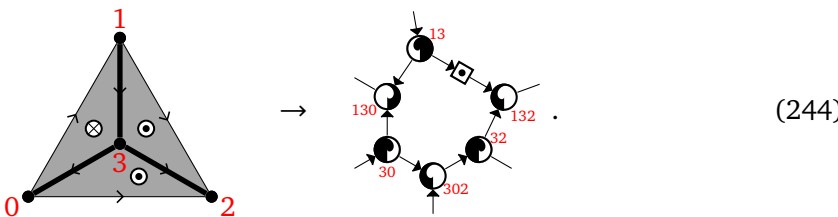

$$(244)$$

In fact, there are four different corner weight tensors, depending whether the orientation (dual orientation) of the edge (face) points towards or away from the vertex (volume) of the corner. However, the other three corner weights can be constructed from the one specific corner weight given above where both the orientation and dual orientation point towards the vertex and volume. E.g., the corner weight where the face is pointing away from the volume of the corner is obtained by inverting the dual orientation by conjugating with the 2-valent edge

$$a \rightarrow \!\!\!\bullet\!\!\!\leftarrow\!\!\square\!\!\leftarrow\!\!\!\bullet\!\!\!- b \ . \tag{245}$$

*Moves.* The moves of the face-edge liquid can partially be obtained from the moves of the oriented $1+1$-dimensional liquid from Section 6. Networks of the face-edge liquid consisting only of face tensors separated by 2-valent edges behave like networks representing a cellulation of a 2-manifold. Thus, it makes sense to take all moves of the $1+1$-dimensional liquid of Section 6 as moves for the face tensors of the face-edge liquid. More precisely, we have to take the version of the $1+1$-dimensional liquid with vertex weights from Appendix B, and the vertex weight is related to the corner weight of the face edge liquid: Every vertex in the two-dimensional cellulation corresponds to two corners in the three-dimensional cellulation, one with the volume above and one with the volume below. Thus, we identify the vertex weight of the $1+1$-dimensional liquid with two corner weights of the $2+1$-dimensional liquid

$$a \rightarrow \!\!\!\square\!- b := a \rightarrow \!\!\square\!\!\rightarrow\!\!\!\bullet\!\!\!\leftarrow\!\!\square\!\!\leftarrow\!\!\!\bullet\!\!\!- b \ . \tag{246}$$

The relation between the face-edge liquid and the $1+1$-dimensional liquid in Section 6 can be formalized as a liquid mapping from the latter to the former, which we refer to as the *2D embedding mapping*.

Dually, consider cellulations consisting only of edges that are all connected to the same 2 vertices separated by non-cyclic 2-gon faces. Also these behave like networks of the $1+1$-dimensional liquid alone, and so we also impose the moves of Section 6 for the edge tensors of the presented liquid. Analogously to the previous consideration the 2-dimensional vertex weight is now given by

$$a \rightarrow \!\!\!\square\!- b := a \rightarrow \!\!\square\!\!\rightarrow\!\!\!\bullet\!\!\!\leftarrow\!\!\square\!\!\leftarrow\!\!\!\bullet\!\!\!- b \ . \tag{247}$$

Again, the considerations above can be formalized as a mapping referred to as *dual 2D embedding mapping*.

In addition to the face-liquid moves mapped under the 2D embedding mapping and the dual 2D embedding mapping, we only need a few additional moves which relate face and edge tensors. The most important move is the *corner fusion move*, which we have already seen in Section 3.2 in a simplified form. Including orientations, dual orientations, and corner weights, it is given by

$$\tag{248}$$

which, in network notation, becomes

$$\tag{249}$$

Additionally, there are moves which effectively change the orientation of edges and dual edges. For example the dual orientation of a triangle can be changed via cyclic 2-valent edges

$$\longleftrightarrow \qquad . \tag{250}$$

As the counter-clockwise triangle is not a tensor, we have to construct it using a cyclic 2-gon. This yields a move

$$= \tag{251}$$

called the *dual orientation flip move*. Dually, we can flip a clockwise edge into a counter-clockwise edge by gluing cyclic 2-gons, yielding the *orientation flip move*

$$= \qquad . \tag{252}$$

At last, the Hermiticity moves are simply the $1 + 1$-dimensional Hermiticity moves from Section 6.4 mapped under the 2D embedding mapping and the dual 2D embedding mapping.

### 8.2.2 Relation to quantum double models

As mentioned in Section 3.2, the moves above are very similar to the bi-algebra axioms, and even more similar to the axioms of *weak Hopf algebras*. The latter (as any algebraic structure) define a liquid themselves. As such the two liquids are not exactly equivalent, in particular, because the weak Hopf liquid allows for a consistent flow of time, a feature missing in the face-edge liquid. There is only a liquid mapping from the weak Hopf liquid to the present topological liquid [31]. Thus, every model of the face-edge liquid defines a weak Hopf algebra, but not vice versa.

This suggests that models of the face-edge liquids are equivalent to Kitaev quantum doubles for weak Hopf algebras [15]. Indeed, using the commuting-projector mapping shown in the simplified form in Eq. (99), we find that the obtained Hamiltonians are equal. However, weak Hopf algebras are *not* precisely the right algebraic structure needed to obtain topological models. On the contrary the face-edge liquid yields topological models by construction. Comparing our formalism to the axioms of weak Hopf algebras, we see that the weak Hopf algebras in question need to fulfill a few additional properties. E.g., both the algebra and the co-algebra need to be (special) Frobenius (and *-algebras in the Hermitian case), and the antipode must be involutive. The need for technical details of this kind is apparent from our formalism, while it is not straight-forward to see in existing approaches to fixed-point models.

## 8.3 Equivalence of the face-edge and volume liquid

The volume liquid from Section 8.1 is "topological" due to the known fact that simplicial complexes with Pachner moves are a combinatorial analogue of (piece-wise linear) topological manifolds modulo homeomorphism in the continuum. For the face-edge liquid in Section 8.2, there is no such argument we can rely on. However, we can verify that the latter is topological by showing that it is equivalent to the volume liquid. Note that from our perspective, the

connection to continuum topology is merely a guiding intuition and all that matters is that the liquid defines a sensible notion of deformability to which physical models can be extended.

In this section, we present two weakly inverse mappings between the volume and the face-edge liquids, sketch why they are well-defined and why they are indeed weak inverses of each other. The mapping from the face-edge liquid to the volume liquid has a geometric interpretation. It can be seen as refining a cellulation such that each volume of the refined cellulation corresponds to either an edge or a face of the original cellulation. The mapping is given by the following prescription.

- Every face is replaced by a "double pyramid", that is, we add one vertex $x$ "above" and one vertex $y$ "below" the face, and connect all vertices with $x$ and $y$. For edges with counter-clockwise orientation the corresponding edge of the double pyramid carries an edge weight. E.g., for the clockwise triangle, an edge weight is associated to the 02-edge

$$\tag{253}$$

This volume can be triangulated by two tetrahedra. In network notation we have

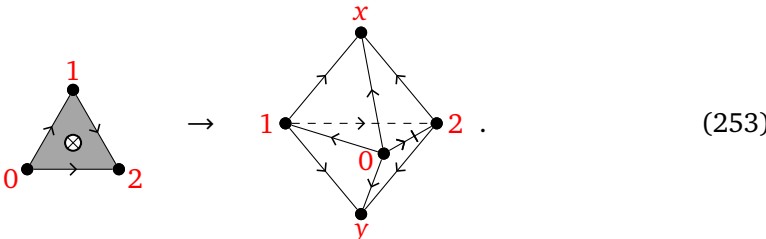

$$\tag{254}$$

The clockwise 2-gon is mapped to a volume which can be glued from two flip hats

$$\tag{255}$$

with the following network notation

$$\tag{256}$$

The counter-clockwise 2-gon is defined analogously, just that edge weights are included for both the 01- and the 10-edge.

- Every edge is replaced by a volume constructed as follows. The two vertices adjacent to the edge ($x$ and $y$ in the figure below) are connected by edges that replace the adjacent faces ($a, b, c$ in the figure) and edge weights are associated to all edges for which the

corresponding faces have clockwise dual orientation. An additional vertex is added between each pair of edges and connected to the vertices. For the 3-valent edge we obtain

$$\tag{257}$$

The volume above has the following triangulation in network notation

$$\tag{258}$$

With a bit of geometric imagination one can verify that all volumes from the two points above fit together and form a "refining" without any holes and overlaps. If an edge is adjacent to a face in the original cellation, the two corresponding volumes after refining share a pair of triangles. Thus, the face-edge bond dimension is mapped to two triangle bond dimensions, that is, every index in a face-edge network corresponds to a pair of indices in a triangle network,

$$| := \begin{matrix} T & T \\ | & | \end{matrix} . \tag{259}$$

At last we check that the edge weights of the refined cellation are distributed correctly over the tensors of the face-edge liquid. The edge weights of edges that separate pairs of triangles constituting composite indices are already included into the triangle and face tensors. They contain the weight if the second triangle in the pair has two clockwise edges. The other edges correspond to corners of the original triangulation and thus those edge weights are mapped to corner weights

$$\tag{260}$$

In order to prove that the above recipe defines a liquid mapping, we would have to give derivations for all the mapped moves. This is a straight-forward and purely combinatorial procedure. However, it is quite tedious and lengthy, thus we only give a quick argument why the mapping is well-defined: The mapping is constructed such that all mapped moves are retriangulations. As it is known that any retriangulation corresponds to a sequence of Pachner moves, it is clear that all mapped moves can be derived.

The mapping from the volume liquid to the face-edge liquid also has a geometric intuition in terms of a refining, such that every edge and face of the refined cell complex can be unambiguously associated to a volume of the original cell complex. To this end, we first split each triangle into two triangles separated by a pillow-like volume, such that every $n$-valent edge becomes $2n$-valent. Then, we replace every such $2n$-valent edge into $n$ 4-valent edges which are cyclically connected by $n$ trivial (non-cyclic) 2-gons. Like this, each original volume turns into one face for each of its faces, and one 4-valent edge for each of its edges.

Applying this to the tetrahedron we get a network consisting of 4 triangles and 6 4-valent edges. As we will see below, this network is equivalent to a simpler one which has one face (the 012 face below) missing and the adjacent edges being only 3-valent. As the counter-clockwise triangle is not explicitly part of our liquid, we have to construct it using the cyclic 2-gon

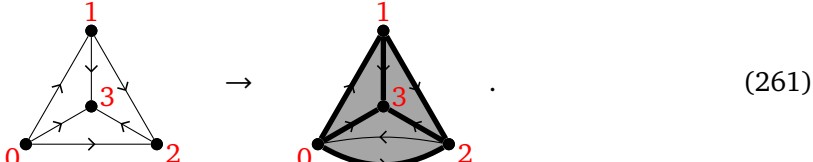

(261)

In network notation this is

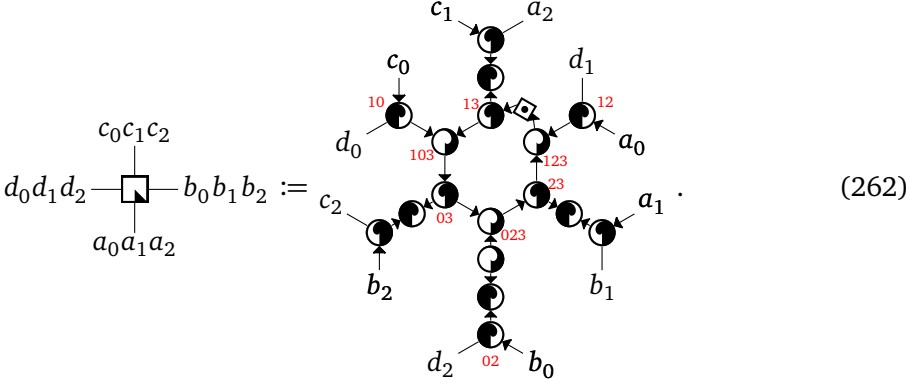

(262)

Regarding the bond dimension variables, we note that a triangle has three edges and thus the triangle bond dimension is mapped to three times the face-edge bond dimension. Therefore in the above equation one index on the left corresponds to three indices on the right,

$$\overset{T}{|} := | \quad | \quad |.$$

(263)

Let us sketch how the mapped moves can be derived from the moves of the face-edge liquid. The edge orientations and dual face orientations can be changed arbitrarily by inserting/moving around cyclic 2-gons and 2-valent edges using the 2-gon cancellation move in Eq. (172) and triangle symmetry move in Eq. (169) for either the edge or face tensors. So, for simplicity, we will neglect those orientations in the following considerations and focus on the derivation of the 2-3 Pachner move. We work with the geometric intuition that tensors are associated to the triangles and edges of a 3-manifold triangulation. Internally, $n$-valent edges have to be decomposed into 3-valent edges. However, the different decompositions are all equivalent using the (dually mapped) 2-dimensional moves, and we assume that these moves are applied implicitly. For the remaining considerations it is convenient to introduce some terminology.

- The corner fusion move depicted in Eq. (248) from left to right is denoted by $C(012|02)$,

- the 2-2 Pachner move, as depicted in Eq. (115), from left to right, by $P_2(012|023)$,

- and the following move

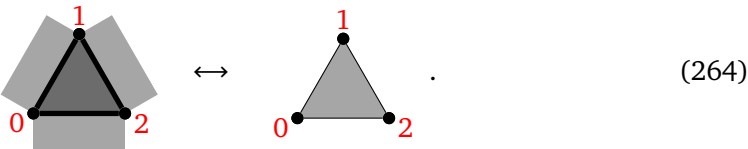

(264)

which replaces a single triangle by two duplicates separated by a pillow-like volume by $T(012)$.

The last move is derived by using the triangle cancellation move to bring one triangle in the corner fusion move in Eq. (248) from the right to the left.

Next, we consider two variants of tetrahedra that are relevant in the 2-3 Pachner move and apply a sequence of the moves above in order to remove several of their faces.

- For a tetrahedron with 3-valent edges where in the mapped network all four triangles and all six edges are represented by tensors

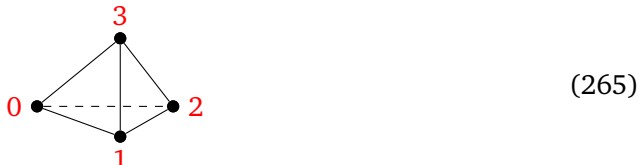

$$\tag{265}$$

we can remove the face 012, such that only the triangles 013, 123, 023, and the edges 03, 13, 23, are represented by tensors. This can be done by the sequence of moves

$$\overline{T(123)} \to \overline{C(123|12)} \to P_2(012|123)$$
$$\to C(013|03) \to T(023) \,, \tag{266}$$

where the bars denote the move in the opposite direction.

- Start with a tetrahedron where all faces and edges are represented by triangle tensors and 3-valent edge tensors, except for the edge 12 which is a trivial 2-valent edge and thus not represented by an tensor. We can remove both the 012- and the 123-face, such that only the triangles 013, 023, and the edge 03 are represented by 3-index tensors. This can be done by the following sequence of moves

$$P_2(012|123) \to C(013|03) \to T(023) \,. \tag{267}$$

Note that it is precisely the move derived in Eq. (266) which allows us to add/remove the 012 face on the right hand side of the tetrahedron mapping in Eq. (262). If we now apply the face-edge mapping to the 2-3 Pachner move, each triangle will be doubled (taking the version of the mapping including the 012 triangle). We can apply the move $T$ in Eq. (264) to reduce each triangle pair to a single triangle. Next, we can apply the moves derived in Eq. (266) and (267) to remove all interior triangles and edges on the left and right, which yields an equation between twice the same network. Applying this procedure in the opposite direction, we have found a derivation of the mapped 2-3 Pachner move from the moves of the face-edge liquid.

We still have to show that the two mappings are weak inverses to each other. The mappings change the bond dimension variables, such that applying both mappings maps every open index to six open indices. Thus, we have to use the generalized notion of weak inverse first mentioned in from Section 5.2. We won't explicitly show that the two mappings applied in sequence (in both orders) are equivalent to via moves acting on non-overlapping patches. However, it is easy to see that the composition of the two mappings defines a topology-preserving refinement of the cellulation, which can be undone by moves.

# 9 Fermions

In this section we will demonstrate how fixed-point models with fermionic degrees of freedom can be formalized as liquid models. In the first part we will introduce fermionic tensors, the tensor type which is the domain of fermionic liquid models. In the second part we will discuss the kind of liquid that fermionic systems typically extend to, namely combinatorial representations of spin manifolds. In the third part, we will illustrate the formalism in $1 + 1$ dimensions, by giving a liquid which has a model corresponding to the Kitaev chain.

## 9.1 Fermionic tensors

In Section 2.1, we have demonstrated how quantum spin systems can be formulated in terms of tensor networks. In many condensed matter models we also have fermionic degrees of freedom. We could just use a Jordan-Wigner transformation to write the fermionic system as a spin system. However, such a transformation is generally non-local, and even in $1+1$ dimensions where it is local in principle, it changes the homogeneity of the model. That is, a translation-invariant fermionic system (with periodic boundary conditions) does not translate into a translation-invariant spin system/tensor network.

We can still write fermionic systems as tensor networks. However, the "tensors" cannot be just arrays, as for spin systems, but have to take the canonical anti-commutation relations for fermions into account. A fermionic operator acting on $n$ modes labelled $0, \ldots, n-1$, can be expanded as

$$
\sum_{\substack{s_0, \ldots, s_{n-1} \\ s'_0, \ldots, s'_{n-1}}} A^{s_0, \ldots, s_{n-1}}_{s'_0, \ldots, s'_{n-1}}
$$

$$
(c_0^\dagger)^{s_0} \cdots (c_{n-1}^\dagger)^{s_{n-1}} |0\rangle \langle 0| (c_y)^{s'_{n-1}} \cdots (c_0)^{s'_0} ,
\tag{268}
$$

where the $s_i$ and $s'_i$ are either 0 or 1 depending on whether the fermionic degree of freedom is occupied or not. We observe the following.

- The operator must preserve fermion parity. That is, $A$ can have non-zero entries only when

$$
\sum_i s_i + \sum_i s'_i = 0 ,
\tag{269}
$$

  where the summation is understood mod 2.

- In order to specify the operator, we have to both specify $A$, but also the ordering of creation and annihilation operators, in the case above $0', \ldots, n-1', n-1, \ldots, 0$. The same fermionic operator may also be written down with any other ordering, just that then also the coefficients $A$ change. E.g., if we exchange 0 and 1 at the end of the ordering above, the anti-commutation of $c_0$ and $c_1$ tells us that we have to modify $A$ by

$$
(A')^{s_0, \ldots, s_{n-1}}_{s'_0, \ldots, s'_{n-1}} = A^{s_0, \ldots, s_{n-1}}_{s'_0, \ldots, s'_{n-1}} \quad (-1)^{s_0 s_1} .
\tag{270}
$$

More generally, we can consider degrees of freedom with $i$ configurations without a fermionic charge, and $j$ configurations with a fermionic charge, instead of only having only one charge-free (non-occupied) and one charged (occupied) configuration. This motivates the following definitions:

A *fermionic tensor* is an equivalence class of pairs $(A, O)$, where $A$ is an array, and $O$ is an ordering of its indices (compare also Ref. [39]). The $i+j$ configurations of each index of $A$ are divided into $i$ *even* configurations, writing $|x| = 0 \in \mathbb{Z}_2$ for $0 \le x < i$, and $j$ *odd* configurations, writing $|j| = 1 \in \mathbb{Z}_2$ for $i \le x < i+j$. $A$ has to have even parity, that is

$$
A_{i,j,\ldots} = 0 \quad \text{if} \quad |i| + |j| + \ldots \ne 0 .
\tag{271}
$$

Two pairs $(A, O)$ and $(A', O')$ are equivalent if $O'$ and $O$ are related by a transposition of two consecutive indices $x$ and $y$, and $A$ and $A'$ are related as

$$
O' = \tau_{xy}(O) ,
$$

$$
(A')_{s_0, s_1, \ldots} = A_{s_0, s_1, \ldots} (-1)^{|s_x||s_y|} .
\tag{272}
$$

A conventional fermionic operator acting on $n$ modes can be represented by a fermionic tensor with $2n$ indices, each with only one even and one odd configuration.

The tensor product of two fermionic tensors is the tensor product of arrays, together with the concatenation of orderings:

$$(A_1, O_1) \otimes (A_2, O_2) = (A_1 \otimes A_2, O_1 \cap O_2) \,. \tag{273}$$

The order in which we concatenate $O_1$ and $O_2$ does not matter, as the minus signs collected from exchanging all indices of $A_1$ with all indices of $A_2$ is trivial due to the even parity constraint in Eq. (271). The contraction of two indices $x$ and $y$ of a fermionic tensor $(A, O)$ consists of the following steps.

- Go to a representative where $y$ comes right after $x$ in $O$.

- Contract $x$ and $y$ in $A$.

- Remove $x$ and $y$ from $O$.

Roughly, the philosophy of this work is that, once we have chosen a particular liquid, we can interpret the same equations in terms of fermionic tensors to obtain fermionic fixed point models. However, fermionic tensors do not obey exactly the same graphical calculus as array tensors. There are two small differences.

The first difference is that contracting indices $x$ and $y$ for a fermionic tensor is different from contracting $y$ and $x$, as we explicitly specified that $y$ is *after* $x$ in $O$. If we instead wanted $y$ to be *before* $x$, we would have to exchange them yielding a factor of $(-1)^{|s_x|}$ in $A$ before the contraction, and hence a different result. The ordering in the contraction can be incorporated into the graphical calculus by associating a *bond direction* to each bond in a network, represented by an arrow, e.g.,

$$\to \!\!\!\!\!\!- \!\!A\!\!-\!\!\!\!\!\!\square\!\!- \,. \tag{274}$$

Note that also the open indices have a bond direction. An inwards pointing bond direction means we multiply by the fermion parity $(-1)^{|x|}$, where $x$ is the configuration of the open index. An outwards bond direction is assumed by default.

The second difference is that contracting two index pairs $a, a'$ and $b, b'$ one after the other is different from blocking them into a single index pair $ab, a'b'$ that is contracted. In order to perform the former contractions, we would have to order the indices like $aa'bb'$, which differs from $aba'b'$ by a sign of $(-1)^{|b||a'|}$. This can be fixed by dividing the indices into *input indices* and *output indices*, only allowing contractions between one input and one output index, and choosing the opposite order for the index parts when we block input indices versus output indices. In this case, the input/output structure can also be used to choose a canonical bond direction from input to output.

## 9.2 Liquids with spin structure

It would be very much in the spirit of this work to take, e.g., the liquid in Section 6, look for models in fermionic tensors, and interpret them as fixed point models for fermionic topological phases in $1+1$ dimensions. If we have an orientation, the input/output structure can be chosen according to whether the corresponding triangle edge is clockwise or counter-clockwise. However, it turns out that the correct choice of contraction directions is a bit involved.

In Section 6, we have seen that if we want to consider fixed-point models with *complex* tensors describing ordinary quantum spin systems, the latter should be defined on *oriented* manifolds and we should impose a relation between the orientation and complex conjugation, namely Hermiticity. If we use *fermionic tensors* for fixed-point models of fermionic quantum

matter, the situation is similiar: These models should be defined on *spin* manifolds and we should impose a close relation between the *spin structure* and the *fermion parity*. What results might be regarded a fixed-point lattice version of the *spin-statistics relation* known in quantum field theory.

In order to get a liquid whose networks represent spin manifolds, we need a combinatorial representation of a *spin structure* [10]. A spin structure is a $\mathbb{Z}_2$-valued 1-cochain $\eta$, whose boundary is the second *Stiefel-Whitney class*, represented by a 2-cocycle $\omega_2$. In an $n$-dimensional simplicial complex, $\eta$ can be represented as a subset (or $\mathbb{Z}_2$ colouring) of $n-1$-simplices, and $\omega_2$ as a subset (or colouring) of $n-2$-simplices. The boundary relation is obvious: The $\mathbb{Z}_2$-colour of an $n-2$-simplex in the boundary of $\eta$ is the sum of $\mathbb{Z}_2$-colours of adjacent $n-1$-simplices. A formula for computing $\omega_2$ in terms of the combinatorics of a simplicial complex is given in Ref. [40].

The relation between the combinatorial spin structure and the fermionic tensor network is simple: At every fermionic bond crossing a $n-1$-simplex of the spin structure, we have to insert the $(-1)^{P_f}$ operator. Equivalently, we reverse the bond direction, which is otherwise chosen to point towards the left relative to the branching-structure direction of the edge and the underlying orientation.

## 9.3 The liquid in $1+1$ dimensions

In this section we describe a topological liquid with spin structure in $1+1$ dimensions, and discuss its models using fermionic tensors.

### 9.3.1 Spin structures in $1+1$ dimensions

In $1+1$ dimensions, $\omega_2$ is a $\mathbb{Z}_2$-colouring of vertices, and the formula for the colouring of a vertex $v$ in a simplicial complex is given by

$$\omega_2(v) = 1 + \#E_0(v) + \#T_0(v) \pmod 2, \tag{275}$$

where $\#E_0(v)$ is the number of edges starting in $v$, and $\#T_0(v)$ is the number of triangles which have $v$ as their 0th vertex (when numbering them according to the branching structure).

$\eta$ is a collection of edges, which form a pattern of lines whose (modulo 2) endpoints are $\omega$, and which are closed otherwise. Consider, e.g., the following patch of triangulation with a combinatorial spin structure $\eta$

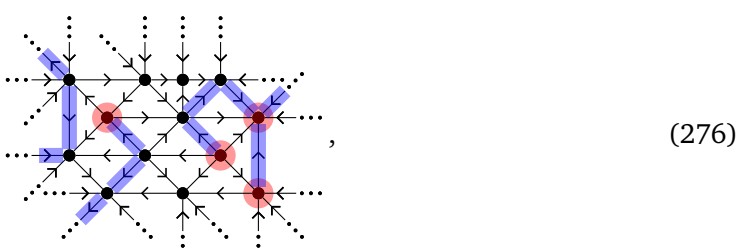

$$\tag{276}$$

where the $\omega_2$-vertices were marked red, and the $\eta$-edges were marked blue.

While $\omega_2$ is fixed, there are many possible choices of $\eta$. The precise choice is irrelevant to a large degree, as different choices are considered equivalent if they are related by *homology moves*. A homology move changes $\eta$ by adding the boundary of a triangle (modulo 2), e.g.,

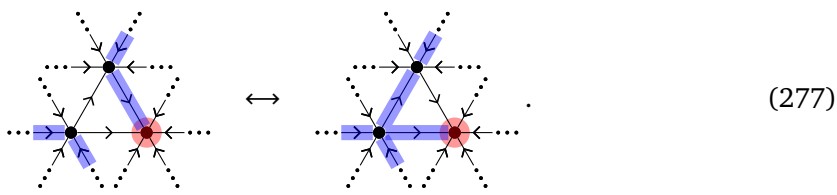

$$\tag{277}$$

Equivalence classes of $\eta$-triangulations under homology moves (as well as Pachner moves) are in one to one correspondence with spin manifolds. If the manifold is *spinnable*, there are as many equivalence classes as there are first homology classes of the manifold, though there is no canonical identification between the two. Otherwise, there are none. In $1+1$ dimensions, all oriented manifolds are spinnable. Of course, the spin structure also has to be incorporated into the Pachner moves. As the latter exchange a disk with a disk, and a disk has trivial 1-homology, all different ways of adding $\eta$ to a Pachner move are equivalent.

### 9.3.2 The liquid

A triangulation with orientation and $\eta$-chain is represented by a liquid (with bond directions) in the following way: As in the non-spin case, there are tensors for the clockwise and counter-clockwise triangles. The bond direction at an edge which is not part of $\eta$ is towards the left when looking along the branching-structure orientation of that edge (in order to know what "left" means we need the underlying global orientation). At an edge which is part of $\eta$, it is the other way round. With this encoding, the equations corresponding to homology moves are automatically fulfilled by any model, as simultaneously flipping all bond directions around a fixed tensor does not change anything due to the global even-parity constraint fulfilled by any fermionic tensor.

   As in previous sections, we are looking for a simplified liquid, whose networks can be interpreted as cellulations with other types of faces. In order to do the same for $\eta$-triangulations, we have to think about how to equip arbitrary cellulations with spin structure.

   The generalisation of chains and boundaries to arbitrary cellulations is obvious. A generalised rule for $\omega_2$ is the following: For every type of face, we have to specify one *special corner*, and we replace $\#T_0(v)$ in Eq. (275) by the number of adjacent faces for which $v$ is in the special corner. We denote the special corner by a small angle, e.g., the special corner of the branching-structure triangle is the 0-vertex

$$\qquad \qquad \qquad \qquad \qquad \qquad \qquad \qquad \qquad \qquad \qquad \qquad \qquad \qquad \qquad \qquad \tag{278}$$

Consider the sum of the $\omega_2$ colourings of all vertices of a triangulation of a manifold with boundary. Every vertex, every edge and every triangle contributes exactly 1 to this sum, so we see that we obtain the *Euler characteristic* (modulo 2) of the manifold. If we want to use a different type of face, we have to define it via a $\eta$-triangulation. As the Euler characteristic (modulo 2) of a disk is 1, $\eta$ will always have an odd number of "open ends" at the boundary of the new face. Without loss of generality, we can choose one single open end at the special corner.

   The simplified liquid is very similar to the non-spin case in Section 6, just that we have to include bond directions determined by the spin structure. The tensors and their interpretations as faces (with special corners) are the following.

- The clockwise triangle

$$\qquad \qquad \qquad \qquad \qquad \qquad \qquad \qquad \qquad \qquad \qquad \qquad \qquad \qquad \qquad \qquad \tag{279}$$

- The clockwise cyclic 2-gon. Note that this tensor looses its rotation symmetry due to the

spin structure (it gets replaces by a modified symmetry move though, see Eq. (291)),

$$\qquad\qquad\qquad \rightarrow \qquad \qquad . \qquad\qquad\qquad (280)$$

- The counter-clockwise cyclic 2-gon

$$\qquad\qquad\qquad \rightarrow \qquad \qquad . \qquad\qquad\qquad (281)$$

The moves are the same as in Section 6, just that we have to add a choice of $\eta$ on the left and right.

- The *spin 2-2 Pachner move*

$$\qquad\qquad\qquad \leftrightarrow \qquad\qquad . \qquad\qquad\qquad (282)$$

This move does not change $\omega_2$, so we can choose $\eta$ to be trivial. In network notation, we get

$$\qquad\qquad\qquad = \qquad\qquad . \qquad\qquad\qquad (283)$$

- The *spin triangle cancellation move*

$$\qquad\qquad\qquad \leftrightarrow \qquad\qquad . \qquad\qquad\qquad (284)$$

This move adds/removes the interior vertex 0 with odd $\omega_2$-colour, and changes the $\omega_2$-colour of the boundary vertex 0. The minimal choice of $\eta$ which corrects this, is the 10 edge on the left. In network notation,

$$\qquad\qquad\qquad = \qquad\qquad , \qquad\qquad\qquad (285)$$

the bond direction corresponding to that edge is reversed.

- The *spin (012) triangle symmetry move*

$$\qquad\qquad\qquad \leftrightarrow \qquad\qquad . \qquad\qquad\qquad (286)$$

Again, nothing changes with $\omega_2$, so $\eta$ can be chosen empty. In network notation, again, we find

$$\text{(287)}$$

- The *spin 2-gon cancellation move*

$$\text{(288)}$$

Here we chose the position of the special corners such that again $\omega_2$ does not change. In network notation, this is

$$a \longrightarrow\!\!\!\triangleright\!\!\rightarrow\!\!\triangleleft\!\!\longleftarrow b = a \longrightarrow\!\!\longleftarrow b \tag{289}$$

The special corner of the spin 2-gon spoils its index permutation symmetry,

$$\text{(290)}$$

As the position of the special corner changes, $\omega_2$ changes for both vertices 0 and 1, so we have to add an $\eta$-edge between them on one side. In network notation, we have to add an inwards arrow to one of the open indices

$$a \longrightarrow\!\!\!\triangleright\!\!\longrightarrow b = a \longrightarrow\!\!\rightarrow\!\!\triangleleft\!\!\longleftarrow b \ . \tag{291}$$

Note that, analogous to the non-spin case in Section 6, this symmetry move is derived directly from the spin triangle cancellation move in Eq. (285). From there, the analogous move for the counter-clockwise 2-gon,

$$a \longrightarrow\!\!\!\triangleright\!\!\longrightarrow b = a \longrightarrow\!\!\rightarrow\!\!\triangleleft\!\!\longleftarrow b \ , \tag{292}$$

is derived via the 2-gon cancellation move in Eq. 289.

### 9.3.3 Hermiticity

The Hermiticity condition is another point where it appears natural to distinguish between particle and hole sectors. If we express a fermionic Hamiltonian as an ordinary operator in Fock space, we would expect this operator to be Hermitian, e.g.,

$$\text{(293)}$$

where $H$ is the array tensor representing a fermionic tensor with respect to the index ordering $abb'a'$. However, if we read the above equation for $H$ as a fermionic tensor, then the index orderings on the two sides are reverse to each other, giving a reordering sign of

$$(-1)^{|a|(|b|+|b'|+|a'|)+|b|(|a'|+|b'|)+|b'||a'|} = (-1)^{|a|+|b|+|a||b|+|a'||b'|} \ . \tag{294}$$

Thus, the Hermiticity condition for a fermionic tensor should equate the orientation-reversed tensor with its complex conjugate where we also reverse the index ordering. This operation is compatible with contraction and tensor product since it also exchanges input and output indices.

Fermionic liquid models should be invariant under exactly this Hermiticity operation. That is, in our triangle-liquid example, the counter-clockwise triangle tensor should be obtained from the clockwise triangle tensor by complex conjugation and inversion of index ordering.

### 9.3.4 Models

Let us look for models of the liquid using fermionic tensors which fulfil both the spin-statistics relation and the fermionic Hermiticity relation. To this end, we choose a fixed index ordering for each tensor, and see what the equations mean for the array tensors representing the fermionic tensors with this ordering. Then we look at what minus signs we pick up from re-ordering the indices on both sides of the equation in order to perform the contractions, and equate the two sides. A choice of orderings that turns out to be particularly convenient is

$$
\begin{array}{ccc}
\underset{0}{\overset{1\diagdown\ \diagup 2}{\triangle}} \ , & 0 \mathbin{-\!\!\triangleright\!\!-} 1 \ , & 0 \mathbin{-\!\!\triangleright\!\!-} 1 \ .
\end{array}
\tag{295}
$$

In order to compute the reordering sign appearing in the fermionic liquid moves, we can proceed as follows. We start by concatenating the index sequences of the individual tensor copies in an arbitrary order. Then, we use index transpositions to move indices that are to be contracted next to each other, and then remove them. We record the minus signs collected when we perform the index transpositions on the way. We do this for both sides of an equation. In the end, we move the indices, such that the orderings on each side of the equation are equal. Of course, we can also cancel reordering signs on both sides.

In the following, we use a short-hand notation for sign calculations in contractions. E.g., $(bc + ab)|x'abdxc|(xx')$ denotes an intermediate step in the computation of the reordering sign, with index ordering $x'abdxc$, where we still need to contract $x$ and $x'$ (in that order), and we already collected a sign of $(-1)^{|b||c|+|a||b|}$. For the spin 2-2 Pachner move we get the following

$$
\begin{aligned}
|x'abdxc|(xx') &= |dayy'bc|(yy') \, , \\
(dx)|abdc| &= |dabc| \, , \\
(dx + cd)|abcd| &= (d)|abcd| \, , \\
|abcd| &= |abcd| \, .
\end{aligned}
\tag{296}
$$

We find that all the reordering signs on the left and right cancel. The other reordering signs are computed in Appendix E. Interestingly, also all other signs cancel. Note that this would not have been the case without the spin structure modification. Vanishing of the reordering sign is not a general property of fermionic liquid models, though. First of all, we would have obtained non-trivial reordering signs, if we had chosen a different index ordering in Eq. (295). Second, the fact that we can find an index ordering for which the reordering signs vanish seems to be specific to the $1 + 1$-dimensional case, and we do not find the same to be true in dimensions $2 + 1$ or higher.

The vanishing of the reordering signs implies that the models of the liquid in fermionic tensors are in one-to-one correspondence to the models in array tensors which have a $\mathbb{Z}_2$-grading. Note that the latter are agnostic of the bond directions, and the liquid we get after dropping those is equal to its non-spin analogue in Section 6.3. A fixed array tensor model might allow

for different inequivalent $\mathbb{Z}_2$-gradings, corresponding to different fermionic models. Technically, there always exists a grading by considering every configuration as even. Those models are trivial though, in the sense that they do not have any fermionic charges.

We point out that despite there being a one-to-one correspondence, fermionic spin-topological and $\mathbb{Z}_2$-graded topological models are still different models. In particular, the one-to-one correspondence will break down when we add other (non-topological) moves, such as invertibility, or commutativity.

### 9.3.5 Kitaev chain

In this section we consider the simplest fermionic model describing the only non-trivial phase of the spin-structure triangle liquid, which turns out to be equivalent to the Kitaev chain. The $\mathbb{Z}_2$-graded algebra that it is based on is probably the simplest one one can think of, namely the group algebra of $\mathbb{Z}_2$ itself. Written as arrays for the fixed index ordering in Eq. 295, the tensors of the model are given by

$$
\begin{array}{c}
{}^{b}\diagdown\diagup^{a} \\
\uparrow \\
c
\end{array}
= \frac{1}{\sqrt{2}} \cdot \delta_{a+b,c} = \frac{1}{\sqrt{2}} \left( \begin{pmatrix} 1 & 0 \\ 0 & 1 \end{pmatrix} \begin{pmatrix} 0 & 1 \\ 1 & 0 \end{pmatrix} \right),
$$

$$
a \blacktriangleright b = \delta_{a,b} = \begin{pmatrix} 1 & 0 \\ 0 & 1 \end{pmatrix}, \tag{297}
$$

$$
a \blacktriangleright b = \delta_{a,b} = \begin{pmatrix} 1 & 0 \\ 0 & 1 \end{pmatrix}.
$$

Here, $a$, $b$ and $c$ are understood as elements of $\mathbb{Z}_2$ and in the expressions for the triangle tensor $a$ and $b$ label rows and columns, while $c = 0, 1$ refers to the first and second matrix, respectively. It can be easily seen that the model is also Hermitian: All tensors are real, only supported in the particle sector (by construction, as we used plain fermionic tensors), and invariant under orientation reversal. Thus, the model is invariant under $K$, $R$, and orientation reversal separately, and certainly under all three operations together.

The *Kitaev chain* [41] to which this model is equivalent, is a fermionic chain with a nearest-neighbour Hamiltonian of Majorana fermionic operators

$$
H = -\sum_i (c_i + c_i^\dagger)(c_{i+1} - c_{i+1}^\dagger). \tag{298}
$$

It is a commuting-projector model, with the projector given by

$$
\begin{aligned}
P &= \frac{1}{2}(\mathbb{1} + (c_0 + c_0^\dagger)(c_1 - c_1^\dagger)) \\
&= \frac{1}{2}\left( |0\rangle\langle 0| + c_0^\dagger |0\rangle\langle 0| c_0 + c_1^\dagger |0\rangle\langle 0| c_1 + c_0^\dagger c_1^\dagger c_1 c_0 - c_1 c_0 + c_1^\dagger c_0 + c_0^\dagger c_1 - c_0^\dagger c_1^\dagger \right).
\end{aligned} \tag{299}
$$

Applying the expansion in Eq. (268) yields

$$
A^{s_0 s_1}_{s_0' s_1'} = \frac{1}{2} \delta_{s_0 + s_1, s_0' + s_1'} (-1)^{s_0 s_1 + s_0' s_1'}. \tag{300}
$$

$A$ becomes a fermionic tensor with index ordering $s_0' s_1' s_1 s_0$.

In order to compare this commuting-projector model with our liquid model, we use a liquid mapping identifying a projector with a rhombus-like cell of space-time, similar to the construction in Section 4.5.3,

$$
\begin{array}{ccc}
\begin{array}{c} 1 \\ 0 \diamond 2 \\ 3 \end{array} & \rightarrow & \begin{array}{c} 1 \\ 0 \lozenge 2 \\ 3 \end{array}
\end{array} . \tag{301}
$$

The shown cellulation of the rhombus yields the mapping

$$
\tag{302}
$$

In order to evaluate the network on the right-hand side, we first compute the reordering sign we get when we bring the indices into the ordering $dcba$, starting from the orderings in Eq. (295), to find

$$
|daxy'x'cyb|(xx')(yy') = x|day'cyb|(yy') = x + yb|dacb| \\
= x + yb + a(c+b)|dacb| = dc + ab|abcd| \, .
\tag{303}
$$

So for the chosen index ordering, the array representing the fermionic tensor is given by

$$
\sum_x (\frac{1}{\sqrt{2}}\delta_{d+x,a})(\frac{1}{\sqrt{2}}\delta_{c+x,b})(-1)^{dc+ab} = \frac{1}{2}\delta_{a+b,c+d}(-1)^{dc+ab} \, ,
\tag{304}
$$

which exactly equals the array in Eq. (300).

## 10 Topological order in $3+1$ dimensions

In this section, we sketch two liquids for topological order in $3+1$ dimensions. One is a very straight-forward liquid based on *simplicial complexes*, and the other one is analogous to the face-edge liquid in $2+1$ dimensions, with volumes and faces being represented by tensors.

### 10.1 The 4-cell liquid

In this section, we will sketch what is probably the most straight-forward generalization of the volume liquid in $2+1$ dimensions to $3+1$ dimensions: There is one 5-index tensor for every 4-simplex of a (branching structure) triangulation of a 4-manifold, and if two 4-simplices share a 3-simplex, the corresponding tensors are connected by a bond.

The liquid we will sketch describes topological manifolds *without* an orientation, for reasons of variety and because the resulting liquid is a little more simple. The main tensor of the liquid is the 4-simplex

$$
\tag{305}
$$

In 4 dimensions, there are 3-3, 2-4, and 1-5 Pachner moves for this tensor, and there are many different versions of those moves due to the edge orientations. One particular 3-3 Pachner move is given by

$$
\longleftrightarrow \quad \text{same 1-skeleton} \, .
\tag{306}
$$

In network notation, this is

$$
\tag{307}
$$

As in the lower-dimensional cases, we can restrict to only this single 3-3 Pachner move if we introduce additional bond dimensions, tensors, and moves, which have geometric interpretations in terms of more general cellulations. The 3-cells for the additional bond dimensions are just the 3-cells for the additional tensors of the $2+1$-dimensional volume liquid, e.g., there are two flip hat bond dimensions, and a 2-gon bond dimension. As in $2 + 1$ dimensions, we need cancellation moves which allow us to derive 2-4 and 1-5 Pachner moves from the 3-3 Pachner moves, and symmetry moves which allow us to derive Pachner moves with different edge orientations. Permuting the vertices of the 4-simplex changes the edge orientations though, so we have to glue tensors called *4-flip hats* in order to flip them back. A 4-flip hat is given by 4-cells consisting of two flip hats and two tetrahedra, e.g.,

$$
\tag{308}
$$

We can flip an edge of a 4-dimplex by gluing three 4-flip hats to three boundary tetrahedra sharing that edge. E.g., the (01) 4-*simplex symmetry move* equates a 4-simplex mirrored at the 01 edge with one whose 01 edge was flipped. More precisely, we use a more powerful version of this move where one of the three 4-flip hats was brought to the other side:

$$
\longleftrightarrow \quad \text{same 1-skeleton .}
\tag{309}
$$

In network notation, this gives

$$
\tag{310}
$$

We get 3 different versions of the tensor in Eq. (308), depending on how we choose the orientations of the edges adjacent to the vertices 2 and 3. Using those, we also get a (12)-, (23)- and (34) tetrahedron permutation move, which generate the whole 4-simplex symmetry group. Moreover, the 4-flip hats have symmetries which however change the favourite edge of the involved 2-gon, and we need additional tensors for changing the latter.

The most important example of a cancellation move is the *4-simplex cancellation move*

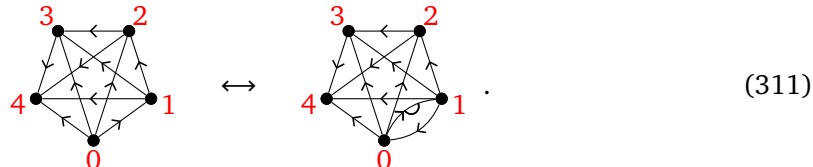

(311)

The left hand side consists of two 4-simplices, glued at two of their tetrahedra, whereas the right hand side consists of 3 4-flip hats glued at their tetrahedra in a cyclic fashion. In terms of networks, we find

$$\text{(312)}$$

There are also cancellation moves for the 4-flip hat, and so on.

In order to get the most general physical phases with gappable (i.e. topological) boundary, we would also have to introduce face weights analogous to the edge weights in $2+1$ dimension. Any face in (the interior of) a 4-dimensional cellulation corresponds to a cycle of 4-cells adjacent at 3-cells. E.g., the loop on the left side of Eq. (307) corresponds to the 024 face, and the one on the right side to the 135 face. Into each such loop we have to insert exactly one face weight.

We do not guess the bond dimensions, tensors, and moves from scratch, but follow some systematics, which we will outline briefly. The different bond dimensions correspond to different 3-cells, and the tensors to different 4-cells of those cellulations. The moves are equations between two different cellulations of the 4-ball, and if we glue both sides of a move together, we get a cellulation of a 4-sphere, which can be seen as the boundary of a 5-cell.

In order to find those 3-cells, 4-cells and 5-cells, we need an operation called the *stellar cone*, which transforms a $n$-cell into a $n+1$-cell by the following procedure. First, add an additional vertex called *central vertex*. Then, for every boundary $x$-cell, add an $x+1$-cell which is spanned by this $x$-cell and the central vertex (the original $n$-cell together with the central vertex span the $n+1$-cell itself). If there is a branching structure, there are two different choices of orientation for the new edges. Either they are all pointing towards the central vertex, or away from it.

In general, in the $n$-dimensional $n$-cell liquid, we can take as $x$-cells the stellar cones of the $x-1$-cells, which are the same as the $x-1$-cells of the $n-1$-dimensional $n-1$-cell liquid. E.g., the 3-3 Pachner moves yield a 5-simplex, which is the stellar cone of the 4-simplex representing a tensor. The 4-flip hat tensor is the stellar cone of the flip hat, which is a bond dimension of the present liquid, as well as a tensor of the 3-dimensional volume liquid (Eq. (200)), as well as the (01) triangle symmetry move of the unoriented 2-dimensional face liquid (Eq. (125)). At the same time, the 4-flip hat is the (01) tetrahedron symmetry move of the unoriented version of the 3-dimensional volume liquid. As another example, both the (01) 4-simplex symmetry move as well as the 4-simplex cancellation move yield the same 5-cell which the stellar cone of the 4-flip hat.

As the last example shows, the 5-cells can be decomposed into two 4-ball triangulations in different ways. We certainly do not want to choose all those decompositions, as, e.g., different decompositions of the 5-simplex yield all different variants of the 4-dimensional Pachner moves already. However, it always suffices to take a single Pachner move (one of the ones with the most open indices), together with symmetry cancellation moves from which we can derive all others.

The idea that fixed-point models for topological order in general dimensions can be described by "Pachner-move invariant simplex tensors" is rather straight-forward, and has been explicitly spelled out, e.g., in Ref. [16]. Our contribution here is to give a framework which allows us to arrive at a refined set of moves, containing only a single Pachner move together with a collection of simpler "auxiliary moves". An example for a model is the so-called *Kashaev invariant*, for which the 4-simplex tensor has explicitly been spelled out in Ref. [42].

## 10.2 The volume-face liquid

In this section, we sketch a less straight-forward topological liquid in $3+1$ dimensions. It is similar to the face-edge liquid in $2+1$ dimensions, in that we associate tensors to $d-1$-cells and $d-2$-cells in a $d$-dimensional cellulation. There is one tensor at every volume and one tensor at every face. If a volume and a face are adjacent, the corresponding tensors share a bond. Note that in a 4-dimensional cell complex, a face can be adjacent to more or less than two volumes. Faces adjacent to exactly 2 volumes are not represented by tensors; instead, the two volumes are directly connected by a bond. When restricting to networks with such trivial faces, we get a mapping from 3-dimensional cell complexes to 4-dimensional cell complexes, which we will call the *volume mapping*. It makes sense to use the volumes of the simplified volume liquid in $2+1$ dimensions (Section. 8.1) as volume tensors, and make all of the $2+1$-dimensional moves into volume-only moves of the $3+1$-dimensional liquid.

### 10.2.1 The liquid

A face in a 4-dimensional cell complex is Poincaré dual to another face. The adjacent volumes, and thus indices, correspond to the edges of that dual face. The full shape of the face is specified by both the shape of the face itself and the shape of the dual face. E.g., when we have a triangle face, whose dual face is a 4-gon, we will call this a 4-valent triangle. Similar to 2-valent faces, pillow-like volumes whose boundary consists of two equal faces are just represented by a direct bond between those faces. Restricting to cell complexes with triangle faces (with different dual faces), separated by such trivial pillow-like volumes, yields a mapping from $1+1$ to $3+1$-dimensional cell complexes, which we will call the *triangle face mapping*. So it makes sense to take the triangle and cyclic 2-gon as dual shapes for the triangle faces, together with all the mapped $1+1$-dimensional moves. Analogously, we get a *2-gon face mapping* by restricting to cyclic 2-gon faces with different dual faces. We will draw the 2-dimensional liquid formed by the 2-gon faces as filled circles.

Each edge is equipped with an orientation, and each volume is equipped with a dual orientation, i.e., a favourite adjacent 4-cell. Those orientations allow us to pick a 01 edge and a 01 volume of a non-cyclic and dually non-cyclic 3-valent triangle. Considering the face itself and its 01 edge inside its 01 volume, we can decide whether the face is clockwise or counter-clockwise relative to the global orientation.

Analogous to the face-edge liquid in $2+1$ dimensions, we need to introduce a 2-index *corner weight* tensor in order to get models for a very general class of phases

$$\longrightarrow\!\!\boxdot\!\!\longrightarrow . \tag{313}$$

At every pair of edge and adjacent 4-cell, there is an alternating cycle of face and volume tensors. We demand that, in a network representing a 4-manifold, there is one weight tensor inserted at every such cycle. More precisely, the weight tensors are of different tensors depending on the edge-4-cell pair and the face and volume between which they are inserted. The tensor depends on whether the face is a 2-gon or a triangle, whether the edge is the 01, 02, or 12 edge of the triangle or the favourite or non-favourite edge of the 2-gon, and whether the dual orientation of the volume points towards or away from the 4-cell. The corner weight

depicted above is for the favourite edge of a 2-gon, with the volume pointing towards the 4-cell. All other corner weights can be constructed from this single one. E.g., the favourite edge 2-gon corner weight for the volume pointing away from the 2-gon is obtained by

$$a \longrightarrow\!\!\bullet\!\!\leftarrow\!\!\square\!\!\leftarrow\!\!\bullet\!\!\longrightarrow b \ . \tag{314}$$

Or, following Eq. (215), the corner weight for the 01 edge of the triangle is obtained by

$$b \longrightarrow\!\!\boxtimes\!\!\longrightarrow f \ := \ b \longrightarrow\!\!\blacksquare \cdots \square \longrightarrow f \ . \tag{315}$$

Similarly, the corner weights for the 02 edge and the 12 edge

$$\longrightarrow\!\!\boxtimes\!\!\longrightarrow , \qquad \longrightarrow\!\!\boxtimes\!\!\longrightarrow \tag{316}$$

can be constructed, e.g., following Eq. (219) for the case of the 02 edge.

Edges of $2+1$-dimensional cell complexes stay edges under the volume mapping, and then have two adjacent 4-cells. Thus, the edge weights of the $2+1$-dimensional volume liquid can be constructed from two corner weights of the present $3+1$-dimensional liquid. For the 2-gon edge weight, we get

$$a \longrightarrow\!\!\longrightarrow b \ := \ a \longrightarrow\!\!\square\!\!\rightarrow\!\!\bullet\!\!\leftarrow\!\!\square\!\!\leftarrow\!\!\bullet\!\!\longrightarrow b \ . \tag{317}$$

Or, for the 01 edge weight

$$a \longrightarrow\!\!\longrightarrow b \ := \ a \longrightarrow\!\!\boxtimes\!\!\rightarrow\!\!\bullet\!\!\leftarrow\!\!\boxtimes\!\!\leftarrow\!\!\bullet\!\!\longrightarrow b \ . \tag{318}$$

Vertices in $1+1$-dimensional cell complexes become 4-cells under the triangle face mapping, and are then adjacent to three edges. Thus, the vertex weight of the $1+1$-dimensional triangle face liquid can be constructed from three corner weights

$$a \longrightarrow\!\!\square\!\!\longrightarrow b \ := \ a \longrightarrow\!\!\boxtimes\!\!\rightarrow\!\!\boxtimes\!\!\rightarrow\!\!\boxtimes\!\!\longrightarrow b \ . \tag{319}$$

Similarly, the vertex weight of the $1+1$-dimensional 2-gon face liquid consists of two corner weights

$$a \longrightarrow\!\!\blacksquare\!\!\longrightarrow b \ := \ a \longrightarrow\!\!\boxtimes\!\!-\!\!\diamond\!\!-\!\!\boxtimes\!\!-\!\!\diamond\!\!\longrightarrow b \ . \tag{320}$$

So far, we got one copy of the 3-dimensional volume liquid, and two copies of the 2-dimensional face-liquid. We now need moves which connect the tensors of those liquids, by "pulling faces through volumes". Roughly, every such move can be constructed from a quadrupel consisting of a volume $V$, a dual face $F_D$, a special face of the volume, and a special edge of the dual face. The move involves one $V$ volume tensor for each edge of $F_D$, and one $F_D$-valent $F_V$ face tensor for each face $F_V$ of $V$. The volume and face tensor corresponding to the special edge and face are on one side of the move, and all the others on the other side.

As an example, pick for $V$ the 01 flip hat (with one of the triangles as special face), and for $F_D$ the triangle (with the 02 edge as special edge). The corresponding pull-through move is given by

$$\tag{321}$$

Here, the empty circles represent the triangle face tensors, and the full circles represent the 2-gon face tensors (both 3-valent in this case). Note that the two $1+1$-dimensional liquids formed by the triangle and 2-gon tensors do *not* form a $2+1$-dimensional face-edge liquid together, as in Section 8.2.

As another example, pick for $V$ the 12 flip hat, and for $F_D$ the cyclic 2-gon. We get a move that pulls the 2-valent face-atoms through the flip hat and thereby changes its orientation,

$$
\tag{322}
$$

As a further example, pick for $V$ the tetrahedron, and for $F_D$ the triangle. We get

$$
\tag{323}
$$

Unfortunately, the pull-through moves as described above are not quite enough to have a fully topological liquid (such that there is an invertible mapping to the 4-simplex liquid sketched in the section above). We also need to allow moves where $V$ has two special faces, such that the left hand side consists of a volume tensor with *two* face tensors. However, this move would not be topological (that is, it would prevent a mapping back from the 4-simplex liquid to the present liquid). In order to make it topological, we need to add a "projector onto two neighbouring triangles", which we can build from 4 flip hats,

$$
\tag{324}
$$

### 10.2.2 Models

A well-known class of fixed-point models for topological phases in $3+1$ dimensions are *second order gauge theories*. Analogously to ordinary gauge theories being based on a gauge group, a second order gauge theory is based on a *2-group*. A 2-group is concretely defined by what is called a *crossed module*. The latter consists of two groups $G$ and $H$, with a homomorphism

$$
h : H \to G
\tag{325}
$$

from $H$ to $G$, and an action

$$
\alpha : G \times H \to H
\tag{326}
$$

of $G$ on $H$.

Following Ref. [43], a more condensed representation of (equivalence classes of) 2-groups is given by a group $\Pi_1$, a commutative group $\Pi_2$ (arising from $G$ and $H$ as the kernel and co-kernel of $h$), an action $\alpha$, and a $\Pi_2$-valued group 3-cocycle of $\Pi_1$ with action $\alpha$, that is, a map

$$
\beta : \Pi_1 \times \Pi_1 \times \Pi_1 \to \Pi_2 \, ,
\tag{327}
$$

such that

$$
\begin{aligned}
&\beta(ab,c,d) + \beta(a,b,cd) \\
&= \alpha(a,\beta(b,c,d)) + \beta(a,bc,d) + \beta(a,b,c) \, ,
\end{aligned}
\tag{328}
$$

where we denoted group multiplication in $\Pi_2$ additively.

Like the fusion category models of the $2+1$-dimensional volume liquid, 2-group models of the present liquid are most conveniently formulated using label-dependent tensors. The labels are the elements of $\Pi_1$, and might be thought of as being located at the edges of the complex. The dimension of the indices at a face is either $|\Pi_2|$ if the edge labels around the face multiply up to 1, and 0 otherwise. All face tensors are given by delta functions (in every valid edge label configuration), e.g.,

$$
\begin{array}{c}
a \quad\searsow\quad b \\[-4pt]
\phantom{.}\text{\Large$\bigcirc$}\phantom{.} \\[-4pt]
c
\end{array}
\;=\;
\begin{array}{c}
a \quad\searsow\quad b \\[-4pt]
\bullet \\[-4pt]
c
\end{array}
\;. \tag{329}
$$

where $a$, $b$ and $c$ are elements of $\Pi_2$. If we denote the label of, e.g., the 03 edge of the tetrahedron by $e_{03}$, then tetrahedron is given by

$$
d \;-\!\!\boxed{\phantom{x}}\!\!-\; b \;=\;
\begin{cases}
\frac{1}{|\Pi_2|} & \text{if} \quad \alpha(e_{01}, a) - b + c - d \\
& \quad = \beta(e_{01}, e_{12}, e_{23}) \\
0 & \text{otherwise}
\end{cases}
\;. \tag{330}
$$

The edge weights are all identity matrices. The flip hats can be obtained from the tetrahedron by the mapping in Eq. (223). The resulting dimension of the 2-gon index is $|\Pi_1||\Pi_2|^2$. Of course, we can also interpret the edge labels as indices, copy and block them into the face indices to obtain ordinary tensors.

## 11 Summary and conclusion

In this work we introduced a systematic graphical language which allows us to think about fixed-point models for (topological) phases for various scenarios in a unified way, and stimulates and facilitates the search for new families of fixed-point models corresponding to combinatorial representations of space-time. The following diagram summarises central concepts of our framework and their relations.

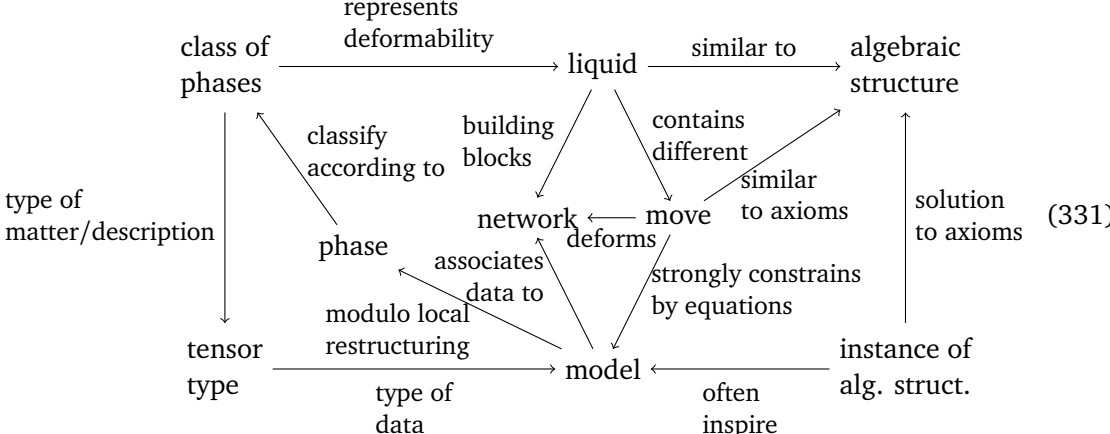

$$\tag{331}$$

There are five main goals we attempt to achieve with the formalism introduced. The first goal is to sort the vast body of existing literature on fixed-point models by introducing a simple, systematic, and unified mathematical language. All of those models are based on algebraic or categorical structures defined by a set of equations. In our point of view, all those equations are manifestations of one central property, namely topological invariance in Euclicean space-time. All other properties, such as commuting-projector Hamiltonians, or PEPS representations with virtual symmetries, are direct consequences of the topolgical invariance, but

not the other way round. That is why we believe that topological invariance should be the point to start with. Instead of "guessing" an algebraic or categorical structure used as an input to fixed-point construction, we go the other way round, deriving algebraic structures from the topological invariance. Tensor networks with their familiar Penrose notation appear as the natural mathematical language, as they represent at the same time combinatorial representations of the space-time topology (and their moves) as well as the path integral itself (and its equations). We would also like to mention that unlike "string diagrams" in conventional algebra or category theory, tensor networks are not formulated with an inherent flow of (real-)time, and thus more natural to represent path integrals in Euclidean space-time. One of the central technical tools that we have developed in this work for the systematic study of different fixed-point ansatzes are liquid mappings. The most important use of liquid mappings in this paper was to define a notion of equivalence of liquids.

The second goal is to obtain a deeper understanding of existing families of models, and formulate them in their most general form. For some known state-sum constructions and fixed-point models, the explanation in the literature for why they take on exactly the form they have, is insufficient. Our work addresses these issues. In particular, we explain the role of an orientation, and show the necessity to add a branching structure to state-sums based on triangulations. It is easy to see that every topologically invariant path integral with topological boundary can be coarse grained into a Pachner-move invariant simplicial tensor network. We show how to get from many Pachner-move equations (due to different branching structures) to a single one by extending the construction to more general cellulations. In Appendix D, we show how to arrive at the usual Turaev-Viro form of the state-sum. Furthermore, we introduce a new path integral picture for weak Hopf algebra based quantum doubles. Quantum double models have been mostly studied from the perspective of commuting-projector Hamiltonians [14,15]. The commutativity follows from the weak Hopf axioms, however, a direct motivation for why weak Hopf algebra related structures are the correct input for those models was still lacking. We demonstrate that those structures directly emerge from a combinatorial version of topological invariance. In particular, the central bi-algebra axiom corresponds to a topological move which "pulls an edge through a face" as in Eq. (249).

The first two goals have been addressed to a large extent in the present paper by working out concrete examples. The other three goals will be worked out in future publications. The third goal is that we can systematically construct new combinatorial representations of topological manifolds, yielding new classes of models. Using our formalism, this can be achieved quickly with a bit of geometric intuition and creativity. Let us sketch one example for an alternative liquid/fixed-point ansatz in 2+1D. Consider cellulations where all vertices are coloured red, blue, or green, and all faces are triangles with one red, one blue, and one green vertex. Now, associate tensors to the volumes and contract between volumes sharing a face. The simplest volume compatible with the colouring is the octahedron,

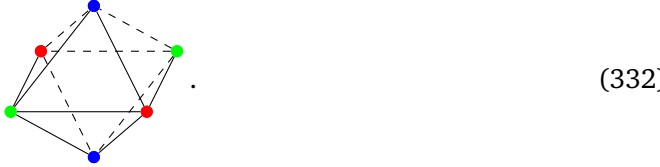 (332)

The major topological moves are given by mapping between the different ways of decomposing

a "diamond" into octahedra, e.g.,

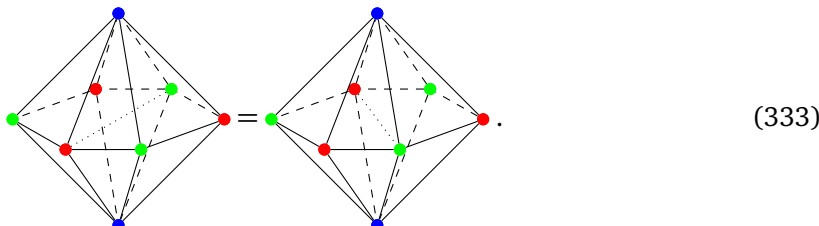

$$\tag{333}$$

Octahedron tensors obeying the moves form a new family of models, one of which turns out to be a path integral representation of the well-known *colour code*. If we add appropriate edge weights, the "tricoloured" liquid will be equivalent to a standard triangular liquid, so the two families of models describe the same phases. In particular, the colour code is known to be phase-equivalent to two copies of the toric code. It is, however, a different microscopic model and has advantages (and disadvantages) over the latter for error-correction purposes. Having different microscopic realisations of the same phases is important for engineering those phases (e.g., for building a quantum computer), and our framework yields a method of systematically constructing such new realisations.

   A fourth goal is to make the study of phases of matter more accessible to automatisation. The standard task on the combinatorial/graphical level of liquids is to find derivations of moves from a given set of moves. This is needed for the verification of liquid mappings, which is important for showing that certain liquids are equivalent. We think that it is possible to automatise the process of finding derivations numerically. Of course, in the general case, finding derivations is an undecidable and certainly hard problem, as tasks like theorem proving can be relatively easily encoded in finding derivations. However, the networks we deal with do not represent arbitrary logical statements, but patches of low-dimensional manifolds. Surely, proving the equivalence of manifolds based on triangulations is a hard and undecidable problem as well in general, but this is only if we scale the complexity of the topology. In our case, the manifold patches have a simple and constant topology (two balls if we are dealing with topological liquids), and we merely scale the size of the triangulation and not the complexity of the topology itself.

   The standard task on the level of models is, of course, finding models. For conventional array tensors, but also for fermionic tensors or tensors with symmetry, the moves turn into polynomial equations for the tensor entries. In principle, even though with a possibly high computational effort, we can find roots to those equations by iterative numerical optimization methods, such as non-linear conjugate-gradient, or Gauss-Newton methods. The cost of each iteration scales with as a high-degree polynomial (depending on the algorithm and on how complicated the liquid is) in the bond dimension, and the volume of initial conditions for which the iteration actually converges to a global minimum might be small. However, we expect, that only the simplest phases, i.e., the ones realizable with a low bond dimension, are physically relevant. Roughly speaking, the higher the bond dimension needed to realize the phase, the more unlikely it will be to encounter it in nature, and the harder it will be to experimentally realize it. So for practical purposes, we can restrict to small bond dimensions where numerical methods might still be feasible. Note that also for known families of fixed-point models, their equations boil down to polynomials. However, due to the systematics of our language we do not have to write a separate algorithm for every different family of fixed-point models, but can take the latter as an input to a single algorithm.

   As a fifth goal, we would like to mention that, apart from obtaining new models for the same phases, our formalism has the potential to go beyond known constructions and obtain fixed-point models for new phases. There are liquids that are not equivalent to the standard ones, which means that the corresponding path-integral tensor networks cannot be coarse-

grained into a standard simplicial form [12]. This does not directly imply that the more general liquids have models for more general phases, but it does indicate that this is at least possible. An exciting candidate for more general phases are chiral phases in 2+1D which are lacking any fixed-point description so far. The generalised liquid models are compatible with the absence of both a topological (i.e. gapped) boundary and commuting-projector Hamiltonians, features characteristic for chiral phases.

In addition to these broad goals, there are the following three concrete ways to generalize the scope of the formalism, some of which have been mentioned briefly in the main text.

First, we have focused on liquids for topological order in the bulk. Boundaries, anyons, and other sorts of defects are described by liquid models as well, as we have sketched for boundaries in Section 3.3. E.g., there is a liquid describing anyons within the 3-dimensional face-edge liquid, whose models turn out to be similar to representations of quantum doubles of weak Hopf algebras. Or, there is a liquid describing boundaries of the 3-dimensional volume liquid, whose models are similar to modules of fusion categories. We have already seen the possibility to add extra structures like orientations in Section 6 or spin structures in Section 9, and the possibility to add beyond-topological moves, such as the ones that guarantee invertibility of the model in Section 7. A much more novel pursuit would be the formalisation of conformal, not topological, field theories in terms of liquid models. To this end, one would need a semi-combinatorial representation of conformal manifolds, together with moves preserving the conformal structure [5].

Second, all of our liquid models are microscopic physical models defined by a concrete local partition function. This is in contrast to the description of phases via more abstract and indirect invariants, such as (non-fully extended) axiomatic TQFT, giving rise to structures like (non-special) commutative Frobenius algebras in $1 + 1$ dimensions, or modular tensor categories in $2 + 1$ dimensions. All these structures can be formulated as liquids as well, as long as they have a finite set of generators and relations (which roughly appears to be the case for TQFTs extended down to at least the circle). The relation between those more abstract invariant liquids and the concrete microscopic liquids is formalised by a liquid mapping from the former to the latter. The most famous example for this is the quantum double, or Drinfeld centre of fusion categories, or Hopf algebras.

Third, another point that was not the focus of this work is the role of tensor types. We saw that models with symmetries, fermions, or models which are deformable only up to pre-factors, can be incorporated by using different tensor types. We also saw that physics imposes a fixed relation of certain tensor types with certain kinds of extra structures added to liquids, such as complex tensors with orientation via Hermiticity, or fermionic tensors with spin structures via a spin statistics relation. What we did not mention so far is that certain restrictions to "exactly solvable" classes of models can also be formulated as tensor types, such as non-interacting (i.e., Gaussian, quadratic, free) fermionic models, or models that can be formulated within the stabilizer formalism. It is the hope that this work stimulates such further endeavours.

## Acknowledgements

We thank the DFG (CRC183 project B01, for which this is an internode Berlin-Cologne publication, and EI 519/15-1) and the Studienstiftung des Deutschen Volkes for support. Many thanks goes to A. Nietner, M. Kesselring, and N. Tarantino for lots of inspiring and fruitful discussions.

---

[5]In contrast to topological manifolds, equivalence classes of conformal manifolds do not form a discrete set, but a continuous space. Still, the dimension of this space is finite for fixed topology, so it should suffice to additionally assign continuous variables to some places of the triangulation, such as one at every vertex.

## A  Overview over the main definitions

Since all the relevant concepts were introduced in the main text in a step by step way, it is useful to summarise them in this appendix.

A *liquid* consists of a finite set of tensor variables and bond dimension variables, and finite set of tensor-network equations for those variables. Those are equations between two networks whose open indices and their bond dimension variables match, and are called *moves* of the liquid. A *model* of the liquid associates to every bond dimension variable a bond dimension and to every tensor variable a tensor, such that all the equations given by the moves hold.

Moves can be composed to yield other moves, and such a composition is called a *derivation* of the resulting move. A *liquid mapping* from a *source* liquid $A$ to a *target* liquid $B$ associates 1) to every bond dimension variable of $A$ a collection of bond dimension variables of $B$ and 2) to every tensor variable of $A$ a network of $B$, such that every index of the $A$-tensor variable corresponds to an according collection of open indices of the $B$-network. Applying this replacement to all tensor copies in a move of $A$, we obtain another move, called the *mapped move*. The mapped moves have to be derived moves of $B$. The mapping can be applied to a model of $B$ to yield a model of $A$. The compatibility of the mapping with the moves precisely ensures that the moves hold for th e$A$-model if they hold for the $B$-model.

For two liquid mappings $A \to B$ and $B \to A$ let us refer to their composition $A \to B \to A$ as the *double mapping*. Two such mappings are *weakly inverse* to another if we can form an invertible domain wall between a model and the double mapping of the model, i.e., a network and its double mapping are equivalent up to moves apart from near the open indices. The same has to hold for the other double mapping, the composition $B \to A \to B$. Liquids $A$ and $B$ for which there exists a pair of mappings weakly inverse to another are *equivalent* in the sense that the phases they capture are in one-to-one correspondence.

## B  Vertex weights for the triangle liquid

In this appendix, we discuss the following variant of the above liquid, which is slightly more general. The main reason for introducing this variant is that it shows up as sub-liquid of a $2+1$-dimensional liquid in Section 8.2.

- The variant has one more tensor variable, called the *vertex weight*

$$\longrightarrow\!\Box\!\!\blacktriangleleft\!\!- \ . \tag{334}$$

- The *weighted oriented triangle cancellation move* replaces the oriented triangle cancellation move

$$a \rightarrow\!\!\!\! \text{⬭} \text{⬭} \!\!\leftarrow\!\! b = a \rightarrow\!\!\text{●}\!\!\leftarrow\!\! b \ . \tag{335}$$

- There is the additional *weight commutation move*

$$a \rightarrow\!\Box\!\rightarrow\!\!\text{●}\!\!- b \underset{c}{} = a \rightarrow\!\!\text{●}\!\rightarrow\!\Box\!- b \underset{c}{} \ . \tag{336}$$

Every vertex of the triangulation corresponds to a loop of bonds in the network. Each vertex weight is bound to one such loop, that is, it can be moved around that loop using the weight commutation and other moves. Topological manifolds are represented by networks, where each loop has exactly one vertex weight bound to it. Vertices/loops with more or less vertex

weights can never be removed, as the weighted cyclic triangle cancellation move involves exactly one vertex weight. Thus, they should be thought of as some kind of singularity. The position of the vertex weight corresponds to an edge in the triangulation, which can be interpreted as the "favourite edge" of the corresponding vertex. The weight commutation move makes the evaluation of networks independent of those favourite edge decorations.

The variation is "more general" than the original liquid, in that there is a liquid mapping

$$a\text{—}\square\hspace{-2pt}\blacktriangleleft\, b := a\text{——}b \tag{337}$$

from the variant to the original liquid, but no obvious inverse mapping. Indeed, the models of the variant (in real array tensors) are slightly more general. E.g., for each $\alpha \in \mathbb{R}$ [6], there is the following model where all tensors are scalars: The triangle is the scalar $\alpha^{-1/2}$, the 2-gons are the scalar 1, and the vertex weight is the scalar $\alpha$. One can easily see that the evaluation of this model on a space-time manifold $M$ is $\alpha^{\chi(M)}$, where $\chi$ is the euler characteristic. Note, however, that as a physical model using projective tensors (as explained in Section 4.5), this model is immediately trivial. In fact, using this tensor type, we do not get any new phases compared to the liquid without vertex weights.

## C  Remaining moves for the volume liquid in $3$ dimensions

In this appendix, we complete the moves of the simplified $2 + 1$-dimensional liquid from Section 8.1.2. In order to generate the full orientation-preserving symmetry group of the tetrahedron, we have to add the (012) *tetrahedron symmetry move*

$$
\begin{array}{c}
\end{array}
\tag{338}
$$

We also need the remaining symmetry moves of the flip hats, i.e., for ones for the counter-clockwise 01, the clockwise 12 and the counter-clockwise 12 flip hats

$$
\tag{339}
$$

$$
\tag{340}
$$

$$
\tag{341}
$$

Regarding the cancellation moves, we need to add the cancellation move for the 12 flip hats

$$a\text{—}\square\,\overset{}{\smile}\,\blacksquare\text{—}b = a\text{——}b \tag{342}$$

and the 2-*gon flip cancellation move*

$$a\text{—}\lozenge\,\lozenge\text{—}b = a\text{——}b \,. \tag{343}$$

---

[6]If we drop the hermiticity move, this is a model also for complex $\alpha$.

# D   From the face-edge liquid to Turaev-Viro models

In this section, we describe how to reshape complex models of the 3-dimensional face-edge liquid into state-sums of Turaev-Viro form. First of all, we further extend the liquid by introducing the non-cyclic 2-gon as a further bond dimension variable. The latter can be triangulated with two triangles

$$
0 \bullet\!\!\underset{\phantom{0}}{\overset{\phantom{0}}{\frown}}\!\!\bullet 1 \quad \rightarrow \quad 0 \bullet\!\!\overset{\phantom{0}}{\underset{2}{\bullet}}\!\!\bullet 1 \ , \tag{344}
$$

so, the new bond dimension is defined by

$$
\underset{01}{C} := \underset{012}{T}\ \underset{102}{T} \ . \tag{345}
$$

Note that the favourite edge is needed to determine the ordering of the two triangles. We also need it to define when a 2-gon is clockwise or counter-clockwise. With the new 2-cell bond dimension, we can construct banana-like volumes whose boundaries consist of non-cyclic 2-gons glued at edges, e.g.,

$$
\tag{346}
$$

The boundary of the volume on the right consists of 3 2-gons, two in the front and one in the back. If we want to construct this new tensor, we have to replace every 2-gon by two triangles as in Eq. (344), and triangulate the resulting volume. Precisely the same volume was triangulated in Eq. (257), and thus the corresponding tensor is the same as the edge tensor of the equivalent face-edge liquid, defined in Eq. (258). So bananas define a (mapping from a) $1+1$-dimensional face liquid.

The *01 hat* can be triangulated by two tetrahedra

$$
\tag{347}
$$

In network notation, we have

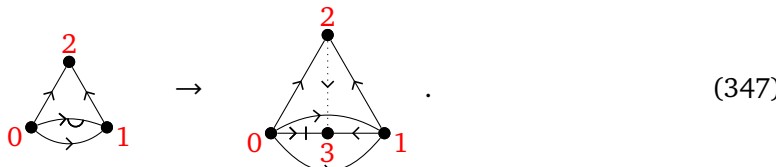

$$
\tag{348}
$$

This tensor defines a topological boundary for the banana liquid. Consider the following re-cellulation from two flip hats glued at a triangle on the left to one flip hat and a banana glued at a 2-gon on the right

$$
\rightarrow \quad \text{same 1-skeleton} \ . \tag{349}
$$

This corresponds to the move whose simplified version is depicted in Eq. (102)

$$
\tag{350}
$$

Note that there is also a counter-clockwise version of the 01 hat, for which the 13 instead of the 03 edge carries an edge weight. Also, there are the 02 and the 12 hats (together with their counter-clockwise counterparts), which are the same apart from different edge orientations of the 02 and 12 edges in Eq. (347). The topological moves further imply that those representations commute, thus they form a single representation of the product of the three algebras.

Consider an arbitrary 3-volume with boundary, the network $P$ representing it, and an arbitrary edge on the boundary of the volume together with the two adjacent triangles, e.g.,

$$\tag{351}$$

We can glue two hats at their 2-gons, and then glue the resulting "double-hat" to the two adjacent faces above. This corresponds to a topological move

$$\tag{352}$$

Which hats we take depends on the edge orientations. In our case, we get a move

$$\tag{353}$$

This is as far as we get on the combinatorial liquid level, and now we have to make use of the fact that we're looking for models of the liquid in complex tensors. All the algebras and representations in question have extra properties due to the topological moves which make them block-diagonalizable. That is, we can go to a basis where the algebra is given by

$$\beta cd \text{—}\bullet\text{—} \gamma ef = \tag{354}$$

and the representation is given by

$$\beta cx \text{—}\square\text{—} \gamma dy = \tag{355}$$

Here, $\alpha, \beta, \dots$ are called *irreducible representation indices*, $a, b, \dots$ are called *block indices*, and $x, y$ *multiplicity indices*. Note that this is a fake tensor network notation, as the dimension of both block and multiplicity index are allowed to dependent on the value of the irreducible representation indices.

As we saw above, the vector space of the triangle is equipped with a representation of three times the banana algebra. Going to the block-diagonal basis, we can decompose the vector

space into three indices corresponding to irreducible representations of the banana algebra, three block indices, and one joint multiplicity index. The irreducible representation and block indices can be associated to the three edges of the triangle each. Note that the dimension of each block index depends on the dimension of the corresponding irreducible representation index, and the multiplicity index depends on the values of all three irreducible representation indices (this dimension becomes the $N_c^{ab}$).

All we need to show is that 1) we can get rid of the block indices and 2) the two irreducible representation indices at an edge coming from the two adjacent triangles can be unified into one. To this end, we look at Eq. (353) with the representations in their block-diagonal form

$$
\text{(356)}
$$

Applying these procedure to all edges, we get a tensor $\widetilde{P}$ with irreducible representation indices at all edges and multiplicity indices at all faces. Now, we plug the above equation into the network representing a cellulation. For each edge of the cellulation, we get 1) a completely disconnected loop of block indices, and 2) a loop of delta tensors connected to the tensors at the adjacent volumes. The loop 1) can be contracted to a scalar which can be incorporated into the edge weight, and the loop 2) can be contracted to a single delta tensor, e.g., for an edge with 3 adjacent volumes we get

$$
\text{(357)}
$$

where $m$ consists of the multiplicities of the different irreducible representations.

# E  Reordering signs for the remaining fermionic moves

In this appendix, we compute the reordering signs for the remaining moves of the $1 + 1$-dimensional fermionic liquid, and find that they all cancel out.

- For the spin triangle cancellation move Eq. (285)

$$
\begin{aligned}
|y'x'axyb|(xx')(yy') &= |ab|, \\
(a+x+y)|ba| &= |ab|, \\
|ab| &= |ab|.
\end{aligned}
\tag{358}
$$

- For the spin (012) triangle symmetry move Eq. (287)

$$
\begin{aligned}
|y'cxax'by|(xx')(yy') &= |abc| \\
(ax)|bcxx'a| &= |abc|, \\
(a)|bca| &= |abc|, \\
|abc| &= |abc|.
\end{aligned}
\tag{359}
$$

- For the spin 2-gon cancellation move Eq. (289)

$$
\begin{aligned}
|axbx'|(xx') &= |ba|, \\
|ab| &= |ab|.
\end{aligned}
\tag{360}
$$

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
