# Peer review of "A unified diagrammatic approach to topological fixed point models"

_SciPost Physics, doi:SciPost Phys. Core 5, 038 (2022)_

## Round 2 · Referee Report · Anonymous (Referee 1) · 2021-1-20

Strengths

1. Very clear exposition
2. integrates lots of results scattered in the literature

Weaknesses

1. The authors promise to be more general than known approaches, but at the end they only discuss fixed point models
2. Intermingle mathematically rigorous statements with sloppy ones
3. No clear message or results

Report

This paper is rather a review paper than a research paper, and explains how topological field theory and their Euclidean path integral representations in terms of state-sums can be interpreted as tensor networks. The goal is "to introduce a comprehensive systematic, unified, easily accessible, and generalized language for understanding and exploring fixed-point models for quantum phases of matter"; the authors seem to have succeeded partly in their endeavor, but their constructions seem to be effectively equivalent to the ones already present in the literature and it is not clear at all that they are any simpler.

The authors set out to define tensor networks for ground states of quantum spin Hamiltonians of interest by Trotterizing Euclidean path integrals, and claim that this is "the key difference" with known constructions. Such constructions are certainly not new, and seem to have been used to construct the first PEPS representations of topological states such as the toric code.

The main body of the paper discusses fixed point quantum spin models (zero-correlation length) in one, two and three dimensions. The fact that those models can be represented as tensor networks might sound nontrivial, but the authors show convincingly that this indeed trivially follows from the path integral construction (this seems to have been known since the early days of tensor networks). There might however be a fundamental flaw in such constructions for systems that are away from the zero correlation length idealizations such as the AKLT model: in the latter case the Trotterization will lead to an integer spin bond dimension, while it should of course be half-integer. It is not clear how this topological feature emerges from the construction described in the paper.

In summary, this paper is certainly a beautiful summary and exposition of the tensor network way of looking at topological field theories. However, it is difficult to identify any new ideas which would allow this formalism to go beyond what is already known in the field.

Some more remarks:

1. In equation 16 of their paper, the authors formulate a conjecture about Trotterization. The exponential decay seems to be in violation with the bounds derived in e.g [M.B. Hastings, Phys. Rev. B 73, 085115 (2006)]. In the same section, the definitions used for notions such as quasi-adiabatic evolution are very confusing to the reader not yet familiar with them.

2. In the outlook, the authors stress that a big advantage of their method is given by the fact that "the moves turn into polynomial equations for the tensor entries; we can find roots to those equations by applying a Gauss-Newton method". The equations that they have to solve are most probably equivalent to the pentagon equations in the field of fusion category, and those are notoriously hard to solve and definitely not simply solvable by Gauss-Newton methods.

  • validity: high
  • significance: ok
  • originality: ok
  • clarity: good
  • formatting: good
  • grammar: excellent

Author:  Andreas Bauer  on 2021-02-22  [id 1260]

(in reply to Report 1 on 2021-01-20)

We would like to thank the first referee very much for the comprehensive and detailed report. We are happy to read that the referee thinks that we can present a “beautiful summary and exposition of the tensor network way of looking at topological field theories”. However, it seems that what has become much less clear are the innovative aspects of our work. We have taken these remarks seriously. Subsequently, we will comment on these remarks to clarify what we have done to accommodate those elements of criticism. We have attached a version of the manuscript with corresponding suggestions for changes and additions in red. The three main weaknesses listed in the report are addressed in the following.

  • “The authors promise to be more general than known approaches, but at the end they only discuss fixed point models”

It is fully true that the topic of the paper is fixed-point models. We do not claim that our formalism generalizes to non-fixed-point models, but that it is a tool that can be used to systematically create new families of fixed-point models. Only Section 2 and parts of Section 3 also deal with non-fixed-point models. Their purpose is to explain to the reader how tensor-network path integrals relate to Hamiltonian models, and how our phase definition at the fixed-point is related to non-fixed-point phase definitions.

  • “Intermingle mathematically rigorous statements with sloppy ones”

The mathematical statements in this work can be divided into 3 groups: 1) Statements about families of fixed-point models in our formalism, 2) statements about the classification of models within one family, and 3) statements about non-fixed-point models. Statements in the first group is what the core of this paper is about: We do neither aim to obtain a classification of phases on a non-fixed-point level, nor to obtain a systematic classification of the solutions of any fixed-point equation. All statements in the first group are purely combinatorial, rigorous, and self-contained. Statements in the second group are rigorous but not self-contained and referring to the literature on the classification of finite-dimensional *-algebras. Statements in the third group (in Sections 2 and 3) might not always be formulated in full mathematical precision, which is deliberate, as this is not the main scope of the paper.

  • “No clear message or results.”

The message is that the presented formalism allows us to derive all known families of fixed-point models in a unified way, obtain a deeper understanding of why those fixed-point models have the precise form they have, and construct new families of fixed-point models. We have tried to make the message more clear in the conclusion (see suggested changes in red).

In addition to these general remarks we would like to answer some of the questions raised in the report in more detail. It seems that there are several misconceptions about our work which we would like to clarify in the following.

  • “This paper is rather a review paper than a research paper, and explains how topological field theory and their Euclidean path integral representations in terms of state-sums can be interpreted as tensor networks.”

We agree that our work is not a typical research paper in the sense that it does not aim to answer a specific question that has been posed before. Instead, we propose a very general method which we introduce pedagogically and at hand of many simple examples that are also familiar from existing literature. However, we believe that the amount by which our presentation differs from other presentations, and the amount of new findings and insights both on a level of global understanding and on a level of concrete technical findings (examples given below) exceeds by far what is common for a review.

  • “The goal is "to introduce a comprehensive systematic, unified, easily accessible, and generalized language for understanding and exploring fixed-point models for quantum phases of matter"; the authors seem to have succeeded partly in their endeavor, but their constructions seem to be effectively equivalent to the ones already present in the literature and it is not clear at all that they are any simpler.”

Of course, the question of whether one or another formalism is simpler is subjective to a large degree. However, the evaluation of a tensor network only involves sums and products of numbers, and we do believe that it is easier accessible to the average undergrad physics student than the definition of spherical fusion categories. Moreover, spherical fusion categories only describe one specific family of fixed-point models for topological order in the bulk in 2+1D, whereas liquid models can formalize fixed-point models of different combinatorial structures, in different dimensions, and for arbitrary defects. On a more technical level, we would like to remark that the language of category theory has an inherent flow of time, as morphisms go from a source to a target. In order to be compatible with topology, this flow of time has to be removed by adding extra structures such as rigidity and sphericality. Tensor networks do not have any notion of a flow of time from the very beginning, and are therefore more naturally suited to describe Euclidean path integrals.

  • “The authors set out to define tensor networks for ground states of quantum spin Hamiltonians of interest by Trotterizing Euclidean path integrals, and claim that this is "the key difference" with known constructions. Such constructions are certainly not new, and seem to have been used to construct the first PEPS representations of topological states such as the toric code.”

The difference to existing tensor-network ansatzes for topological order is that we do not consider “tensor networks for ground states” in the sense of MPS or PEPS at all. Instead, we work with a path integral tensor network living in Euclidean space-time. We have tried to stress this even more (see changes in red in Section 2) in order to remove this possible source of confusion. We do not know a place in the literature where our prescription of turning a continuous time evolution into a path-integral tensor network is spelled out explicitly. Surely, many people are aware that Trotterization yields a product of local operators and therefore a tensor network, but without fixing a unit cell, blocking, and truncating we cannot define a sensible continuum limit. We do agree that this should be the obvious way of representing a continuous time evolution as a tensor network, and for exactly that reason we believe that it’s important to have this spelled out somewhere explicitly as a conjecture.

  • "There might however be a fundamental flaw in such constructions for systems that are away from the zero correlation length idealizations such as the AKLT model: in the latter case the Trotterization will lead to an integer spin bond dimension, while it should of course be half-integer. It is not clear how this topological feature emerges from the construction described in the paper."

There is no contradiction to the Trotterization procedure for models with symmetry such as the AKLT model. The symmetry group of the AKLT model is SO(3) (not SU(2)), which does not have any half-integer representations. The Hamiltonian H, as well as e^{-\beta H}, as well as tensors of our path-integral tensor network, are all invariant under the proper SO(3) symmetry and thus do not have any half-integer components. This is not a contradiction to the system being in a non-trivial SPT phase. We would like to remark that in this case, also the truncation tensors (called I^x_n in the paper) are restricted to be invariant under the proper SO(3) symmetry.

It is only the MPS representation of the ground state where we would expect a half-integer projective SO(3) representation on the virtual indices, not the path integral tensor network. So the prescribed problem of not getting half-integer representations is rather one of the problems of the conventional way of representing ground states via MPS, which is resolved in our picture. By the way, this isn’t actually a problem in the conventional way either: An MPS representation of an SPT state (like the AKLT model) does not need to have projective representations on the virtual indices. One can easily construct MPS representations that are invariant under the proper SO(3) symmetry, e.g., by simply tensoring an identity with a spin-1/2 representation on the virtual level. This comes at the expense of introducing a degeneracy of the MPS transfer operator, though. In fact, in our fixed-point picture, MPS representations and topological boundaries are one and the same thing (e.g., look at Eq. (75) restricted to the boundary -- it's an MPS), and the degeneracy of the transfer operator of the SO(3)-symmetric MPS is the same as the ground state degeneracy of the model on an open interval due to "protected edge modes". In general, the tensor-network path integral of a SPT model is properly invariant under the SPT symmetry; projective representations are only needed if we want to have non-degenerate boundaries/MPS representations.

  • “In summary, this paper is certainly a beautiful summary and exposition of the tensor network way of looking at topological field theories. However, it is difficult to identify any new ideas which would allow this formalism to go beyond what is already known in the field.”

We agree that all the examples for liquids we give are related to some existing class of models. However, there are various ways in which our exposition of the material goes beyond what is already known in the field. We will try to make this more clear in the text. For now, let us name a few examples: There is no good justification why the Turaev-Viro state-sum in its precise form is supposed to describe the most general topological phases with gappable boundary. In our formalism it is very easy to see that every 2+1D tensor-network path integral with topological deformability can be blocked into one with one tensor per tetrahedron of a branching structure triangulation, invariant under Pachner moves. The idea of directly using “Pachner-move simplex tensors” instead of the more fine-tuned version of the state-sum can also be found in [2]. However, this reference does not solve the problem of how to impose the independence of the path integral under changes of branching structure. We give a very elegant solution to this problem, by introducing auxiliary tensors and equations, which can be interpreted in terms of more general cellulations. This explains why it suffices to consider one single Pentagon equation corresponding only to one specific choice of branching structure. In Appendix C, we show how to get from the “Pachner-move invariant simplex tensor” to the more refined form of the Turaev-Viro state-sum. Quantum double models based on weak Hopf algebras have been worked out in [1]. However, the very simple tensor-network path-integral we give has not been known so far. In particular, we find that our interpretation of the weak Hopf algebra axioms as topology-preserving moves is very enlightening and leads to a much deeper understanding of the corresponding models. In general, the idea of considering other forms of state-sums that are not triangulation-based is new, and opens up a new area with a lot of space for creativity. Such new state-sums yield new families of models. Even when those new models correspond to the same phases as known ones, having different microscopic representations of a phase is useful for practical purposes such as engineering them in practice. We have added a small example in the conclusion -- the so-called color code can be seen as a model of a liquid on tri-colored cellulations in 2+1D. Apart from getting new models for the same phases, our formalism also has the potential to describe new phases that do not have any fixed-point models so far: It is actually not true that every topological path integral can be blocked into the “standard form” with one tensor per simplex. Roughly speaking, this is because there can be tensors exactly at the vertices of the coarse-grained triangular lattice which cannot be homogeneously split up onto the surrounding triangles. We made this point explicit in an earlier publication [3]. We believe that this could be a major step towards achieving a true microscopic classification also for chiral phases, one of the major open problems in the field.

While it is true that the present manuscript does not work out any classes of models that are not related to any other models known in the literature, the goal of the paper is to set the stage for such future endeavours.

  • "In equation 16 of their paper, the authors formulate a conjecture about Trotterization. The exponential decay seems to be in violation with the bounds derived in e.g. [M.B. Hastings, Phys. Rev. B 73, 085115 (2006)]"

We don't see the contradiction of our Trotterization conjecture with results in [M.B. Hastings, Phys. Rev. B 73, 085115 (2006)]. We would be very thankful if the referee could point out more precisely which are the bounds in that paper that seem to be violated. Note that we're not constructing MPS ground state representations, but only Trotterizing the time evolution. This is similar to the question about representing a finite-temperature density matrix as an MPO which is discussed in section IV of the above reference. Also the tensors in our path integral tensor network correspond to a constant, finite imaginary time, whereas the ground state MPS tensors correspond to a block with infinite length in time. Truncating the former is much easier than truncating the latter. We believe that the former is possible without any conditions, whereas MPS/PEPS representations are only believed to exist for gapped Hamiltonians, and even there, some doubt remains about to which extent this is possible for chiral phases in 2+1D. Our conjecture is based on the observation that in algorithms like iTEBD, the entanglement spectrum after a finite (imaginary or real) time evolution decays exponentially, and this exponential decay does not depend crucially on the chosen Trotter step. If there's evidence that the decay in the entanglement spectrum might be sensitive to the Trotter step in certain circumstances, we would be very interested to know. After all, this is still a conjecture.

  • “In the same section, the definitions used for notions such as quasi-adiabatic evolution are very confusing to the reader not yet familiar with them.”

We rewrote the corresponding paragraphs (see changes in red in Section 3), and hope that this makes it less confusing. If this is not the case, we would be very happy to hear which specific parts the Referee finds confusing.

  • "In the outlook, the authors stress that a big advantage of their method is given by the fact that "the moves turn into polynomial equations for the tensor entries; we can find roots to those equations by applying a Gauss-Newton method". The equations that they have to solve are most probably equivalent to the pentagon equations in the field of fusion category, and those are notoriously hard to solve and definitely not simply solvable by Gauss-Newton methods."

While we do believe that the proposed numerical methods can in principle find fixed-point models in a finite time, we agree that their runtime scaling depending on the bond dimension can be bad. Note that we would only consider such algorithms for low bond dimensions, where they might still be practically feasible. We have elaborated on this more in the suggested changes at the end of Section 10.

We have indeed implemented both a Gauss-Newton and a non-linear conjugate-gradient method which can in principle find the models of an arbitrary liquid (which can be directly input in a compact form). Applied to, e.g., the equations for 1+1D in Section 5 for bond dimension 4, the algorithm finds all solutions, corresponding to the trivial, complex number, 2x2 matrix, and quaternion algebras, and their direct sums. We are aware that in 2+1D, the computational cost for finding solutions might by far exceed what is doable on a laptop in a few seconds (as is the case with our current implementations), but we do not expect any fundamental obstruction to this. If the referee knows of any existing attempts of numerically finding solutions to the pentagon equation for small fusion rings, we would be very interested in a reference, as we are not aware of any.

[1] Kitaev models based on unitary quantum groupoids, Liang Chang, arXiv:1309.4181 [2] A Tensor Network Study of Topological Quantum Phases of Matter, Burak Şahinoğlu, http://othes.univie.ac.at/43085/1/44119.pdf [3] Generalized topological state-sum constructions and their universality, Andreas Bauer, arXiv:1909.03031

Attachment:

tensor_lattice_introduction.pdf

---

## Round 2 · Referee Report · Anonymous (Referee 2) · 2022-1-30

Report

This paper looks at constructing a unified diagrammatic framework to understand topological phases of matter. This is done by considering tensor networks representing Euclidean path integrals, up to certain equivalences. As examples it is shown how this framework can be applied to certain spin systems in 2+1-dimensions, and Fermionic systems in 3+1-dimensions.

The framework considered in this paper is relatively high level, which gives both its major strengths and weaknesses. As the authors concede, they do not consider how such fixed-point models may be constructed, nor does this work seek to classify such systems. Given this, it is difficult to determine what value this framework adds relative to existing approaches. While there are some aspects of this approach that markedly differ from previous work (most notably the focus on path integrals instead of ground states), it is far from clear whether these differences actually add any explanatory or predictive power to the framework.

The structure and style of the paper strike a rather nice balance between the motivation and ideas, and the nitty-gritty details. While it is impressive that their unified framework can be applied to all of the exemplary systems they considered, I don't think that alone is enough for me to recommend this work for publication in its current state. If, however, the authors could focus a little more on the unique aspects of their framework and make a convincing case that it might possess some explanatory/predictive power beyond existing approaches, then I think this work would certainly be worth of publication.

Typos/grammar mistakes:

Abstract: "is reminiscent to that of" should be "is reminiscent of".

Abstract: "We illustrate our formalism at hand of simple examples" doesn't scan. Perhaps something like "We illustrate our formalism with simple examples" or something similar.

Page 21: "1+-dimensional case", I think this should read "1+1-dimensional case"?

Page 39: "dimer cristal" should be "dimer crystal"

Page 40: "remain elusive as of to date" should be "remain elusive to date"

Page 63: "coloring" should be "colouring"

---

## Editorial Decision

published